# Shortwave Radiation on Horizontal and Incline Surfaces—One Year of Solar Radiation Measurements at Athalassa, an Inland Location in Cyprus

**Stelios Pashiardis [1], Soteris A. Kalogirou [1,\*] and Alekos Pelengaris [2]**

1 Department of Mechanical Engineering and Materials Science and Engineering, Cyprus University of Technology, P.O. Box 50329, 3603 Limassol, Cyprus
2 Department of Cyprus Public Work, Ministry of Transport Communications and Works, 1424 Nicosia, Cyprus
\* Correspondence: soteris.kalogirou@cut.ac.cy; Tel.: +357-2500-2621; Fax: +357-2500-2637

**Abstract:** Athalassa is the main actinometric station of Cyprus and is located in the center of the island at a height of about 160 m. The station is equipped with shortwave and longwave radiation instruments. The time step of the measurements is 10 min, and hourly and daily values were derived for the period of June 2020–May 2021. The solar data underwent an extensive quality control process based mainly on the suggested tests of Baseline Surface Radiation Network (BSRN) for both the hourly and daily datasets. More than 98% of the data were within the limits recommended by the BSRN and other radiation networks. A statistical analysis of the shortwave solar radiation components was then performed. Linear and quadratic relationships were established between various radiation components, and their diurnal and monthly variability was assessed. The annual average daily global radiation amount was approximately 19 MJ/m², whereas the amounts of horizontal beam and diffuse radiation were 12.9 MJ/m² and 4.7 MJ/m², respectively. Regarding the modeling of diffuse irradiance, the BRL diffuse fraction model (Boland-Ridley-Lauret) was applied. The results showed that the BRL model can satisfactorily estimate both the diffuse solar irradiance as well as the direct normal irradiance. Furthermore, the levels of the shortwave components were estimated based on the classification of four categories of the clearness index. The annual average of the direct normal beam radiation on clear days was 27.3 MJ/m², and the direct horizontal radiation was 17.7 MJ/m². Finally, the total energy received by an inclined surface was estimated based on measurements on the horizontal surfaces. In practice, photovoltaics were installed with an annual permanent slope of 26° with respect to the horizontal surface, and in a southern direction.

**Keywords:** shortwave radiation measurements; quality control; statistical analysis; energy on inclined surfaces; Cyprus

## 1. Introduction

The knowledge of the local solar radiation profile is essential to properly design solar energy systems. Solar energy is used in many applications that seek to optimize the capture of solar energy, so that it is possible to achieve energy savings and carry out sustainable energy consumption. Furthermore, solar radiation data are important for applications in meteorology, agriculture, and crop modeling as well as in the health sector [1]. The most frequent solar radiation variable registered by weather stations is global irradiance measured on a horizontal surface. Additionally, solar radiation fractions, such as direct and diffuse fractions, are required for several applications. For example, the direct component is a basic input to predict the performances and design for concentrating solar plants. Since global solar irradiance is more commonly measured at radiometric stations, the components of direct and diffuse must be modeled. Different models can be found in the literature to estimate these components [2–5]. However, because of the limited spatial coverage and

representativeness, site-based radiation observations have drawbacks when used in many regional and global applications. Therefore, remote sensing techniques are used [1,6–8].

Solar radiation (SR) is considered to be one of the most important parameters of climate elements. The amount of solar radiation received at the ground level is the main source of energy for the heating of the ground and the atmosphere, evapotranspiration, and photosynthesis. Therefore, the World Meteorological Organization (WMO) has defined the surface radiation budget as an essential climate variable (ECV) that affects the physical, chemical, and biological systems which characterize the Earth's climate [9]. It affects other climate variables and is crucial for climate change, renewable energy, agriculture, architecture, and hydrology [10].

Global irradiance ($G$) estimation and the prediction of its components, i.e., direct ($B$), and diffuse ($D$) radiation are widely discussed in the literature. Considerable work has been done since the pioneer contribution by Liu and Jordan (1960) [11]. Since then, several models and comparative analyses have been presented and discussed. An extensive review of the measurements and modeling of all shortwave radiation components was given by Muneer (2004) [2], Badescu (2008) [3], Myers (2013) [4] and more recently by Wald (2020) [5]. In general, different models exist to predict solar irradiance. Two categories of solar radiation models can be distinguished: (a) models that are based on meteorological data, such as cloudiness, sunshine duration, turbidity of the atmosphere and (b) models that use global horizontal irradiance [12]. Examples from the first category are the Bird model (1981) [13,14], METSTAT [15], the REST2 [16] and Ineichen model [17]. Details about the computation procedure of the said models are given by Myers [4]. Gueymard [12] found that these models are consistently capable of predicting solar irradiance. In the second group, we can classify the models which use the decomposition method by which global irradiance is split into its direct and diffuse components and are basically based on the clearness index [12]. In this category we could classify the work of Orgill and Hollands (1977) [18], Erbs et al. (1982) [19], Skartveit et al. (1998) [20], Boland et al. (2008) [21] and Saioa et al. (2020) [22], to mention a few of them. The separation models are still highly popular. They are often described as site-dependent in the literature, since they are essentially empirical in nature. The Bolland model was found to perform satisfactorily at different sites.

The first objective of this work is to assess the measurements obtained by pyranometers and pyrheliometer through an extensive quality control procedure and perform statistical analysis on the measured and derived parameters. This information will be useful to engineers concerned to solar energy capture systems and energy efficiency who can therefore take knowledge of the local radiation levels. High accuracy measurements of direct normal irradiances can be obtained by pyrheliometers, but these are costly and therefore are too scarce for proper temporal or spatial coverage. Then, the direct horizontal irradiance can be easily calculated, and the diffuse component can be estimated from the difference of global and direct horizontal irradiances.

The second objective of this work is the statistical analysis of the downward shortwave solar radiation components and the implementation of a BRL diffuse fraction model (Boland-Ridley-Lauret), which is a multiple predictor logistic model developed by Ridley et al. (2010) [23] and can be used to estimate the diffuse, and later, the direct radiation component. The model is a function of the clearness index ($k_t$), the apparent solar time (AST), the solar altitude ($\alpha_s$), the daily clearness index ($K_T$) and a persistence parameter $\psi$, which is an average of both a lag and lead of the clearness index. Furthermore, the levels of the shortwave components are estimated based on the classification of four categories of clearness index.

The third objective of the work concerns the estimation of total energy received by an inclined surface, based on measurements on horizontal surfaces. In practice, photovoltaics are installed with a slope with respect to a horizontal surface and in a southern direction. For the estimation of solar radiation on an inclined surface, there are three types of solar radiation, namely, (i) beam; (ii) diffuse; and (iii) reflected solar radiation from

the surfaces surrounding it. Hence, there are three conversion factors, i.e., for beam ($R_B$), diffuse ($R_D$), and reflected ($R_R$) solar radiation [24]. These conversion factors convert the beam and diffuse solar radiation of a horizontal surface to those of an inclined surface.

The above methodology of analysis will give valuable information concerning the application of solar radiation in renewable energy resources projects.

## 2. Materials and Methods

### 2.1. Site Topography Description and Climate Conditions

Athalassa Radiosonde Station is located within a forest area at the height of about 160 m, which is about 10 Km distance from the center of Nicosia in the eastern part of the city and is managed by the Meteorological Department (Figure 1). The local coordinates of the station and the annual values of various climatological parameters are shown in Table 1. Three different types of meteorological stations operate simultaneously:

- Climatological station;
- Radiosonde Upper Air station;
- Actinometric station.

**Table 1.** Station's coordinates and mean annual air temperature ($T_a$), mean annual precipitation (Prec) and mean annual number of hours of sunshine duration.

| Station | Long. (E) (deg.) | Lat. (N) (deg.) | Elev. (m) | Distance from Coast (km) | $T_a$ (°C) | Prec. (mm) | Sunshine Duration (hrs) |
|---|---|---|---|---|---|---|---|
| Athalassa | 33.396 | 35.141 | 158 | 20 | 19.8 | 315 | 3285 |

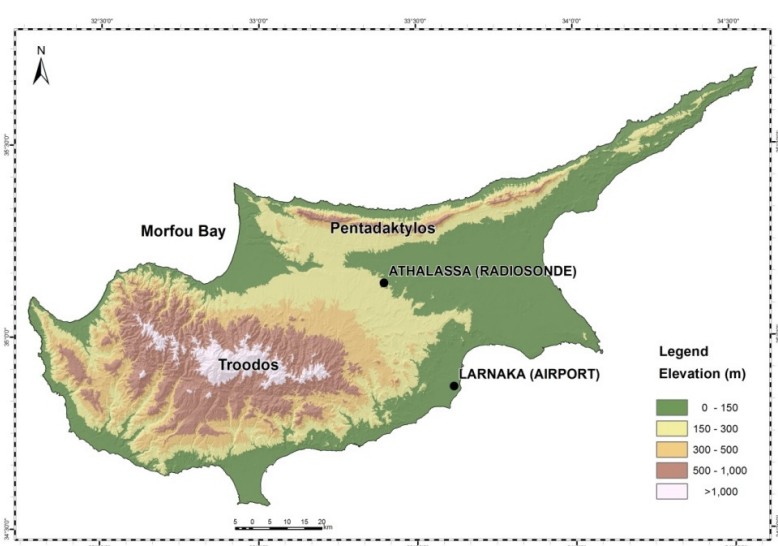

**Figure 1.** Map of Cyprus showing the location of Athalassa actinometric station.

The average air temperature in summer is about 30 °C, while during the winter it is about 11 °C. The maximum air temperature exceeds the value of 40 °C in some days in the summer, while the lowest air temperature reaches −3 °C in winter. The highest recorded air temperature is 45.6 °C. The annual average number of days with air frost is 2, while ground frost is recorded as 17 days. The average annual rainfall is about 315 mm and occurs between October and May. The summer season is dry with clear sky conditions. Periodically, the island is under the effect of the Saharan Air Layer which is characterized by high content of mineral dust. Dust conditions are more frequent in spring and autumn, although they are observed in some cases in winter and summer. The prevailing wind direction is westerly. The annual mean daily sunshine duration is 9.0 h, while the annual mean daily pan evaporation is about 6.0 mm. The average annual daily global radiation is 18.5 MJ/m$^2$ and the cumulative annual irradiation is about 6760 MJ/m$^2$. The annual

amount of sunshine duration is about 3285 h. Clear and partly cloudy days exceed the value of 80% of the total number of days in a year.

*2.2. Solar Radiation Measurements*

As is indicated in Table 2, the actinometric station is equipped with pyranometers for measuring global, diffuse, and reflected solar radiation, a pyrheliometer for measuring direct normal radiation as well as the sunshine duration and photosynthetic photon flux density for measuring global and diffuse PAR radiation components. All instruments are obtained from Kipp & Zonen Company except the Quantum sensors of PAR which were obtained from Light Company. All instruments are factory calibrated. The shortwave components (Global (*G*), Diffuse (*D*), (Direct Normal ($B_n$) and Diffuse PAR (*DPPFD*)) are installed on a 2AP solar tracker system. Diffuse radiation (*D*) and *DPPFD* are protected from the direct sunlight with a shading ball arm. PAR is expressed either in terms of Photosynthetic Photon Flux density (*PPFD*) in units of μmol /s/m$^2$, since ph otosynthesis is a quantum process, or in terms of energy (PAR, Photosynthetically Active Radiation) in units of W/m$^2$, which is more suitable for energy balance studies. *PPFD* is recorded in the database, and is converted into energy units according to the McCree (1972) [25] conversion factor of 4.57 μmol/J.

All the instruments are checked on a daily basis by the meteorological staff of the Radiosonde station. Routine checks consist of cleaning the domes, inspection of cable connections and instrument leveling, checking the proper functioning of the solar tracker system including the shading of the pyranometer which measures the diffuse radiation, as well as the proper orientation of the pyrheliometer.

The period of measurements is June 2020 to May 2021. A Campbell Scientific Instrument data-logger (model CR10) located at the site, monitors and stores the data at 10 min intervals. The sensors are scanned every 10 s and average, maximum and minimum values at the 10 min intervals are calculated and stored. The stored data are downloaded to a desktop computer periodically. The data refer to GMT time (LST = GMT + 2 h) in winter and LST = GMT + 3 h in summer. A database was developed in order to manage the large volume of data. The database system is also used for checking the real-time measurements, becoming therefore, a comprehensive quality control system through graphics and tables or transferring the raw data to a special statistical package or spreadsheet program. The data management includes daily or monthly reports which can be used effectively for quality control of the data and detects any missing data. Details about the method of estimation of various solar variables and the quality control procedure which was followed are given in the next sections.

*2.3. Estimation of Solar and Extraterrestrial Irradiances on a Horizontal Surface*

The horizontal beam irradiance (*B*) can be calculated from the normal beam irradiance ($B_n$) from the following equation [26]:

$$B = B_n * \cos \theta_z \tag{1}$$

where $\theta_z$ is the solar zenith angle (SZA) which is given by:

$$\cos \theta_z = \cos \phi * \cos \delta * \cos \omega + \sin \phi * \sin \delta \tag{2}$$

where, $\phi$ is the latitude of the location, $\delta$ is the solar declination angle (deg) and $\omega$ is the hour angle (deg). The solar declination is estimated by the following equation:

$$\delta = 23.45 * \sin[360 * (284 + d_n)/365] \tag{3}$$

where $d_n$ is the day number of the year. Then, the hourly global irradiance (*G*) is the sum of diffuse and direct horizontal irradiance:

$$G = D + B \tag{4}$$

**Table 2.** Solar Shortwave Radiation equipment installed at the actinometric station of Athalassa.

| Parameter | Symbol | Unit | Type | Model | Manufacturer | Spectral Range | Calibration Factor | Max Output | Note |
|---|---|---|---|---|---|---|---|---|---|
| Global Horizontal Irradiance | $G_t$ | W m$^{-2}$ | Pyranometer | CM6B | Kipp&Zonen | 305–2800 nm | 12.93 µV/(W m$^{-2}$) | 1500 W m$^{-2}$ | On 2AP Suntracker |
| Direct Normal Irradiance | $B_n$ | W m$^{-2}$ | Pyrheliometer | CH1 | Kipp&Zonen | 350–1500 nm | 11.03 µV/(W m$^{-2}$) | 1500 W m$^{-2}$ | On 2AP Suntracker |
| Direct Horizontal Irradiance | $B$ | W m$^{-2}$ | Pyrheliometer | CH1 | Kipp&Zonen | 350–1500 nm | 11.03 µV/(W m$^{-2}$) | 1500 W m$^{-2}$ | $B = B_n *\cos\theta_z$ ($\theta_z$ = Solar Zen. Angle) |
| Sunshine Duration | $S$ | W m$^{-2}$ | Pyrheliometer | CH1 | Kipp&Zonen | 350–1500 nm | 11.03 µV/(W m$^{-2}$) | 1500 W m$^{-2}$ | Time in min when $B_n > 120$ W/m$^2$ |
| Diffuse Horizontal Irradiance | $D$ | W m$^{-2}$ | Pyranometer | CM6B | Kipp&Zonen | 305–2800 nm | 9.88 µV/(W m$^{-2}$) | 1500 W m$^{-2}$ | On 2AP with shading ball arm |
| Reflected Horizontal Irradiance | $R$ | W m$^{-2}$ | Pyranometer | CM11 | Kipp&Zonen | 310–2800 nm | 5.12 µV/(W m$^{-2}$) | 1500 W m$^{-2}$ | Albedometer |
| Photosynthetic Photon Flux Dens. | $GPPFD$ | µmol s$^{-1}$m$^{-2}$ | Quantum Sens. | Li190SA | Light Comp. | 400–700 nm | 347.087 µmol s$^{-1}$m$^{-2}$mV$^{-1}$ | 2500 µmol s$^{-1}$m$^{-2}$ | 4.57 µmol s$^{-1}$m$^{-2}$ $PPFD = 1$ Wm$^{-2}$ PAR |
| Diffuse PPFD | $DPPFD$ | µmol s$^{-1}$m$^{-2}$ | Quantum Sens. | Li190SA | Light Comp. | 400–700 nm | 4.96 µmol s$^{-1}$m$^{-2}$µV$^{-1}$ | 2500 µmol s$^{-1}$m$^{-2}$ | On 2AP with shading ball arm |

The irradiance falling on a plain at normal incidence at the top of the atmosphere ($G_{0n}$) can be estimated from the following equation:

$$G_{0n} = G_{sc} * [1 + 0.033 * \cos(360 * d_n/365)] \ \left(W/m^2\right) \tag{5}$$

where $G_{sc}$ is the solar constant (1367 W/m²). According to Wald (2020) [5] the respective solar constant for PAR is 534 W/m².

Then, the irradiance on a horizontal plain at the top of the atmosphere can be estimated by the following equation:

$$G_0 = G_{0n} * \cos\theta_z = G_{0n} * (\cos\phi * \cos\delta * \cos\omega + \sin\phi * \sin\delta) \ \left(W/m^2\right) \tag{6}$$

The daily total global irradiation which is obtained by a horizontal plain at the top of the atmosphere ($G_{0d}$) is given by the following equation:

$$G_{0d} = (24 * 3.6/\pi) * G_{0n} * [(\cos\phi * \cos\delta * \sin\omega_s + (\pi * \omega_s/180) * \sin\phi * \sin\delta)] \ \left(kJ/m^2\right) \tag{7}$$

where $\omega_s$ is the sunset hour angle and is given by:

$$\omega_s = \cos^{-1}(-\tan\phi * \tan\delta) \tag{8}$$

The daily sums of global and horizontal beam radiation are obtained from the hourly values. Furthermore, the astronomical day length ($S_{0d}$) which is the computed time during which the center of the solar disk is above an altitude of zero degrees (without allowance for atmospheric refraction) is given by:

$$S_{0d} = (1/7.5) * \cos^{-1}(-\tan\phi * \tan\delta) \ (h) \tag{9}$$

The clear sky irradiance was estimated using the Haurwitz (1945) [27] model which is a function of the solar zenith angle. This model was tested by Ianetz et al. (2007) [28] in Israel showing high performance. The equation involved in the said model is shown below:

$$G_c = 1098 * \cos\theta_z * \exp(-0.057/\cos\theta_z) \tag{10}$$

Almost a similar equation was obtained using the maximum values of global irradiances of the study station:

$$G_c = 2170.5 * (\cos\theta_z)^{-0.175} * \exp(-0.578/\cos\theta_z) \tag{11}$$

### 2.4. Estimation of Radiation Indices

The following indices are used for the assessment of radiation levels on the surface: $k_n$, $k_t$, $k_b$, and $k_d$. $K_n$ is the ratio of direct normal irradiance ($B_n$) to the extraterrestrial one ($G_{0n}$). The ratio of horizontal beam irradiance ($B$) to the global irradiance $G$ is defined as $k_b$; $k_d$ is the fraction of diffuse irradiance ($D$) to the global ($G$); $k_t$ is the clearness index, i.e., the ratio of global irradiance to the horizontal extraterrestrial irradiance:

$$k_n = B_n/G_{0n} \tag{12}$$

$$k_t = G/G_0 \tag{13}$$

$$k_b = B/G \tag{14}$$

$$k_d = D/G \tag{15}$$

The upper bound of the ratios is 1, although $k_d$ can be slightly higher than 1 (i.e., 1.15). This usually occurs in the morning after sunrise or in the afternoon before sunset when the solar elevation angle is low, and the diffuse irradiance is slightly higher than that of global irradiance due to the cosine effect of the lower sun angles. Therefore, to avoid this problem, cases of diffuse irradiances below 5° of sun elevation angle were excluded from the data

set [29]. Only few observations are considered invalid when low $k_d$ values are associated with low $k_t$ values. This occurs when the sensor is covered with environmental debris. The above ratios can be obtained from both the hourly and daily values. The capital letters and subscripts represent the daily values, while the small letters represent the hourly values.

Sky Conditions Classification

Prior to the analysis of the global solar irradiance, it is constructive to categorize the sky conditions at Athalassa. For this purpose, four sky categories have been proposed by Escobedo et al. (2009) [30], based on the relation between hourly irradiances of global, direct and diffuse radiation and the clearness index ($k_t$). Figure 2 indicates the block-averaged curves for global, direct and diffuse solar radiation in terms of $k_t$ (0–1) into subintervals of 0.05 sizes. The four intervals of sky conditions are defined as follows:

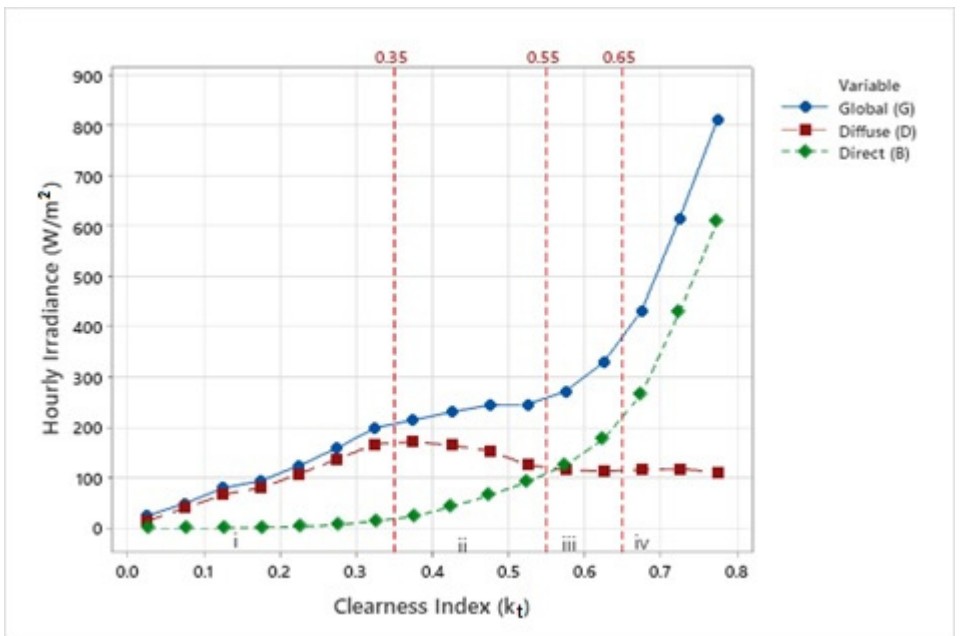

**Figure 2.** Block-average values of global, diffuse and direct components of solar radiation, at the surface, in terms of clearness index intervals. Vertical dashed lines define regions of $k_t$ values associated to sky conditions and identified by i (cloudy sky), ii (partially cloudy with predominance of diffuse component), iii (partially cloudy with predominance of direct component) and iv (clear sky) interval.

*Interval i*: $k_t \leq 0.35$ and the direct component of the global solar radiation at the surface is practically zero. Therefore, global and diffuse solar radiations are equal, and the sky condition is defined as totally covered by clouds or cloudy sky;

*Interval ii*: $0.35 < k_t \leq 0.55$. The global solar radiation is composed by a fraction of diffuse component which is larger than the fraction of direct component, and the diffuse fraction is decreasing with increasing $k_t$. The upper limit of this interval is set where diffuse equals direct component of the solar radiation (approximately at 100 W/m$^2$ and at $k_t$ = 0.55). In this case, the sky condition is defined as partially cloudy with predominance of diffuse component of the solar radiation because the radiation field is predominantly composed by diffuse radiation;

*Interval iii*: $0.55 < k_t \leq 0.65$. The global solar radiation is composed by a fraction of diffuse component that is smaller than the fraction of direct component and the diffuse fraction is decreasing with $k_t$ until 0.65, which is considered as the end of the partially cloudy interval. In this case, the sky condition is partially cloudy with the predominance of a direct component of the solar radiation;

*Interval iv*: $k_t > 0.65$. The global solar radiation at the surface is composed by mainly the direct component of solar radiation and the diffuse contribution is very small, indicating that there is no significant cloud cover. This interval is characterized as clear sky.

The above classification of the four categories is also confirmed by plotting the daily clearness index with the daily relative sunshine duration ($\sigma = S_d/S_{0d}$) as shown in Figure 3.

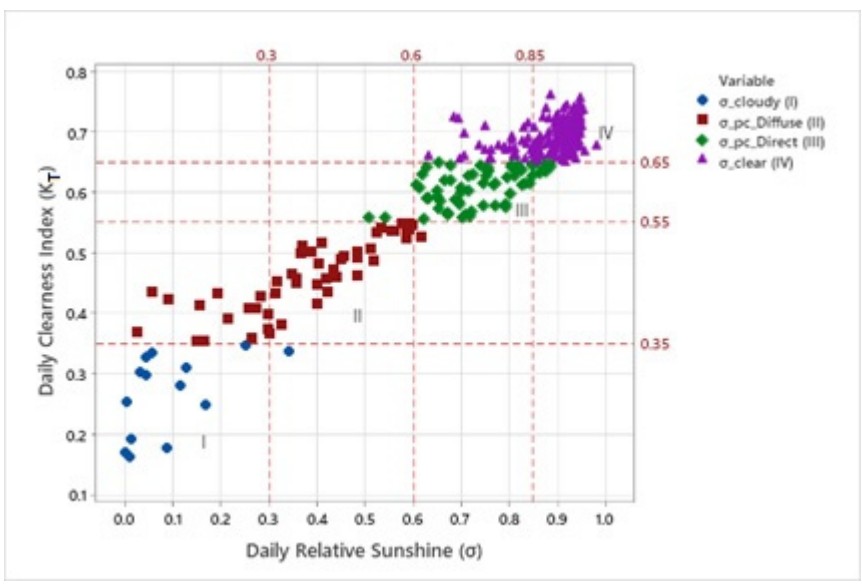

**Figure 3.** Relationship between the daily clearness index ($K_T$) and relative sunshine duration ($\sigma$) which define the four classes of sky conditions.

### 2.5. Quality Control Procedure

With any ground measurement there can be errors in the data that can be systematic or generated by the used instrument. According to Moradi (2009) [31], Muneer and Fairooz (2002) [32] and Younes et al. (2005) [29], any expected source of errors or problems in measuring solar radiation can be classified into three main types:

- Equipment errors and their uncertainties (cosine response, azimuth response, temperature response, spectral selection, stability, non-linearity, shade-ball misalignment);
- Operation related problems and errors (complete or partial shade-ring misalignment, dust, snow, dew, water droplets, bird droppings, etc.);
- Incorrect sensor leveling, shading caused by building structures, electric fields in the vicinity of cables, mechanical loading of cables, orientation and/or improper screening of the vertical sensors from ground-reflected radiation, station shut-down, etc.

Therefore, the stored data were subjected to various quality control tests. The quality control procedure was based mainly on the recommendations provided by the Baseline Surface Radiation Network (BSRN) and other research centers [33–39]. The results of BSRN tests and some comparison tests are presented in the next section.

### 2.5.1. Time Series Plots

Before proceeding with the quality control routine, a visualization of the time series is required. The graphical presentation of the data gives us the opportunity to detect any major problem with the data. It is possible to examine the type of the data we are dealing with, to identify their trends and to easily examine them in order to check if there are any missing values, lags or spikes. The time series plots of the solar shortwave and Total Ultraviolet (UVT) radiation components for the period June 2020 to May 2021 are shown in Figure 4, without any abnormal values.

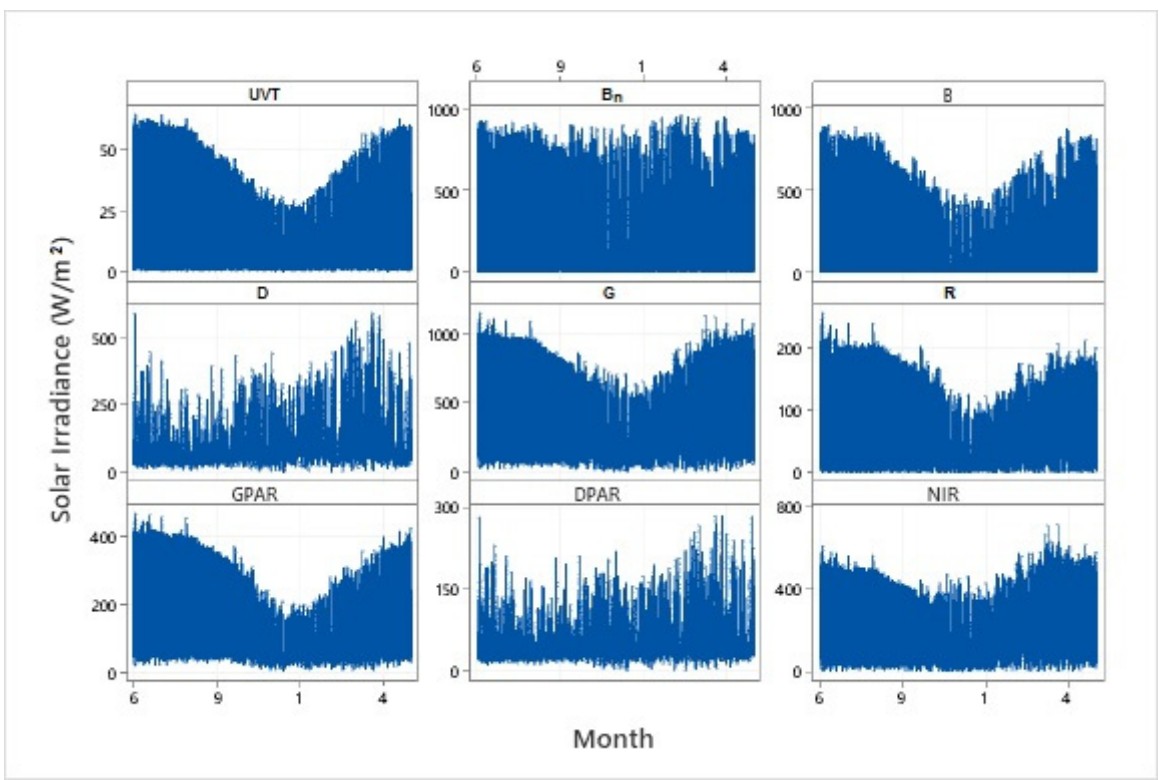

**Figure 4.** Time series plots of solar shortwave radiation components during the period June 2020–May 2021.

By analyzing these plots, we can notice the extreme values of the measured meteorological and radiation parameters. The highest temperature reached the value of 44.8 °C which was recorded on the 4th of September of the year 2020, while the lowest was slightly below 0 °C ($-0.01$ °C). The highest value of UVT irradiance is slightly above 65 W/m$^2$ which is occurred in the summer months.

The highest direct normal irradiance was 968 W/m$^2$, while the highest values of diffuse irradiances were recorded in spring of the year 2021, when frequent events of dust are observed. Generally, diffuse irradiances are lower than 700 W/m$^2$ which is the upper threshold given in the literature [5]. Extreme values were recorded for global irradiance reaching values around 1400 W/m$^2$. These values occur during partially cloudy conditions, when total irradiance may be enhanced due to the reflection of solar irradiance from the base of the clouds and from scattering of direct irradiance due to cloud particles. Such periods of enhanced solar irradiance may last from several seconds to some minutes depending on the cloud motion. Tapakis and Charalambides (2014) [40] have detected values of global irradiance exceeding 1500 W/m$^2$ in Cyprus, that correspond to 150% of the theoretical value computed by clear sky models. An example of such partially cloudy conditions is recorded on the 10th of April 2021. The daily sunshine duration on that day was about 2 h and the highest global irradiance reached the value of 1426 W/m$^2$. Similar cases were observed in more than 10 days of April, May and June when the global irradiance exceeded 1200 W/m$^2$ (enhanced by 125% compared to the theoretical value of clear sky conditions). Such situations were also observed in various areas of the world [40].

The maximum reflected irradiance is 255 W/m$^2$, while maximum *GPAR* is 470 W/m$^2$. The maximum value of Near Infrared Radiation (*NIR*) is about 710 W/m$^2$. *NIR* is estimated from the difference of global and the sum of *UVT* and *GPAR* radiation components:

$$NIR = G - (UVT + GPAR) \tag{16}$$

In order to deepen the visual analysis of the data presented in the previous paragraph, we display the measurement in two-dimensional representation, where the *x*-axis represents the day of the year, the *y*-axis the hour of the day, and the dots color demonstrate the value of the data considered as shown by the color scale from blue to red. This analysis allows visual identification of errors over time, to identify missing values, issues in time reference and abnormal values (Figure 5). In the different cells of Figure 5, the two black lines represent the theoretical sunrises and sunsets. The figure shows no abnormal conditions for global radiation. Similar patterns are also observed for the other variables. The plausibility analysis of the radiation parameters will be carried out in the next section as part of the QC of the measurements.

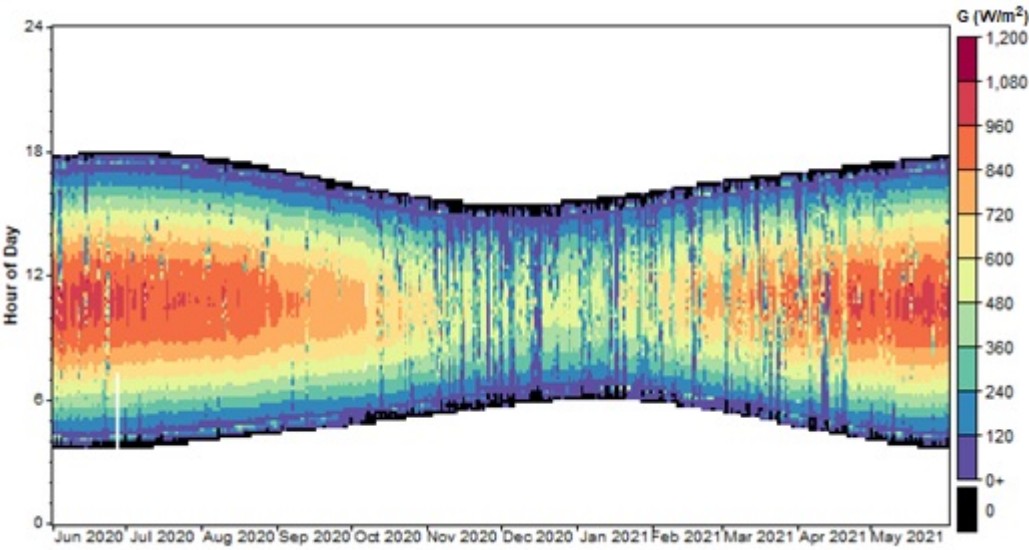

**Figure 5.** Two-dimensional representation of global radiation (*G*).

2.5.2. BSRN Quality Control Procedure

Tests on increments (or step tests) in irradiation is very difficult to define due to the very nature of the solar radiation, since the appearance of a cloud may induce a sudden drop in irradiance which may reach several hundred W/m$^2$. Fluctuations from moment to moment can be very large, and this prevents from setting effective upper and lower bounds. A maximum difference of 1000 W/m$^2$ could be imposed on the irradiance, but this limit is so high that the test would be ineffective. Estevez et al., (2011) [41] have suggested a value of 555 W/m$^2$, which is the case for at least 95% of measurements.

The BSRN QC procedure for the one component tests is based on three tests: the Physically Possible Limits (PPL), the Extremely Rare Limits (ERL) and the Configurable Climatological Limits (CCL) [33]. The PPL procedure is introduced for detecting extremely large errors in radiation data, while the ERL procedure is used to identify measurements exceeding the extremely rare limit. Radiation data exceeding these limits normally occur under very rare conditions and over very short time periods (Table 3). The configurable climatological limits are based on long period of records in the Southern Great Plains of United States. The analysis of the tests was implemented through various software packages, such as Windographer Ver. 4, Minitab ver. 20 and Excel. For each test, a QC flag is defined. If the test is successful, the corresponding flag takes the value of zero, otherwise it takes the value of 1. Thus, a flag of one indicates that the variables involved in the test are erroneous or at least suspect.

The PPL tests check the maximum and minimum limits that can be physically reached by irradiance and aim at detecting extremely large errors in the radiation data. The upper limits of short-wave irradiances depend on the solar zenith angle. The minimal value of solar irradiance must be 0 W/m$^2$, although because of the radiative cooling at night, the limit is set at $-4$ W/m$^2$. The test is applied independently to each of the solar radiation

components shown in Table 3 [33]. $G_{0n}$ is the normal irradiance at the top of the atmosphere and $\theta_z$ is the solar zenith angle (SZA). The horizontal irradiance at the top of the atmosphere is estimated by Equation (6), while the clear sky irradiance was estimated by Equation (11) using data from the study station. The model of clear sky irradiance underestimates the global irradiances at low solar elevation angles. Any measurements outside the PPL limits are excluded from the analysis. As indicated from the graphs of Figure 6, all the data are within the said PPL limits. The blue line represents the PPL limit and the red line the ERL limit, respectively.

**Table 3.** BSRN-Physically Possible Limits (PPL), Extremely Rare Limits (ERL), and Configurable Climatological Limits (CCL) [27]. $L_{dn}$ and $L_{up}$ are the downward and upward longwave irradiances.

| Parameter | Lower Bound (W/m²) | | Upper Bound (W/m²) | | Upper Bound (W/m²) | | Level |
|---|---|---|---|---|---|---|---|
| | PPL | ERL | PPL | ERL | CCL | | |
| $G$ | −4 | −2 | $\min(1.2{*}G_{0n}, 1.5{*}G_{0n}{*}(\cos\theta z)^{1.2} + 100)$ | $1.2{*}G_{0n}{*}(\cos\theta z)^{1.2} + 50$ | $0.97{*}G_{0n}{*}(\cos\theta z)^{1.2} + 55$ | | 2nd |
| | | | | | $0.92{*}G_{0n}{*}(\cos\theta z)^{1.2} + 50$ | | 1st |
| $D$ | −4 | −2 | $\min(0.8{*}G_{0n}, 0.95{*}G_{0n}{*}(\cos\theta z)^{1.2} + 50)$ | $0.75{*}G_{0n}{*}(\cos\theta z)^{1.2} + 30$ | $0.58{*}G_{0n}{*}(\cos\theta z)^{1.2} + 35$ | | 2nd |
| | | | | | $0.52{*}G_{0n}{*}(\cos\theta z)^{1.2} + 30$ | | 1st |
| $B_n$ | −4 | −2 | $G_{0n}$ | $0.95{*}G_{0n}{*}(\cos\theta z)^{0.2} + 10$ | $0.86{*}G_{0n}{*}(\cos\theta z)^{0.2} + 15$ | | 2nd |
| | | | | | $0.82{*}G_{0n}{*}(\cos\theta z)^{0.2} + 10$ | | 1st |
| $B$ | −4 | −2 | $G_{0n}{*}(\cos\theta z)$ | $0.95{*}G_{0n}{*}(\cos\theta z)^{1.2} + 10$ | $0.86{*}G_{0n}{*}(\cos\theta z)^{1.2} + 15$ | | 2nd |
| | | | | | $0.82{*}G_{0n}{*}(\cos\theta z)^{1.2} + 10$ | | 1st |
| $R$ | −4 | −4 | $1.2{*}G_{0n}{*}(\cos\theta z)^{1.2} + 50$ | $G_{0n}{*}(\cos\theta z)^{1.2} + 50$ | $0.95{*}G_{0n}{*}(\cos\theta z)^{1.2} + 55$ | | 2nd |
| | | | | | $0.87{*}G_{0n}{*}(\cos\theta z)^{1.2} + 50$ | | 1st |
| $L_{dn}$ | 40 | 60 | 700 | 500 | LL: 145 | UL: 500 | 2nd |
| | | | | | LL: 190 | UL: 465 | 1st |
| $L_{up}$ | 40 | 60 | 900 | 700 | LL: 210 | UL: 630 | 2nd |
| | | | | | LL: 240 | UL: 590 | 1st |

The graphs show that a very large majority of values pass the said tests. Some values of global irradiances are closed to the ERL limits (Figure 6). As explained earlier, these values occur during partially cloudy conditions, when total irradiance may be enhanced due to the reflection of solar irradiance from the base of the clouds and from scattering of direct irradiance due to cloud particles.

The suggested climatological limits are based on data obtained from Great Plains in the United States. The limits are classified in two levels. The '1st level' have the 'smallest' testing limits, while those referred as '2nd level' gives the second lower limits. The graphical representation of PPL, ERL, and CCL limits of all the shortwave components are shown in Figures 6 and 7. Both global and diffuse irradiances show some values above the 1st and 2nd level limits. Regarding the global radiation there are only 21 and 64 cases above the limits of 1st and 2nd level, respectively. Diffuse irradiances showed lower cases (6 and 33, respectively).

2.5.3. Comparison Tests

The tests of this category refer to (a) comparisons between the average and extreme values of the shortwave components, (b) comparisons between the pyranometers which are installed at the station, (c) comparison of diffuse irradiance with the Rayleigh limit, (d) comparisons of the radiation indices, and (e) comparison tests based on three radiation components ($G = B + D$). The results of these comparisons showed high quality of the data. Regarding the comparisons of the four pyranometers, the process showed that the coefficients of determination ($R^2$) are close to 1.

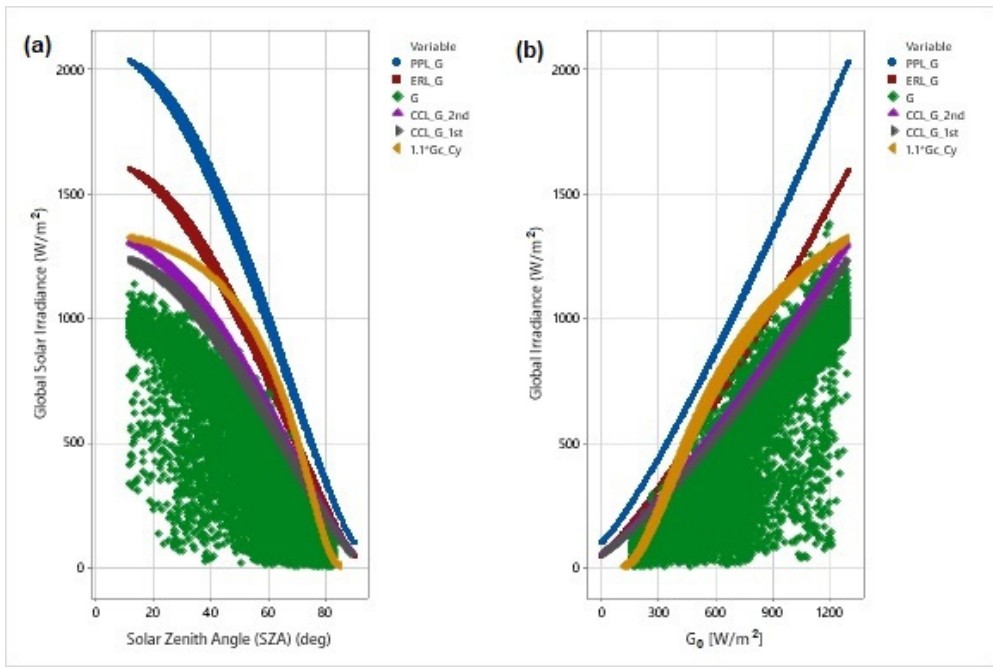

**Figure 6.** (**a**) graphical representation of PPL, ERL and CCL tests of the average 10 min irradiances of the pyranometer on solar tracker (*G*) as a function of solar zenith angle (SZA); and (**b**) graphical representation of PPL, ERL and CCL tests of the maximum 10 min irradiances of the pyranometer on solar tracker (*G*) as a function of extraterrestrial irradiance at the top of the atmosphere ($G_0$).

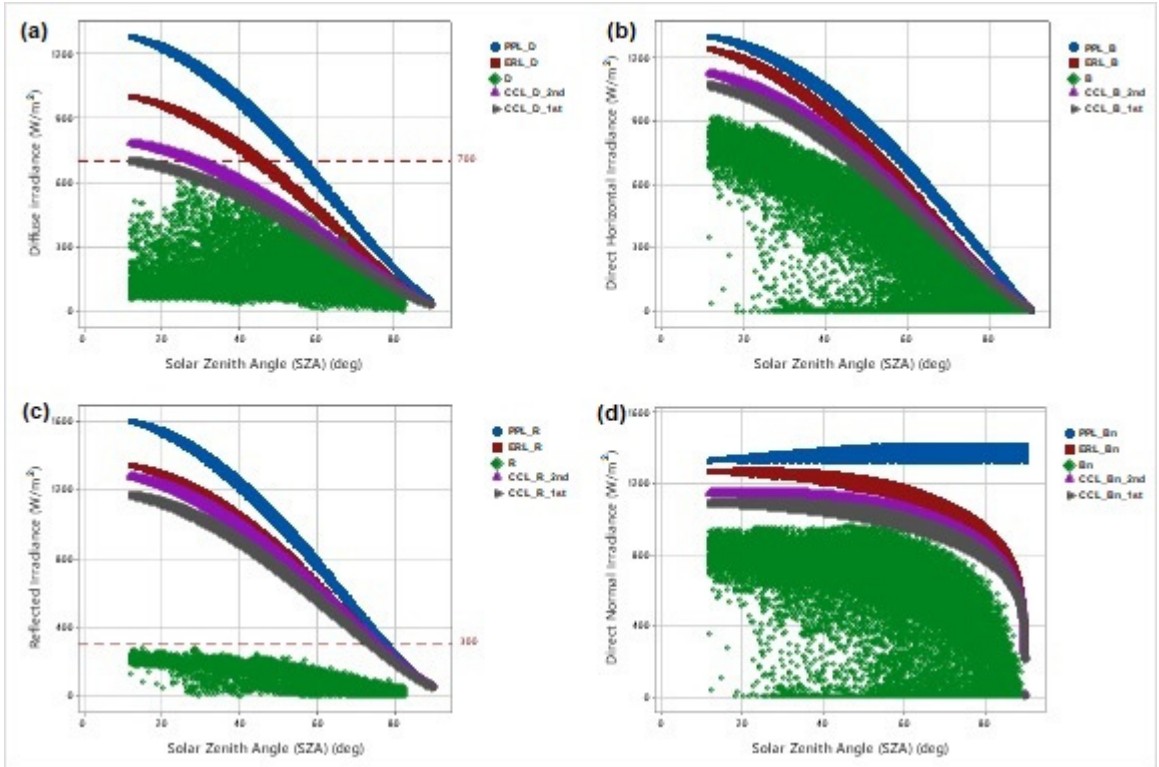

**Figure 7.** Graphical representation of PPL, ERL and CCL tests of the average 10 min irradiances of: (**a**) the diffuse on solar tracker (*D*); (**b**) direct horizontal (*B*); (**c**) reflected (*R*); and (**d**) direct normal irradiance as a function of solar zenith angle (SZA).

Regarding diffuse irradiance the upper limit is 700 W/m$^2$, while the lower limit is the Rayleigh limit which is estimated during clear-sky conditions. The Rayleigh diffuse limit ($R_L$) is estimated from the following formula [33]:

$$R_L = 209.3 * \mu - 708.3 * \mu^2 + 1128.7 * \mu^3 - 911.2 * \mu^4 + 287.85 * \mu^5 + 0.046725 * P \quad (17)$$

where $\mu$ is the cosine of the solar zenith angle and $P$ is the station surface pressure in millibars. Then, the limit of diffuse irradiance is defined as:

$$D > R_L - 1 \text{ for } k_d < 0.8 \text{ and } G > 50 \text{ W/m}^2 \quad (18)$$

The results of this test are shown in Figure 8. Only, limited number of points is lower than $R_L$ limit. The points are located mostly at SZA > 60°.

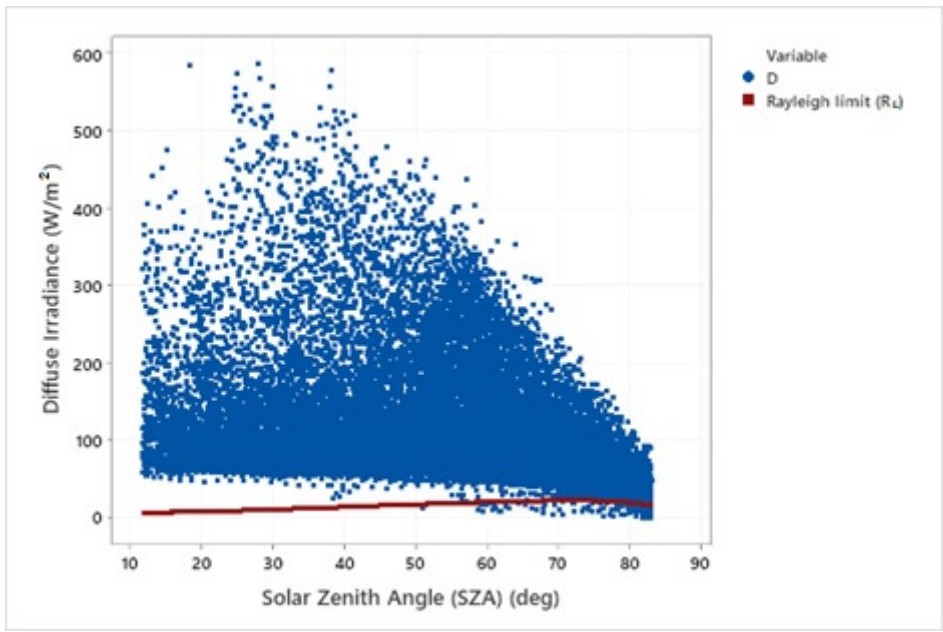

**Figure 8.** Comparison based on the Rayleigh limit for data points that have $k_d < 0.8$ and $G > 500$ W/m$^2$.

The radiation indices tests refer to the diffuse fraction and to NREL SERIS K tests. The diffuse fraction test involves the examination of the consistency of independent measurements. In this case, the components of global and diffuse irradiances are compared based on the following relations, for cases of global > 50 W/m$^2$ [42]:

$$k_d = D/G < 1.05 \text{ for SZA} < 75° \quad (19)$$

$$k_d = D/G < 1.1 \text{ for } 75° < \text{SZA} < 93° \quad (20)$$

All the measurements are lower than the limits shown in the above equations.

The NREL-SERIS K tests were originally proposed by NREL with solar irradiance normalized to extraterrestrial values. The tests are based on the calculation of the diffuse fraction ($k_d$), the clearness index ($k_t$) and the direct normal transmittance ($k_n$). We can define two tests to qualify 'k-values': a test in the $k_d$-$k_t$ space and a second in the $k_n$-$k_t$ space. The limits of measurements in the $k_d$-$k_t$ space as defined by Geuder et al. (2015) [35] are:

- $k_d < 1.05$ for SZA < 75°;
- $k_d < 1.1$ for SZA > 75°;
- $k_d < 0.95$ for $k_t > 0.6$;
- $k_t < 1$

The tests performed in the $k_n$-$k_t$ space have the following limits:

- $k_n > k_t$;
- $k_n < 0.8$;
- $k_t < 1$.

The results of this quality control process are shown in Figure 9a,b. The dotted red lines represent the different tests that are conducted. As it is indicated from the graphs all the measurements satisfy the test conditions, except three points that exceed the limits of $k_t$ and these are the cases of enhancing global irradiances due to the reflections of the clouds. In Figure 9a, the limits of various situations are noted. For example, the limits for $k_t < 0.2$ and $k_d < 0.85$, show overcast conditions [42]. The condition $k_t > 0.5$ and $k_d > 0.8$ tests the misalignment [36], while Muneer (2004) [2] suggested that if there are observations in the space of $0.2 < k_t < 0.5$ and $k_d < 0.2$, then debris are present in the dome of the sensor.

Similar results for the control envelopes are obtained through a statistical outlier analysis as proposed by Younes et al., (2005) [29]. The whole $k_t$ range (from 0 to 1) was divided into ten equal intervals. For every interval the mean and standard deviation of the diffuse fraction ($k_d$) and the direct normal transmittance ($k_n$) were calculated. From this information an envelope may be drawn that connects those points that, respectively, represent the top ($\bar{k}_d + 3 * \sigma_{k_d}$) and bottom curves ($\bar{k}_d - 3 * \sigma_{k_d}$). In a similar way the top and bottom curves are defined for the $k_n$ vs. $k_t$ graph. A mathematical description of the envelope is obtained by fitting second degree polynomial curves (Figure 9c,d). The quadratic equations obtained for the lower and upper curves for the data sets of $k_d$ vs. $k_t$ are the following:

$$flower(k_t) = 0.723 - 1.355 * k_t + 0.539 * k_t^2 \quad R^2 = 0.98 \tag{21}$$

$$fupper(k_t) = 0.763 + 1.007 * k_t - 1.254 * k_t^2 \quad R^2 = 0.98 \tag{22}$$

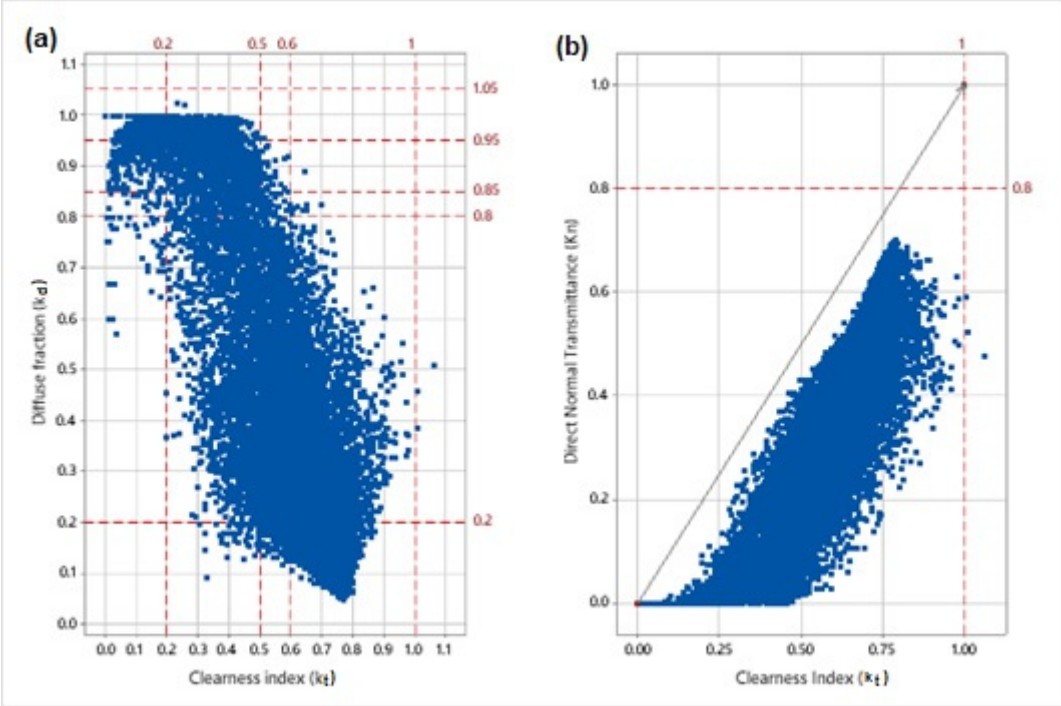

**Figure 9.** *Cont.*

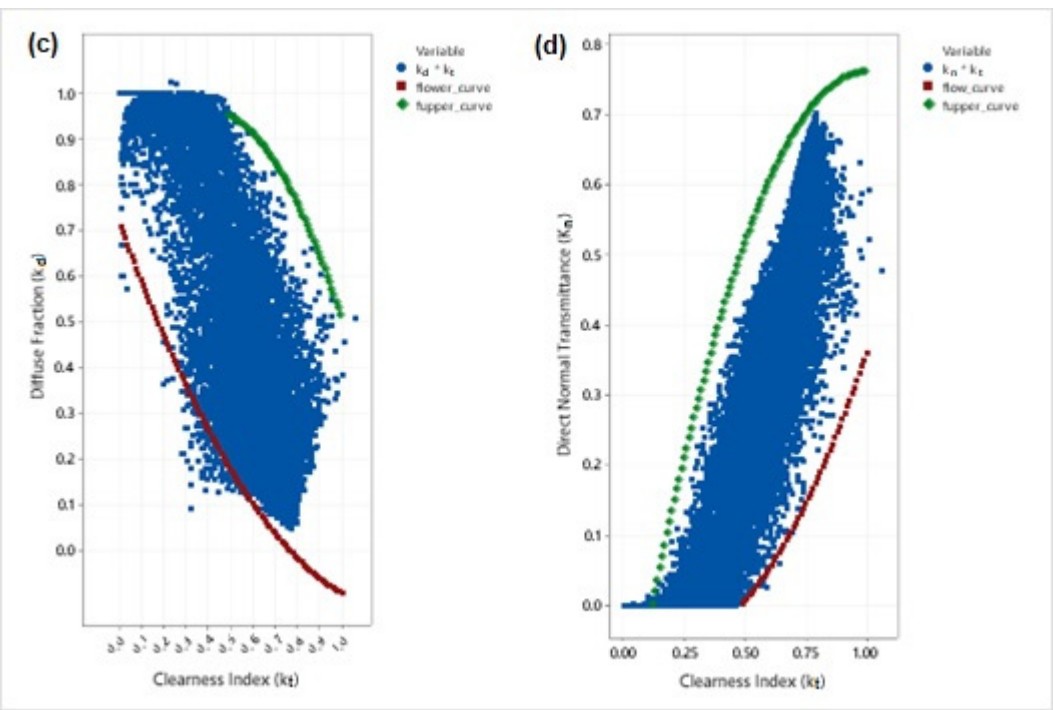

**Figure 9.** Scatter plot of 10-min values in: (**a**) $k_d$-$k_t$ space and (**b**) $k_n$-$k_t$ space and limits for quality control tests. Scatter plot of 10-min values in (**c**) $k_d$-$k_t$ space and (**d**) $k_n$-$k_t$ space and limits for quality control tests.

The respective equations for the graph of $k_n$ vs. $k_t$ have the following forms:

$$flower(k_t) = -0.041 - 0210 * k_t + 0.612 * k_t^2 \; R^2 = 0.70 \tag{23}$$

$$fupper(k_t) = -0.219 + 1.963 * k_t - 0.982 * k_t^2 \; R^2 = 0.93 \tag{24}$$

Figure 9c shows that some points are outside the limits defined by the lower curve, coinciding with the graph of Figure 9a, as shown with the reference curves.

Finally, the three-component test is intended to compare the measured global ($G$) irradiance by the pyranometer and the calculated sum from diffuse ($D$) and direct horizontal irradiance ($B$) which are measured independently:

$$0.92 \leq \frac{G}{B + D} \leq 1.08 \text{ for SZA } \leq 75° \text{ and } G > 50 \text{ W/m}^2 \tag{25}$$

$$0.85 \leq \frac{G}{B + D} \leq 1.15 \text{ for } 75° < \text{SZA} \leq 93° \text{ and } G > 50 \text{ W/m}^2 \tag{26}$$

Ideally, the ratio of measured and estimated global irradiance should be 1.0, but due the instruments' inaccuracy, values far from unity, are often obtained. However, it is not possible to detect which of the three components is wrong. Figure 10a shows the regression analysis for the measured and estimated global irradiances. Although the coefficient of determination is quite high ($R^2 = 0.993$), there is a set of points which are outside the 95% confidence intervals, indicating that one of the components is probably wrong. A misalignment error seems to affect the solar tracking system during the period 25 March 2021–8 April 2021. Figure 10b shows similar results, using the ratio of $G$ to the sum of $B + D$. The reference lines show the limits within which the values of the ratio should be detected. Generally, there is a higher dispersion of points at low elevation angles (or high SZA), which can be attributed to the cosine and directional response error of the pyranometers. According to Kipp and Zonen, the CM21 has a directional response < 10 W/m$^2$ up to 80°

zenith angle. In very low irradiance conditions, the relative uncertainties will go up due to the sensitivity of the sensor.

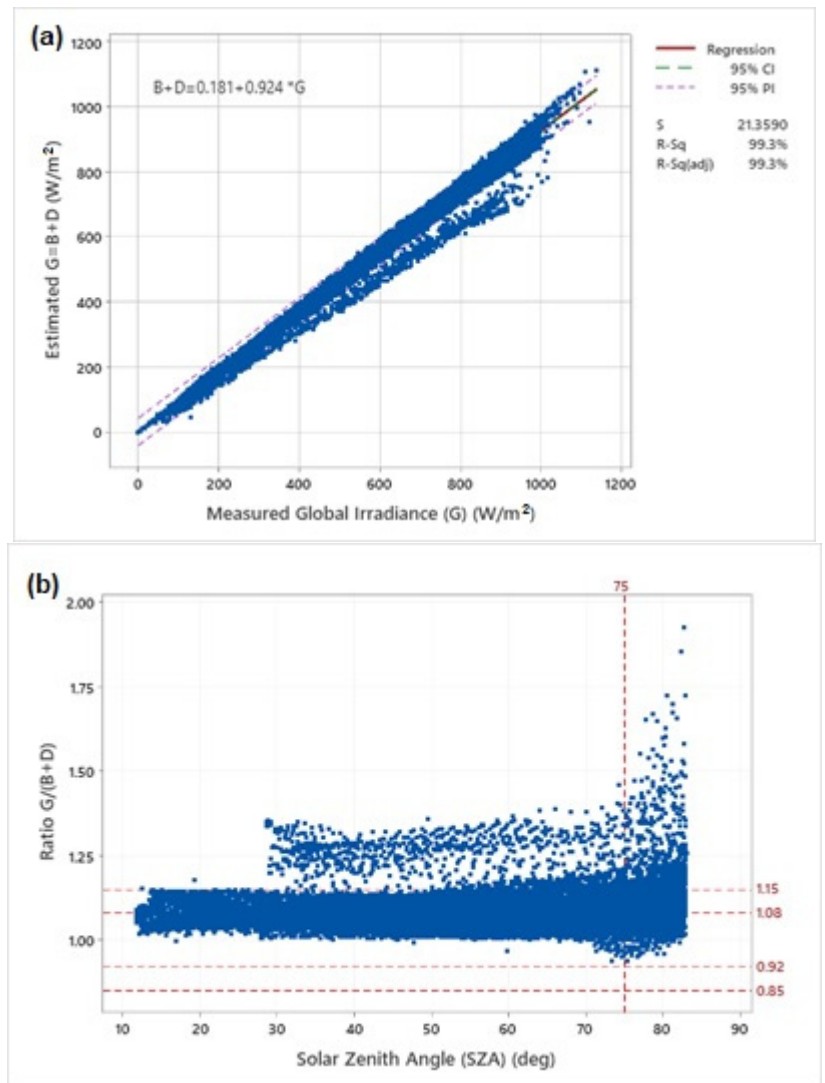

**Figure 10.** (**a**) linear regression of the estimated sum of the measured irradiances ($B + D$) and the relevant global irradiances ($G$); and (**b**) Ratio of $G/(B + D)$ as a function of solar zenith angle (SZA) with the reference lines within the ratio should be detected.

### 2.5.4. Quality Control of Daily Values

The quality control process was also extended to the daily sums of global solar radiation and daily sunshine duration values. Firstly, the daily values of global irradiation ($G_d$) were checked against the extraterrestrial daily irradiation ($G_{0d}$) (Table 4). The daily values of $G_d$ should be lower than $G_{0d}$, but they should be higher than the lowest limit of $0.03 \times G_d$ during overcast conditions [43]. As it can be seen in Figure 11, $G_d$ is lower than the upper limits of $G_{0d}$ and $0.8 \times G_{0d}$. The graph also shows that the daily direct horizontal irradiation ($B_d$) is lower than the ERL threshold of $0.8 \times G_{0d}$. Similarly, the diffuse irradiation ($D_d$) is lower than the threshold of $G_{0d}*0.7$. Generally, all daily radiation values are within their upper limits defined by PPL and ERL criteria. The second test is a comparison test between the extreme values of daily sums of global irradiation and the respective estimated daily sums on clear days. $G_d$ should be lower than the product of $1.1 \times G_{cd}$. As can be seen in Figure 11, $G_d$ is slightly higher than the product of $1.1 \times G_{cd}$ at least in some cases during the first half of the year.

**Table 4.** Daily Upper limits of Physically Possible tests (PPL) and Extremely Rare tests (ERL) (Lower limit of $G_d$ is $0.03 \times G_{0d}$) [5].

| Parameter | PPL ERL | Upper Limit (MJ/m$^2$) |
|---|---|---|
| Global irradiation ($G_d$) vs. $G_{0d}$ | PPL | $G_d < G_{0d}$ |
| Global irradiation ($G_d$) vs. $G_{0d}$ | ERL | $G_d < 0.8 \times G_{0d}$ |
| Global irradiation ($G_d$) vs. $G_{cd}$ | ERL | $G_d < 1.1 \times G_{cd}$ |
| Diffuse irradiation ($D_d$) vs. $G_{0d}$ | PPL | $D_d < 1.0 \times G_{0d}$ |
| Diffuse irradiation ($D_d$) vs. $G_{0d}$ | ERL | $D_d < 0.7 \times G_{0d}$ |
| Beam Normal irrad. ($B_{nd}$) vs. $G_{0nd}$ | PPL | $B_{nd} < 1.0 \times G_{0nd}$ |
| Beam Normal irrad. ($B_{nd}$) vs. $G_{0nd}$ | ERL | $B_{nd} < 0.8 \times G_{0nd}$ |
| Beam Horizontal irrad. ($B_d$) vs. $G_{0d}$ | PPL | $B_d < 1.0 \times G_{0d}$ |
| Beam Horizontal irrad. ($B_d$) vs. $G_{0d}$ | ERL | $B_d < 0.8 \times G_{0d}$ |
| Beam Horizontal irrad. ($B_d$) vs. $B_{cd}$ | PPL | $B_d < 1.1 \times B_{cd}$ |
| Consistency test [Sum $B_d + D_d)/G_d$] | PPL | Range of 0.9 to 1.1 |
| Daily Sunshine Duration ($S_d$) vs. $S_{0d}$ | PRL | $S_d < S_{0d}$ |

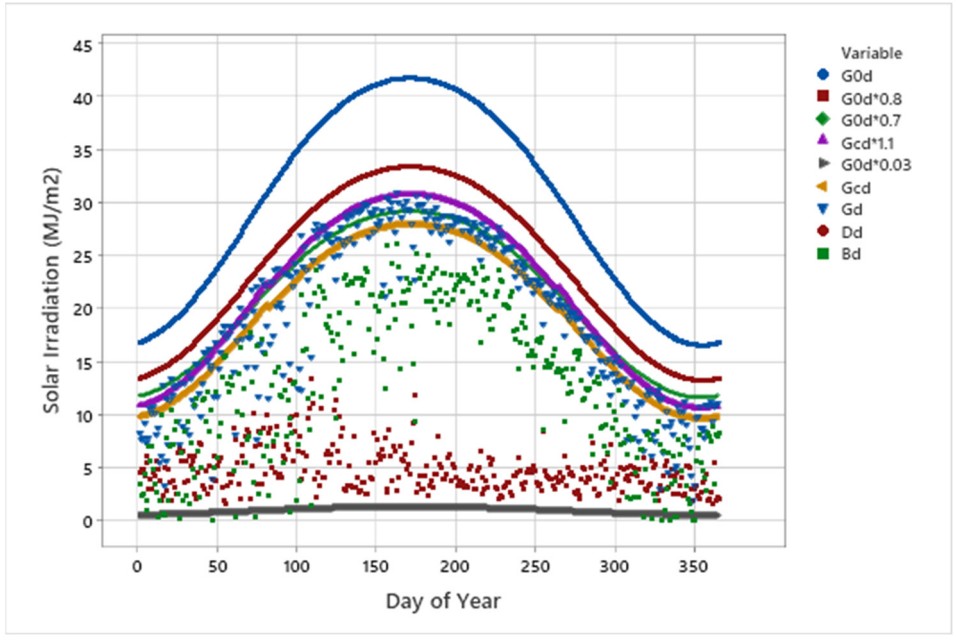

**Figure 11.** Daily totals of global irradiation ($G_d$) compared to the respective extraterrestrial ($G_{0d}$) and clear-sky irradiation ($G_{cd}$). The daily totals of horizontal direct ($B_d$) and diffuse irradiation ($D_d$) are also shown in the graph.

The sums of the daily values of direct horizontal and diffuse irradiation ($B_d + D_d$) are closely compared with the daily global irradiation ($G_d$) as shown in Figure 12.

Figure 13 shows the linear relationship between the daily clearness index ($K_T$) and the daily relative sunshine ($\sigma$). Most of the values are within the 95% prediction intervals. According to Scharmer and Greif [43], if the deviations of the pairs $K_T$ and $\sigma$ from the regression line are greater than 0.3, then the pairs are considered as suspect. Generally, the deviations are lower than 0.3. Additionally, the daily sums of sunshine duration ($S_d$) are lower than the astronomical daylength ($S_{0d}$).

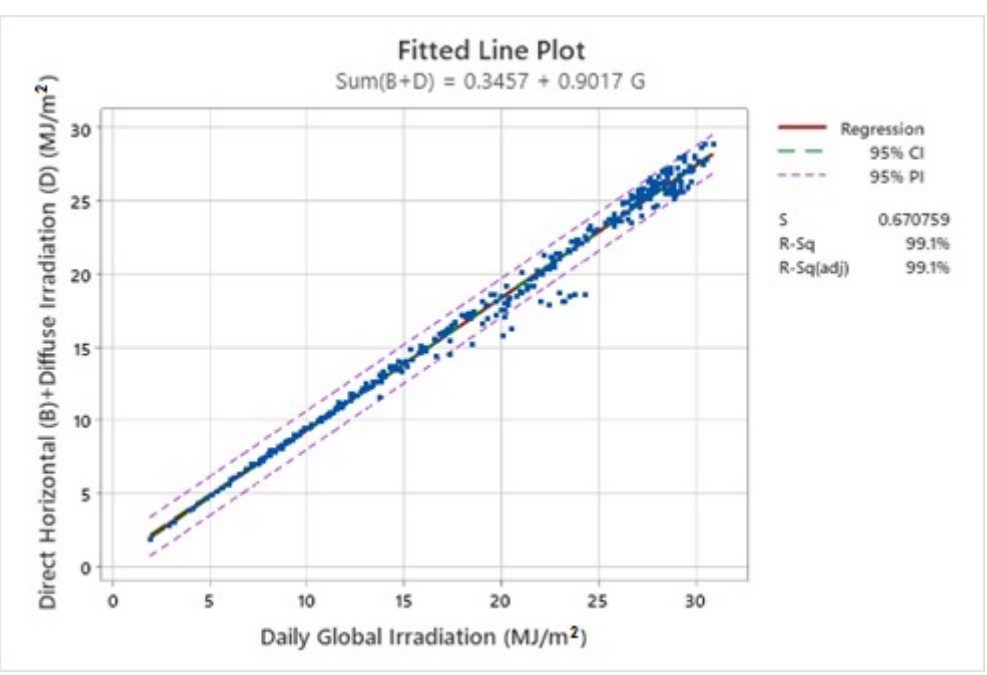

**Figure 12.** Linear regression between the sum ($B_d + D_d$) and global irradiation ($G_d$).

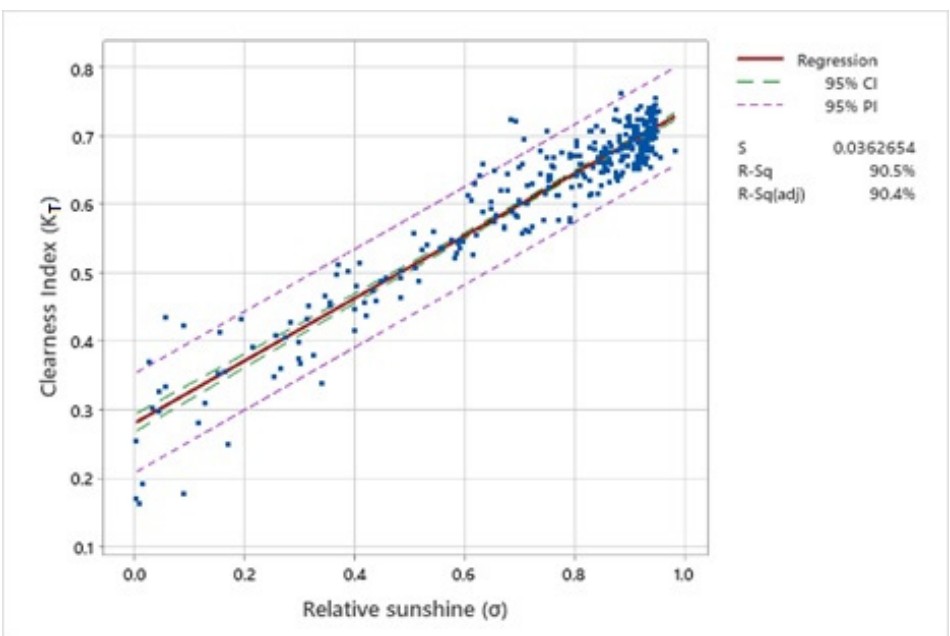

**Figure 13.** Linear relationship between $K_T$ and $\sigma$. $K_T = 0.281 + 0.455 * \sigma$, $R^2 = 0.904$.

The time series plots of the daily values of all the shortwave radiation components are illustrated in Figure 14. High variability is observed in spring.

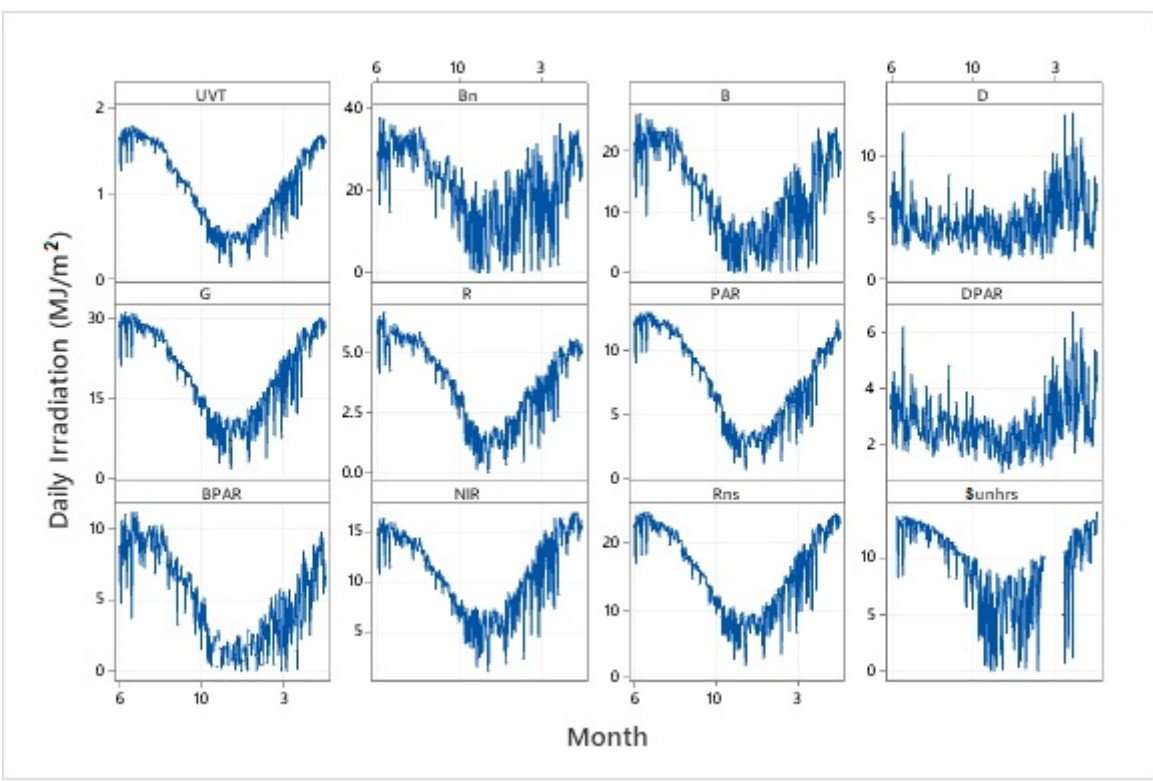

**Figure 14.** Time series plots of daily values of all shortwave radiation components including daily sunshine duration.

## 3. Statistical Analysis

### 3.1. Hourly Values

3.1.1. Solar Radiation Components vs. Solar Zenith Angle and Clearness Index

The relation of extreme values of global radiation as a function of solar zenith angle and the clearness index is shown in Figure 15. The higher values are recorded at low solar zenith angles (SZA) during clear sky conditions. However, extreme values are recorded when scatter clouds reflect additional solar radiation on the ground surface. Therefore, on such occasions the global irradiance reaches the value of about 1400 W/m$^2$ (Figure 15). Similar values are reported by Tapakis and Charalambides (2014) [40]. The lower values of all the radiation components are observed at high SZA (low solar elevation) during cloudy conditions.

Global irradiance is decreased almost exponentially with increasing SZA for a given $k_t$ interval. This relationship can be expressed as:

$$G = G_{\max} * (\cos \theta_z)^b \tag{27}$$

where $G_{\max}$ is the maximum global irradiance for each $k_t$ interval and b describes how $G$ changes with the cosine of SZA ($\theta_z$). The dependence of $G_{\max}$ on $k_t$ can be described by cubic equation as:

$$G_{\max} = -50.77 + 2033 * k_t - 4931 * k_t^2 + 4721 * k_t^3 \tag{28}$$

Almost similar graphs to global irradiances are obtained for direct normal ($B_n$) and direct horizontal ($B$) irradiances. The diffuse irradiance shows high values (about 500 and 600 W/m$^2$) at low solar zenith angles but in most cases it remains relatively constant between 50 and 200 W/m$^2$.

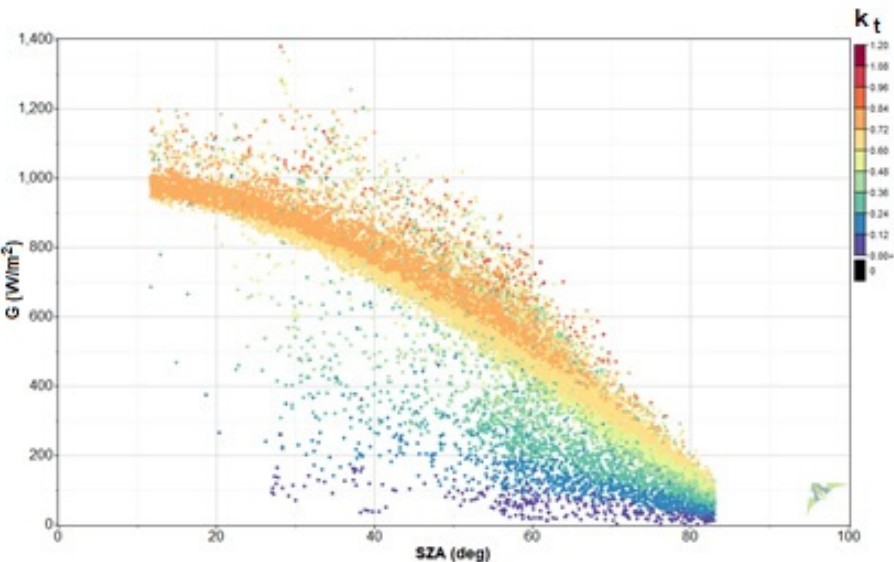

**Figure 15.** Solar global irradiances as a function of solar zenith angle (SZA) and clearness index ($k_t$).

### 3.1.2. Daily Variation of Shortwave Irradiances

In Figure 16, monthly mean hourly (MMH) values of global, direct, and diffuse irradiances on horizontal surfaces in W/m$^2$ are shown by means of isoline diagrams. These graphs show representative values of irradiances for each hour of every month of the year. The highest $B_n$ irradiance is recorded in the summer at noon with values exceeding 800 W/m$^2$ (Figure 16a), while the respective $B$ exceeds the value of 750 W/m$^2$ (Figure 16b). $G$ fluctuates between 450 W/m$^2$ in January and 950 W/m$^2$ in July at local noon (Figure 16d), while $D$ fluctuates between 100 W/m$^2$ and 240 W/m$^2$. In summer, diffuse irradiances range mainly between 100 and 150 W/m$^2$ (Figure 16c).

### 3.1.3. Monthly Variability of Irradiances of Solar Radiation Components

The monthly variability of the irradiances of the shortwave radiation components is demonstrated with a graph of boxplots for each month of the year. The boxplot presents the median and the interquartile range (IQR) as well as the outliers (asterisks) of each variable (Figure 17). The smooth line represents the mean values of irradiances for each month of the year. As it can be seen no outliers are observed for $B$ and $G$ irradiances, while they are observed for the diffuse component in all months of the year. Generally, medians are closed to the mean values. The variability of the global irradiance in the summer months is greater than in other seasons as it is indicated from the length of the boxplots. Similar pattern is shown for the direct horizontal irradiance. On the other hand, the variability of direct normal and diffuse irradiances is higher during the winter and spring. Monthly descriptive statistics for the shortwave solar radiation components are presented in Table 5. Monthly mean values of global irradiances range between 300 and 635 W/m$^2$, while those for direct horizontal irradiances range between 130 and 435 W/m$^2$. The monthly mean values of diffuse irradiance are relatively constant throughout the year ranging mainly between 90 and 170 W/m$^2$.

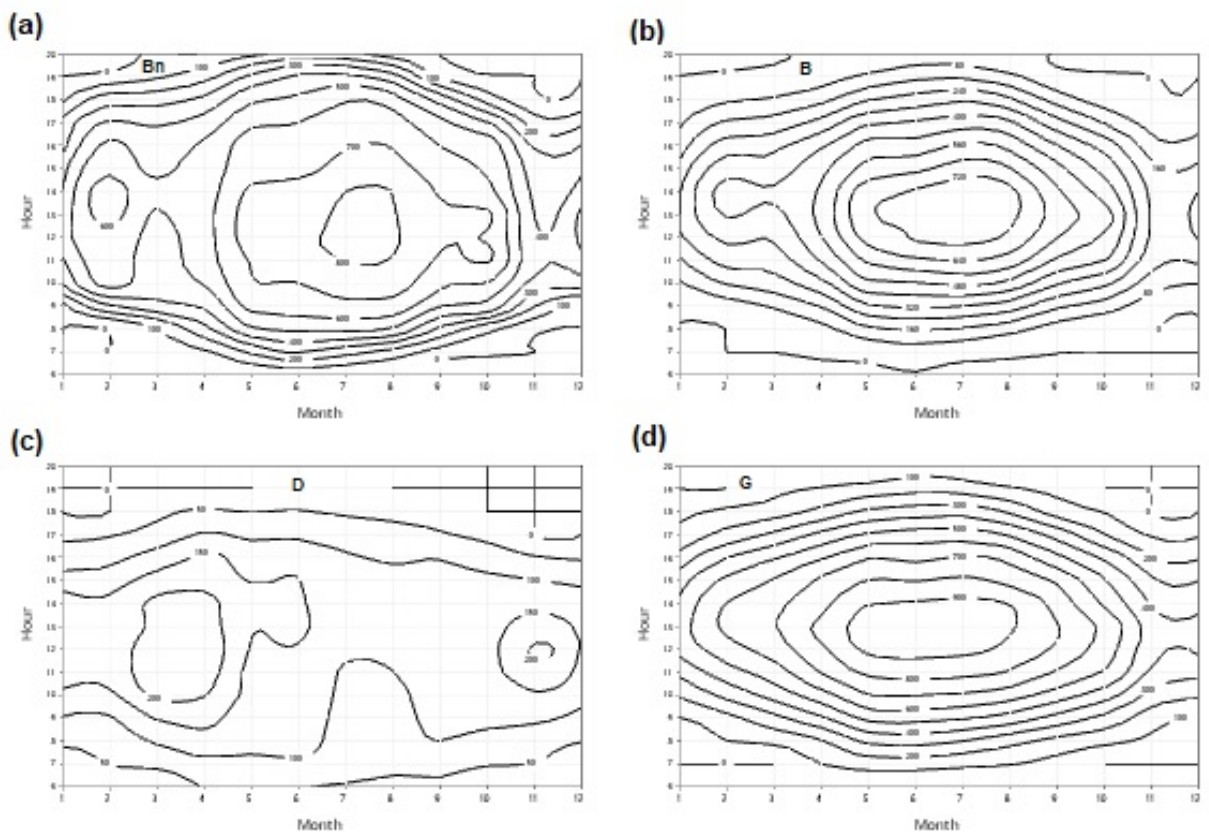

**Figure 16.** Isoline diagrams of monthly mean hourly: (**a**) direct normal; (**b**) direct horizontal; (**c**) diffuse; and (**d**) global irradiance values (W/m$^2$).

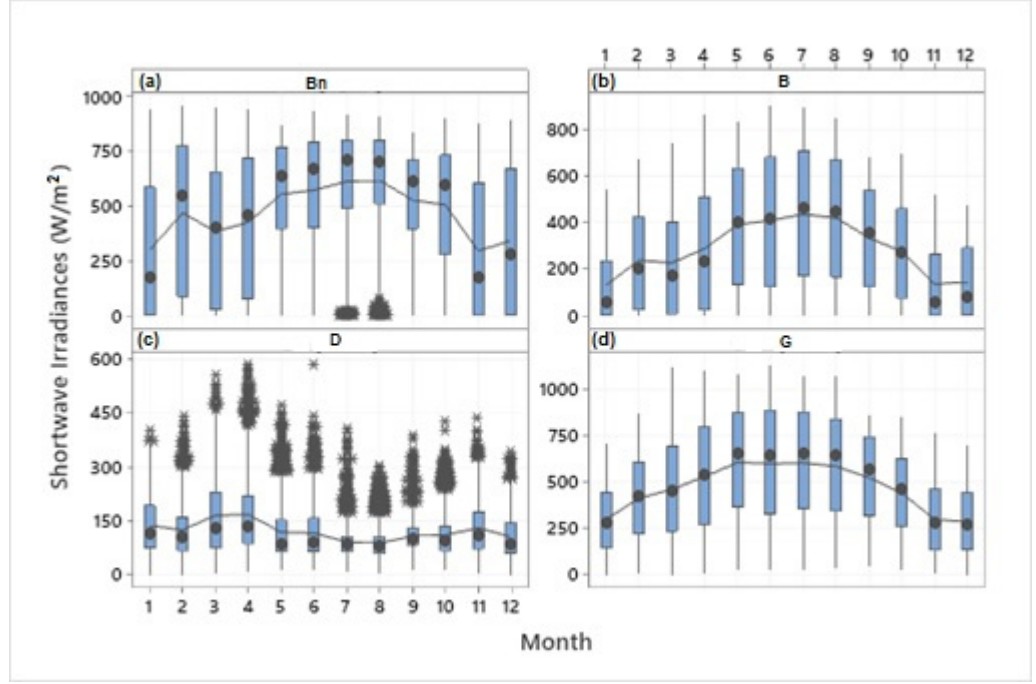

**Figure 17.** Boxplots of irradiances of the shortwave radiation components: (**a**) direct normal ($B_n$); (**b**) direct horizontal ($B$); (**c**) diffuse ($D$); and (**d**) global ($G$). The symbol of median values is represented as circle in the boxplot. The asterisks denote the outliers. The smooth curve represents the mean values of each month of the year.

**Table 5.** Monthly descriptive statistics of the shortwave solar radiation components ($W/m^2$).

| | $B_n$ [$W/m^2$] | | | | | | | $B$ [$W/m^2$] | | | | | | |
|---|---|---|---|---|---|---|---|---|---|---|---|---|---|---|
| Month | N | Mean | StDev | Min | Q1 | Median | Q3 | Max | N | Mean | StDev | Min | Q1 | Median | Q3 | Max |
| 1 | 1841 | 298 | 312 | 0 | 2 | 175 | 584 | 940 | 1841 | 130 | 156 | 0 | 1 | 53 | 235 | 546 |
| 2 | 1796 | 467 | 331 | 0 | 87 | 547 | 771 | 960 | 1796 | 236 | 208 | 0 | 22 | 201 | 421 | 671 |
| 3 | 2187 | 382 | 315 | 0 | 26 | 399 | 655 | 946 | 2187 | 225 | 220 | 0 | 6 | 167 | 399 | 744 |
| 4 | 2318 | 424 | 315 | 0 | 76 | 461 | 716 | 945 | 2318 | 287 | 265 | 0 | 22 | 234 | 507 | 866 |
| 5 | 2576 | 552 | 262 | 0 | 396 | 638 | 765 | 869 | 2576 | 389 | 267 | 0 | 133 | 399 | 632 | 832 |
| 6 | 2548 | 571 | 280 | 0 | 399 | 671 | 790 | 930 | 2548 | 407 | 289 | 0 | 122 | 412 | 678 | 905 |
| 7 | 2627 | 611 | 250 | 0 | 487 | 708 | 799 | 917 | 2627 | 434 | 280 | 0 | 168 | 461 | 704 | 890 |
| 8 | 2473 | 615 | 247 | 0 | 509 | 701 | 798 | 907 | 2473 | 418 | 266 | 0 | 166 | 447 | 665 | 845 |
| 9 | 2197 | 526 | 232 | 0 | 395 | 614 | 707 | 834 | 2197 | 332 | 218 | 0 | 122 | 353 | 535 | 679 |
| 10 | 2052 | 503 | 283 | 0 | 281 | 599 | 730 | 899 | 2052 | 274 | 204 | 0 | 75 | 268 | 457 | 697 |
| 11 | 1821 | 293 | 300 | 0 | 0 | 175 | 603 | 874 | 1821 | 134 | 156 | 0 | 0 | 53 | 263 | 519 |
| 12 | 1800 | 343 | 323 | 0 | 2 | 276 | 667 | 896 | 1800 | 143 | 153 | 0 | 0 | 76 | 289 | 473 |
| Year | 26,236 | 478 | 307 | 0 | 169 | 574 | 743 | 960 | 26,236 | 298 | 258 | 0 | 40 | 261 | 507 | 905 |

| | $D$ [$W/m^2$] | | | | | | | $G$ [$W/m^2$] | | | | | | |
|---|---|---|---|---|---|---|---|---|---|---|---|---|---|---|
| Month | N | Mean | StDev | Min | Q1 | Median | Q3 | Max | N | Mean | StDev | Min | Q1 | Median | Q3 | Max |
| 1 | 1600 | 138 | 80 | 0 | 75 | 114 | 193 | 403 | 1600 | 311 | 177 | 7 | 156 | 287 | 461 | 737 |
| 2 | 1594 | 125 | 77 | 0 | 68 | 104 | 160 | 444 | 1594 | 428 | 223 | 14 | 228 | 439 | 630 | 907 |
| 3 | 1969 | 166 | 111 | 1 | 77 | 130 | 231 | 556 | 1970 | 476 | 262 | 7 | 245 | 464 | 717 | 1119 |
| 4 | 2112 | 168 | 111 | 6 | 88 | 134 | 220 | 586 | 2112 | 549 | 299 | 12 | 277 | 553 | 831 | 1146 |
| 5 | 2346 | 119 | 80 | 12 | 64 | 88 | 153 | 474 | 2346 | 632 | 299 | 34 | 372 | 686 | 915 | 1145 |
| 6 | 2335 | 117 | 73 | 13 | 66 | 90 | 158 | 585 | 1080 | 635 | 298 | 46 | 379 | 679 | 918 | 1142 |
| 7 | 2393 | 92 | 47 | 10 | 64 | 85 | 108 | 408 | 2393 | 628 | 293 | 45 | 373 | 681 | 907 | 1107 |
| 8 | 2254 | 89 | 45 | 5 | 59 | 80 | 105 | 304 | 2254 | 613 | 284 | 54 | 366 | 666 | 877 | 1126 |
| 9 | 1990 | 108 | 46 | 14 | 83 | 103 | 129 | 389 | 1990 | 553 | 250 | 54 | 340 | 600 | 786 | 907 |
| 10 | 1832 | 111 | 61 | 14 | 68 | 96 | 136 | 428 | 1832 | 467 | 215 | 37 | 286 | 489 | 657 | 891 |
| 11 | 1593 | 131 | 77 | 3 | 73 | 111 | 175 | 438 | 1593 | 312 | 186 | 12 | 146 | 290 | 478 | 785 |
| 12 | 1550 | 106 | 64 | 0 | 61 | 85 | 144 | 341 | 1550 | 296 | 170 | 7 | 141 | 279 | 458 | 719 |
| Year | 23,568 | 122 | 79 | 0 | 68 | 98 | 150 | 586 | 22,314 | 504 | 283 | 7 | 254 | 494 | 743 | 1146 |

### 3.1.4. Frequencies and Cumulative Density Curves

The annual cumulative curves (CDF) for each variable are shown in Figure 18. The graphs of CDF curves are useful in determining the percent for different thresholds of irradiances. For example, the 50 percentile represents a value of 480 $W/m^2$ for global irradiances, 575 $W/m^2$ for direct normal, 275 $W/m^2$ for direct horizontal and 100 $W/m^2$ for the diffuse irradiances. The graph indicates that in 50% of the irradiances are less or higher than the above values.

### 3.1.5. Estimated Net Shortwave Radiation ($R_{ns}$)

The short-wave input to a horizontal surface ($G$) consists of both direct-beam ($B$) and diffuse ($D$) radiation, so that $G = B + D$. On a cloudless day, $D$ is about 15–25% of $G$ and the rest represent direct-beam radiation. The pattern of $G$ is of course controlled by the azimuth and altitude of the Sun relative to the horizon, with a single peak at local solar noon. The azimuth and altitude are controlled by Earth–Sun geometrical relationships and vary with latitude, time of day, and date. The peak of $G$ for a clear summer day is approximately 1000 $W/m^2$ and the peak of $G$ for a winter clear day is about 550 $W/m^2$. On a cloudy day $G$ is equal to $D$. Then, the net short-wave radiation ($R_{ns}$) is given by:

$$R_{ns} = G - R \tag{29}$$

The reflected short-wave radiation ($R$) depends on the value of $G$ and the surface albedo ($\rho$), so that:

$$R = \rho * G \tag{30}$$

Although $\rho$ for most surfaces is not perfectly constant through the day, as a first approximation is a reduced mirror-image of $G$. At this site the annual average of $\rho$ is 0.17, so that approximately one sixth of $G$ is lost through the ground reflection.

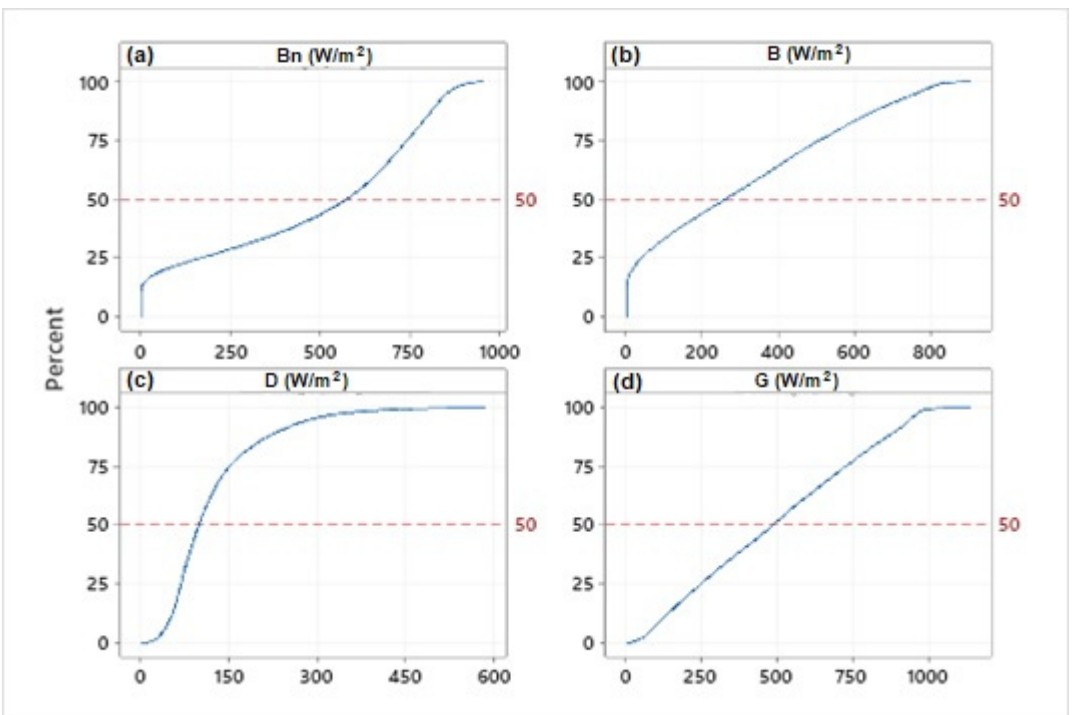

**Figure 18.** Cumulative frequency distribution (CDF) of shortwave solar radiation components: (**a**) direct normal ($B_n$); (**b**) direct horizontal ($B$); (**c**) diffuse ($D$); and (**d**) global ($G$) irradiances (W/m$^2$).

Therefore, $R_{ns}$ on a clear day is approximately of about $0.83 \times G$ in magnitude. The diurnal variation of $R_{ns}$ for each month of the year is shown in Figure 19a. The maximum peak of $R_{ns}$ in the summer months is about 750 W/m$^2$. The monthly variability of $R_{ns}$ irradiances is shown in Figure 19b. High variability is observed in the summer months.

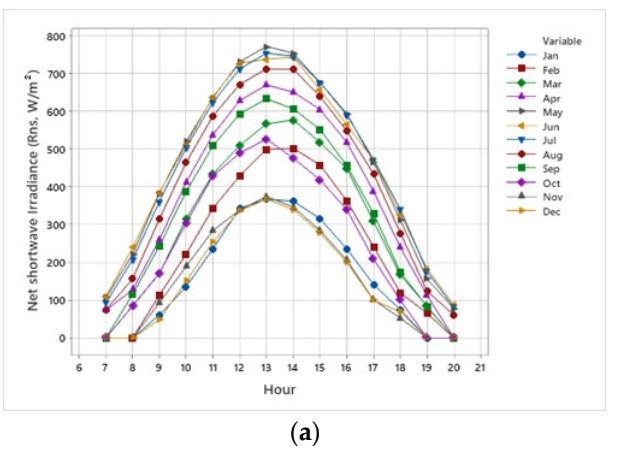

(**a**)

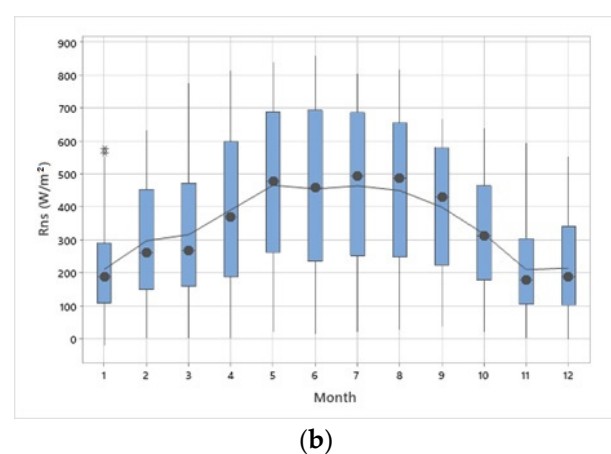

(**b**)

**Figure 19.** (**a**) diurnal variation of the average hourly values of $R_{ns}$ for each month of the year; and (**b**) monthly variability of net shortwave irradiances (W/m$^2$). Symbol * indicates an outlier.

### 3.1.6. Sky Conditions Classification

As it was indicated earlier, weather conditions can be classified into four categories according to the values of clearness index, i.e., (i) cloudy sky ($k_t \leq 0.35$); (ii) partially cloudy with predominance of diffuse component ($0.35 < k_t \leq 0.55$); (iii) partially cloudy with predominance of direct component of the solar radiation ($0.55 < k_t \leq 0.65$); and (iv) clear sky ($k_t > 0.65$).

Table 6 presents the statistical values of all radiation components considering the prevailing weather conditions as classified by the clearness index. In general, the hourly fractions of *UVT* component to global solar irradiance, namely *UVT/G*, increase as sky conditions changed from clear to overcast; by contrast, the fraction (*NIR/G*) decreases with sky conditions changing from clear to overcast, while the fraction of *GPAR/G* remains relatively constant. The maximum hourly values of *G*, *UVT*, GPAR and NIR can be interpreted as the amplitude of the diurnal evolution, and they increase as the cloudy cover decreases.

The monthly statistics of the shortwave radiation components for the four sky conditions are presented in Table 7. The monthly means of the global irradiances under clear sky conditions range between 460 and 760 W/m$^2$, while under cloudy conditions they range between 100 and 180 W/m$^2$. High values are recorded for direct irradiance under clear sky conditions, in contrast to the very low direct irradiance under cloudy conditions.

The global irradiance under different sky conditions can be estimated from the equation:

$$G = a * m^b \tag{31}$$

where *m* is the optical air mass which is estimated by the Kasten and Young (1989) [44] formula:

$$m = (p/p_0) / [\cos\theta_z + 0.50572 * (96.07995 - \theta_z)^{-1.6364}] \tag{32}$$

$$(p/p_0) = \exp(-z/8435.2) \tag{33}$$

where $\theta_z$ is the solar zenith angle in degrees and *z* is the elevation of the station in m. Figure 20 shows the result of Equation (31) for the four sky conditions. The estimated parameters *a* and *b* are shown on the graph.

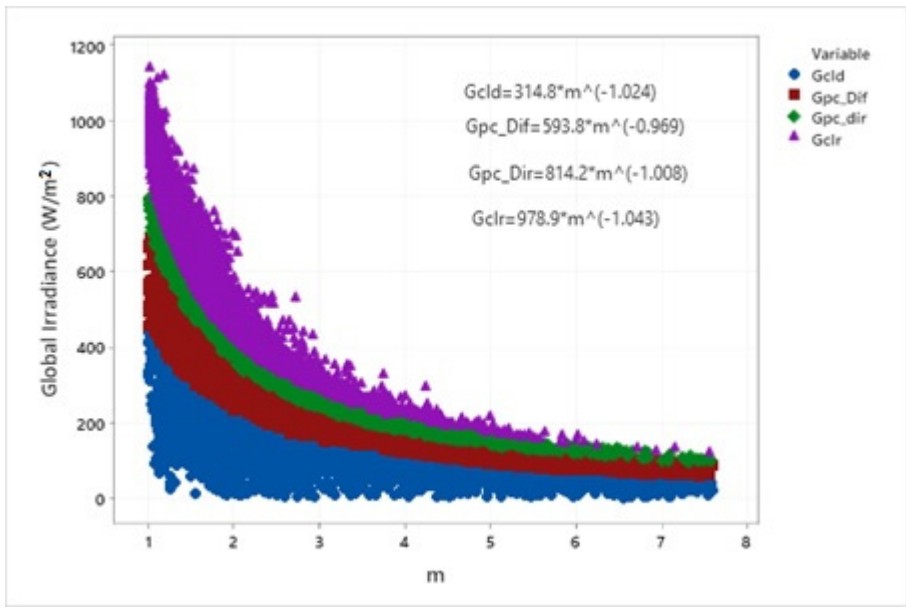

**Figure 20.** Relationship of global irradiance in W/m$^2$ and optical air mass (m).

**Table 6.** Statistical values of all radiation components considering the prevailing weather conditions as classified by the clearness index ($k_t$).

| | | $k_t \leq 0.35$ (Cloudy) | | | | | | $k_t > 0.65$ (Clear Sky) | | | | | |
|---|---|---|---|---|---|---|---|---|---|---|---|---|---|
| Var. | Unit | Fraction of *G* (%) | Mean | Median | Min | Max | Std. Dev. | Fraction of *G* (%) | Mean | Median | Min | Max | Std. Dev. |
| *G* | W/m$^2$ | | 135.9 | 118 | 7 | 446.0 | 82.69 | | 668.7 | 675 | 110 | 1146.0 | 210.27 |
| *UVB* | W/m$^2$ | 0.12 | 0.2 | 0 | 0 | 1.3 | 0.18 | 0.14 | 0.9 | 1 | 0 | 2.2 | 0.53 |
| *UVA* | W/m$^2$ | 6.63 | 9.0 | 8 | 0 | 35.0 | 5.70 | 5.20 | 34.8 | 34 | 3 | 61.5 | 13.08 |
| *UVE* | W/m$^2$ | 0.01 | 0.0 | 0 | 0 | 0.1 | 0.02 | 0.01 | 0.1 | 0 | 0 | 0.2 | 0.06 |
| *UVT* | W/m$^2$ | 6.75 | 9.2 | 8 | 0 | 35.9 | 5.90 | 5.34 | 35.7 | 35 | 3 | 63.6 | 13.60 |
| *Bn* | W/m$^2$ | 9.82 | 13.4 | 0 | 0 | 584.0 | 38.49 | 105.36 | 704.6 | 722 | 1 | 960.0 | 127.44 |
| *B* | W/m$^2$ | 4.08 | 5.6 | 0 | 0 | 201.0 | 16.15 | 71.20 | 476.2 | 467 | 0 | 890.2 | 186.26 |
| *D* | W/m$^2$ | 84.91 | 115.4 | 102 | 1 | 406.0 | 74.39 | 17.59 | 117.6 | 99 | 26 | 578.0 | 65.87 |
| *RF* | W/m$^2$ | 9.25 | 12.6 | 8 | 0 | 69.0 | 13.73 | 18.92 | 126.5 | 129 | 1 | 238.0 | 44.05 |
| *GPPFD* | µmol/s/m$^2$ | | 232.7 | 200 | 24 | 897.0 | 140.67 | | 1150.5 | 1154 | 118 | 2113.0 | 413.32 |
| *GPAR* | W/m$^2$ | 37.46 | 50.9 | 44 | 5 | 196.3 | 30.78 | 37.65 | 251.8 | 253 | 26 | 462.4 | 90.44 |
| *DPPFD* | µmol/s/m$^2$ | | 269.7 | 238 | 2 | 958.0 | 164.95 | | 324.9 | 291 | 89 | 1297.0 | 141.89 |
| *DPAR* | W/m$^2$ | 43.42 | 59.0 | 52 | 0 | 209.6 | 36.10 | 10.63 | 71.1 | 64 | 19 | 283.8 | 31.05 |
| *NIR* | W/m$^2$ | 55.79 | 75.8 | 66 | 0 | 248.1 | 48.39 | 57.02 | 381.3 | 384 | 66 | 750.8 | 114.22 |
| *Lupd* | W/m$^2$ | 293.58 | 399.1 | 390 | 315 | 626.0 | 42.12 | 78.78 | 526.8 | 534 | 318 | 704.0 | 88.78 |
| *Ldnd* | W/m$^2$ | 253.77 | 345.0 | 348 | 244 | 452.0 | 29.10 | 52.34 | 350.1 | 355 | 235 | 461.0 | 40.56 |

| | | $0.35 < k_t \leq 0.55$ (Partially Cloudy ) | | | | | | $0.55 < k_t \leq 0.65$ (Partially Cloudy ) | | | | | |
|---|---|---|---|---|---|---|---|---|---|---|---|---|---|
| Var. | Unit | Fraction of *G* (%) | Mean | Median | Min | Max | Std. Dev. | Fraction of *G* (%) | Mean | Median | Min | Max | Std. Dev. |
| *G* | W/m$^2$ | | 235.1 | 199 | 58 | 700.0 | 133.14 | | 304.6 | 276 | 94 | 817.0 | 139.62 |
| *UVB* | W/m$^2$ | 0.10 | 0.2 | 0 | 0 | 1.6 | 0.25 | 0.08 | 0.3 | 0 | 0 | 1.8 | 0.25 |
| *UVA* | W/m$^2$ | 5.43 | 12.8 | 10 | 2 | 43.1 | 8.25 | 4.79 | 14.6 | 13 | 4 | 50.8 | 8.15 |
| *UVE* | W/m$^2$ | 0.01 | 0.0 | 0 | 0 | 0.2 | 0.03 | 0.01 | 0.0 | 0 | 0 | 0.2 | 0.03 |
| *UVT* | W/m$^2$ | 5.53 | 13.0 | 10 | 2 | 44.7 | 8.49 | 4.88 | 14.9 | 13 | 4 | 52.6 | 8.40 |
| *B$_n$* | W/m$^2$ | 77.75 | 182.8 | 164 | 0 | 691.0 | 142.27 | 140.24 | 427.1 | 446 | 0 | 782.0 | 131.46 |
| *B* | W/m$^2$ | 25.06 | 58.9 | 45 | 0 | 384.9 | 54.30 | 50.73 | 154.5 | 146 | 0 | 568.5 | 76.55 |
| *D* | W/m$^2$ | 64.21 | 151.0 | 118 | 5 | 586.0 | 114.64 | 37.34 | 113.7 | 81 | 16 | 556.0 | 90.69 |
| *RF* | W/m$^2$ | 14.10 | 33.1 | 28 | 0 | 127.0 | 24.96 | 16.42 | 50.0 | 47 | 0 | 162.0 | 29.50 |
| *GPPFD* | µmol/s/m$^2$ | | 395.7 | 347 | 86 | 1335.0 | 214.53 | | 513.1 | 471 | 78 | 1574.0 | 242.81 |
| *GPAR* | W/m$^2$ | 36.84 | 86.6 | 76 | 19 | 292.1 | 46.94 | 36.86 | 112.3 | 103 | 17 | 344.4 | 53.13 |
| *DPPFD* | µmol/s/m$^2$ | | 356.0 | 279 | 56 | 1298.0 | 241.85 | | 294.3 | 231 | 76 | 1227.0 | 190.19 |

**Table 6.** *Cont.*

| | | | | | | | | | | | | | |
|---|---|---|---|---|---|---|---|---|---|---|---|---|---|
| *DPAR* | W/m$^2$ | 33.13 | 77.9 | 61 | 12 | 284.0 | 52.92 | 21.14 | 64.4 | 51 | 17 | 268.5 | 41.62 |
| *NIR* | W/m$^2$ | 57.64 | 135.5 | 114 | 23 | 404.0 | 80.89 | 58.26 | 177.4 | 158 | 52 | 483.9 | 82.41 |
| *Lupd* | W/m$^2$ | 184.95 | 434.8 | 424 | 319 | 659.0 | 53.41 | 146.94 | 447.5 | 432 | 314 | 665.0 | 63.17 |
| *Ldnd* | W/m$^2$ | 144.47 | 339.6 | 342 | 235 | 453.0 | 32.61 | 110.01 | 335.1 | 335 | 235 | 438.0 | 37.00 |

**Table 7.** Monthly statistics of direct normal ($B_n$), direct horizontal ($B$), diffuse ($D$) and global irradiance ($G$) in (W/m$^2$) for cloudy, partially cloudy and clear sky conditions.

| Cloudy ($k_t \leq 0.35$) | | | | | | | | | | | | | | | |
|---|---|---|---|---|---|---|---|---|---|---|---|---|---|---|---|
| $B_n$ | | | | $B$ | | | | $D$ | | | | $G$ | | | |
| Month | Mean | StDev | Median | Max | Mean | StDev | Median | Max | Mean | StDev | Median | Max | Mean | StDev | Median | Max |
| 1 | 10.9 | 28.0 | 0.0 | 213 | 3.7 | 10.6 | 0.0 | 104 | 112.0 | 61.5 | 104.0 | 274 | 118.8 | 64.2 | 112.0 | 277 |
| 2 | 10.7 | 26.6 | 0.0 | 170 | 4.3 | 12.7 | 0.0 | 115 | 126.7 | 70.7 | 116.0 | 325 | 134.3 | 74.5 | 129.0 | 328 |
| 3 | 9.0 | 22.6 | 0.0 | 141 | 4.5 | 11.9 | 0.0 | 90 | 139.6 | 90.5 | 119.0 | 368 | 148.7 | 95.7 | 126.0 | 384 |
| 4 | 9.2 | 24.8 | 0.0 | 161 | 3.9 | 12.6 | 0.0 | 122 | 142.8 | 99.5 | 116.0 | 403 | 151.6 | 104.8 | 120.0 | 427 |
| 5 | 36.7 | 46.1 | 15.5 | 206 | 18.2 | 25.5 | 6.5 | 90 | 126.7 | 89.7 | 97.0 | 365 | 156.9 | 111.3 | 124.5 | 428 |
| 6 | 24.4 | 43.2 | 1.0 | 218 | 13.6 | 28.2 | 0.7 | 166 | 159.2 | 89.7 | 163.0 | 406 | 180.4 | 107.4 | 176.0 | 444 |
| 7 | 114.2 | 103.2 | 111.0 | 317 | 29.8 | 39.4 | 23.9 | 201 | 85.0 | 82.1 | 40.5 | 319 | 122.8 | 104.7 | 67.5 | 396 |
| 8 | 133.8 | 103.4 | 134.0 | 339 | 36.2 | 32.7 | 31.1 | 155 | 89.1 | 77.8 | 51.5 | 297 | 134.6 | 96.5 | 82.0 | 367 |
| 9 | 74.1 | 73.3 | 51.5 | 251 | 20.2 | 24.1 | 15.5 | 118 | 97.7 | 86.1 | 55.5 | 301 | 128.2 | 99.9 | 71.5 | 366 |
| 10 | 44.7 | 58.5 | 24.0 | 248 | 17.7 | 28.9 | 6.6 | 177 | 108.5 | 69.9 | 94.0 | 326 | 134.3 | 85.7 | 110.0 | 342 |
| 11 | 13.7 | 33.3 | 0.0 | 198 | 4.8 | 12.4 | 0.0 | 91 | 107.1 | 68.2 | 90.0 | 276 | 115.5 | 72.0 | 96.0 | 290 |
| 12 | 14.1 | 31.5 | 0.0 | 211 | 4.6 | 11.3 | 0.0 | 109 | 95.7 | 58.0 | 89.0 | 233 | 103.8 | 62.7 | 99.0 | 260 |
| Partly cloudy with predominance of the diffuse component ($0.35 < k_t \leq 0.55$) | | | | | | | | | | | | | | | |
| $B_n$ | | | | $B$ | | | | $D$ | | | | $G$ | | | |
| Month | Mean | StDev | Median | Max | Mean | StDev | Median | Max | Mean | StDev | Median | Max | Mean | StDev | Median | Max |
| 1 | 145.1 | 131.0 | 100.0 | 496 | 51.3 | 47.0 | 37.9 | 247 | 184.1 | 87.3 | 189.5 | 371 | 245.2 | 92.3 | 254.0 | 436 |
| 2 | 189.1 | 126.7 | 186.0 | 522 | 67.6 | 56.4 | 56.6 | 333 | 171.6 | 103.0 | 154.5 | 429 | 252.8 | 120.2 | 248.5 | 495 |
| 3 | 118.7 | 102.5 | 91.5 | 472 | 58.5 | 59.0 | 40.2 | 326 | 244.7 | 124.4 | 239.0 | 494 | 321.6 | 145.7 | 338.0 | 614 |
| 4 | 122.8 | 109.6 | 100.0 | 482 | 50.9 | 52.2 | 34.7 | 378 | 238.3 | 147.6 | 219.0 | 586 | 308.3 | 160.3 | 297.0 | 666 |
| 5 | 227.4 | 116.4 | 222.0 | 465 | 73.8 | 63.4 | 54.0 | 337 | 126.6 | 111.5 | 80.0 | 447 | 223.0 | 158.9 | 152.0 | 692 |
| 6 | 285.9 | 124.9 | 300.0 | 522 | 91.1 | 69.3 | 74.0 | 464 | 110.3 | 105.2 | 59.0 | 441 | 218.2 | 156.9 | 155.0 | 670 |

**Table 7.** *Cont.*

| 7 | 376.5 | 104.1 | 385.0 | 598 | 98.0 | 51.4 | 90.2 | 462 | 57.4 | 51.2 | 44.5 | 408 | 164.6 | 87.3 | 147.5 | 654 |
| 8 | 390.7 | 105.5 | 399.0 | 604 | 103.2 | 55.9 | 92.9 | 388 | 54.5 | 41.9 | 43.5 | 281 | 166.2 | 88.1 | 147.0 | 676 |
| 9 | 308.6 | 102.4 | 317.0 | 514 | 86.3 | 55.2 | 75.1 | 404 | 72.9 | 52.8 | 59.0 | 379 | 178.3 | 95.0 | 157.0 | 612 |
| 10 | 279.1 | 122.2 | 285.0 | 564 | 87.9 | 56.5 | 77.1 | 316 | 112.2 | 84.3 | 77.0 | 404 | 215.8 | 120.4 | 173.0 | 530 |
| 11 | 177.0 | 142.0 | 152.0 | 539 | 62.3 | 57.5 | 47.7 | 286 | 172.3 | 94.1 | 164.0 | 374 | 246.1 | 98.8 | 260.0 | 472 |
| 12 | 193.8 | 147.7 | 162.0 | 540 | 58.4 | 47.3 | 50.6 | 270 | 140.4 | 84.0 | 130.0 | 315 | 212.0 | 91.2 | 200.0 | 394 |

**Partly cloudy with predominance of the direct component ($0.55 < k_t \leq 0.65$)**

| | $B_n$ | | | | $B$ | | | | $D$ | | | | $G$ | | | |
|---|---|---|---|---|---|---|---|---|---|---|---|---|---|---|---|---|
| Month | Mean | StDev | Median | Max | Mean | StDev | Median | Max | Mean | StDev | Median | Max | Mean | StDev | Median | Max |
| 1 | 441.8 | 130.5 | 462.5 | 687 | 168.0 | 70.6 | 162.8 | 333 | 144.0 | 79.8 | 130.5 | 383 | 330.6 | 112.5 | 344.5 | 525 |
| 2 | 436.3 | 129.6 | 454.5 | 688 | 147.5 | 77.5 | 129.0 | 426 | 123.5 | 88.0 | 90.5 | 444 | 293.2 | 133.8 | 256.0 | 589 |
| 3 | 336.8 | 126.5 | 337.0 | 656 | 148.6 | 79.9 | 130.1 | 404 | 206.3 | 138.8 | 180.0 | 515 | 392.3 | 185.8 | 389.5 | 745 |
| 4 | 349.7 | 127.1 | 342.0 | 639 | 150.8 | 89.4 | 127.3 | 532 | 180.8 | 133.5 | 146.0 | 556 | 373.2 | 198.1 | 312.0 | 800 |
| 5 | 447.0 | 99.4 | 457.5 | 654 | 175.7 | 84.6 | 160.8 | 587 | 116.0 | 94.6 | 77.5 | 420 | 328.2 | 163.8 | 286.5 | 787 |
| 6 | 502.3 | 93.7 | 519.0 | 691 | 198.6 | 89.6 | 178.5 | 599 | 96.3 | 73.4 | 67.0 | 379 | 322.1 | 152.0 | 277.0 | 796 |
| 7 | 552.2 | 74.7 | 565.0 | 686 | 236.9 | 68.4 | 230.1 | 457 | 86.3 | 55.0 | 71.0 | 396 | 348.3 | 109.9 | 330.0 | 801 |
| 8 | 564.5 | 70.1 | 569.0 | 722 | 240.7 | 83.6 | 228.8 | 577 | 81.4 | 42.3 | 72.0 | 303 | 345.9 | 121.5 | 328.5 | 780 |
| 9 | 500.5 | 74.9 | 509.0 | 659 | 217.1 | 79.2 | 208.0 | 460 | 101.1 | 49.1 | 92.0 | 389 | 354.6 | 117.2 | 340.5 | 727 |
| 10 | 515.5 | 98.7 | 530.0 | 718 | 199.8 | 78.7 | 189.4 | 440 | 101.5 | 55.0 | 86.0 | 315 | 323.9 | 110.4 | 307.0 | 625 |
| 11 | 478.8 | 120.4 | 501.0 | 692 | 179.6 | 67.9 | 180.9 | 376 | 121.9 | 73.8 | 99.0 | 438 | 321.7 | 105.8 | 308.0 | 562 |
| 12 | 505.0 | 137.7 | 549.0 | 691 | 163.6 | 57.7 | 165.9 | 321 | 97.5 | 61.7 | 72.0 | 322 | 280.7 | 79.9 | 271.0 | 467 |

**Clear ($k_t > 0.65$)**

| | $B_n$ | | | | $B$ | | | | $D$ | | | | $G$ | | | |
|---|---|---|---|---|---|---|---|---|---|---|---|---|---|---|---|---|
| Month | Mean | StDev | Median | Max | Mean | StDev | Median | Max | Mean | StDev | Median | Max | Mean | StDev | Median | Max |
| 1 | 719.1 | 142.5 | 738.5 | 940 | 344.0 | 109.2 | 351.6 | 546 | 119.0 | 71.6 | 94.0 | 403 | 489.9 | 114.9 | 511.5 | 715 |
| 2 | 743.5 | 131.9 | 760.5 | 960 | 403.2 | 139.6 | 409.3 | 671 | 114.4 | 62.8 | 98.5 | 440 | 549.5 | 151.6 | 583.5 | 880 |
| 3 | 660.4 | 158.8 | 657.0 | 946 | 407.1 | 164.9 | 401.2 | 744 | 146.9 | 95.8 | 118.0 | 556 | 625.4 | 191.8 | 668.0 | 1121 |
| 4 | 678.1 | 160.2 | 696.0 | 945 | 490.9 | 192.8 | 486.7 | 866 | 152.7 | 86.7 | 130.0 | 578 | 713.7 | 201.5 | 759.0 | 1109 |
| 5 | 711.3 | 107.6 | 723.0 | 869 | 546.6 | 179.5 | 564.3 | 832 | 118.3 | 69.9 | 90.0 | 474 | 739.2 | 198.6 | 782.0 | 1088 |
| 6 | 752.3 | 99.6 | 761.0 | 930 | 593.9 | 184.1 | 625.5 | 905 | 118.0 | 61.5 | 95.0 | 585 | 762.7 | 199.2 | 814.0 | 1139 |
| 7 | 765.5 | 75.9 | 781.0 | 917 | 614.7 | 163.5 | 652.4 | 890 | 99.6 | 37.7 | 94.0 | 370 | 762.9 | 174.4 | 808.5 | 1081 |

**Table 7.** *Cont.*

| 8 | 768.3 | 77.8 | 778.0 | 907 | 593.8 | 153.2 | 621.6 | 845 | 96.8 | 41.0 | 87.0 | 304 | 741.2 | 164.3 | 781.0 | 1078 |
|---|-------|------|-------|-----|-------|-------|-------|-----|------|------|------|-----|-------|-------|-------|------|
| 9 | 686.6 | 68.0 | 696.0 | 834 | 495.7 | 115.5 | 515.7 | 679 | 119.5 | 33.3 | 116.0 | 388 | 680.2 | 125.7 | 712.0 | 865 |
| 10 | 723.1 | 103.8 | 727.0 | 899 | 442.7 | 119.9 | 454.0 | 697 | 114.6 | 53.6 | 102.0 | 428 | 587.9 | 118.9 | 611.0 | 854 |
| 11 | 675.9 | 111.6 | 688.0 | 874 | 343.8 | 89.3 | 355.6 | 519 | 132.2 | 61.3 | 115.0 | 402 | 502.9 | 99.6 | 517.0 | 765 |
| 12 | 720.4 | 102.1 | 745.0 | 896 | 336.7 | 74.1 | 347.2 | 473 | 102.9 | 54.6 | 80.0 | 341 | 462.2 | 76.8 | 480.0 | 701 |

### 3.1.7. Shortwave Radiation Indices

The definition of various indices was introduced in the section of measurements and estimation methods. The monthly variability of each index is presented in Figure 21. Monthly means of $k_t$ range between 0.5 and 0.7 with the highest occurring in summer. Almost similar pattern is followed by the beam fraction ($k_b$) ranging from 0.35 to 0.7 and the clearness index obtained from normal direct irradiance ($k_n$) ranging from 0.2 to 0.4. On the other hand, the diffuse fraction ($k_d$) shows higher values in winter and the lowest in summer. The variability of the indices is higher in the winter months as it is indicated from the length of the boxes, but outliers are occurred mainly in the summer months.

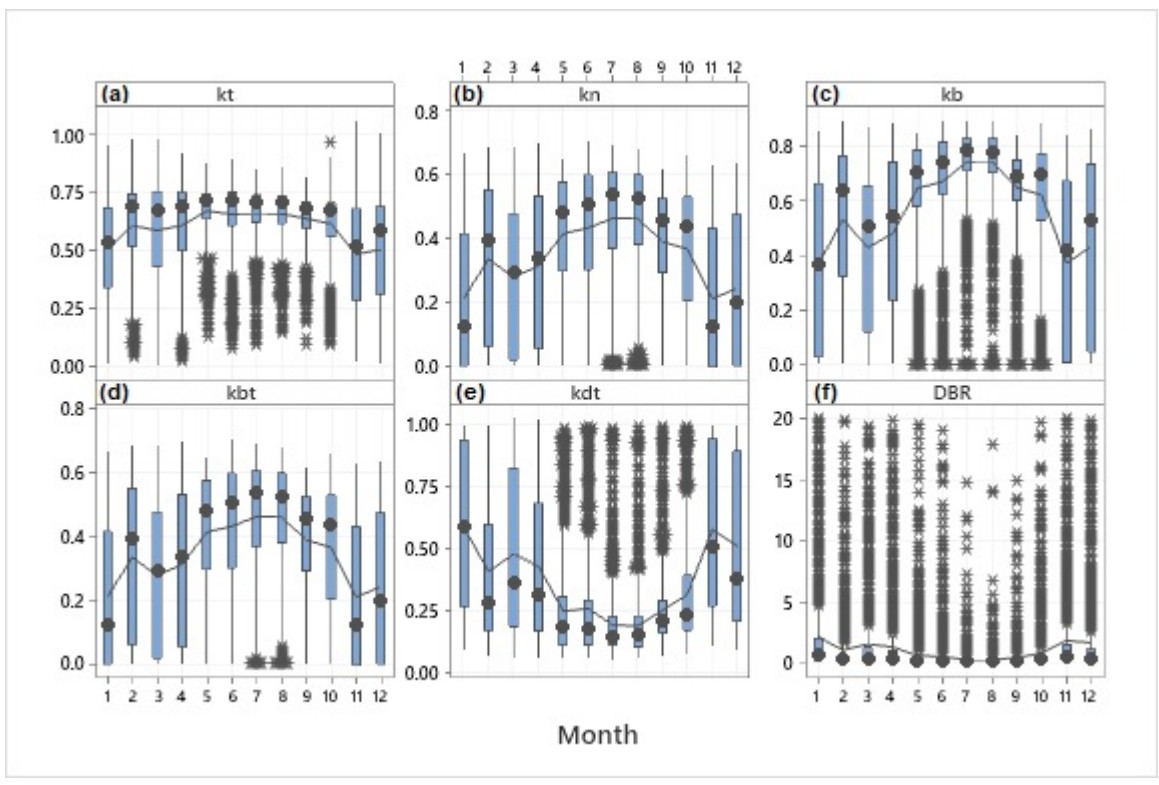

**Figure 21.** Monthly boxplots of: (**a**) clearness index ($k_t$); (**b**) clearness index of direct normal irradiance ($k_n$); (**c**) direct horizontal fraction ($k_b$); (**d**) direct clearness index ($k_{bt}$); (**e**) diffuse fraction ($k_d$); and (**f**) diffuse to beam ratio ($DBR = D/B$).

### 3.1.8. Modeling Radiation Indices

The relationship between the direct normal transmittance ($k_n$), the diffuse fraction ($k_d$), the direct fraction ($k_b$), the beam clearness index ($k_{bt}$) with the clearness index ($k_t$) and DBR with the beam clearness index ($k_{bt}$), are illustrated in Figure 22. The fourth graph of the layout shows the relationship between the direct fraction and the direct normal transmittance which is expressed as a quadratic equation:

$$k_b = 0.023 + 1.993 * k_n - 1.155 * k_n^2 \quad R^2 = 0.968 \tag{34}$$

The correlation between the diffuse to beam ratio and the beam clearness index (Figure 22f) is given as:

$$DBR = 0.316 * k_{bt}^{-0.937} \tag{35}$$

Almost similar statistical values were obtained by Muneer [2] in UK.

The generic multiple predictor logistic model developed by Ridley et al. [23] known as BRL model (Boland-Ridley-Lauret) was used to estimate the diffuse irradiance. The model

is a function of clearness index ($k_t$), the apparent solar time (AST), the solar altitude ($\alpha_s$), the daily clearness index ($K_T$) and the persistence $\psi$ which is an average of both a lag and lead of the clearness index for each time interval of the day. The model was developed based on the one-year data set and has the following form:

$$k_d = 1/(1 + \exp(-5.732 + 5.368 * k_t + 0.031 * AST - 0.011 * a_s + 3.166 * K_T + 2.051 * \psi)) \tag{36}$$

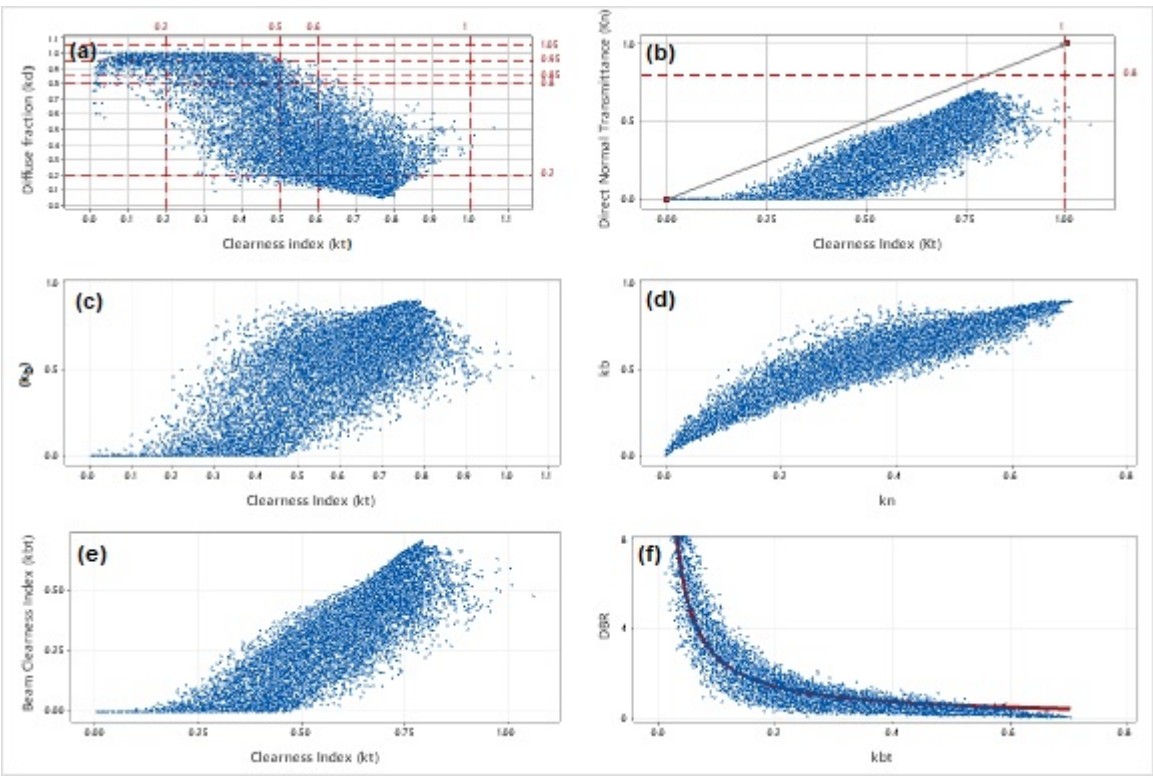

**Figure 22.** (**a**) Diffuse fraction to global irradiance ($k_d$); (**b**) direct normal transmittance ($k_n$); (**c**) fraction of direct horizontal radiation to global irradiance ($k_b$) as a function of clearness index ($k_t$); (**d**) $k_b$ vs. $k_n$; (**e**) $k_{bt}$ vs. $k_t$; and (**f**) DBR vs. $k_{bt}$.

Almost similar values are obtained by Ridley et al. [23] at Adelaide of Australia, i.e., a location with similar climatic conditions with Cyprus. Testing the above equation needs an independent (validation) data set. Then, the estimated direct normal irradiance ($B_{n-est}$) is calculated from the following equation:

$$B_{n-est} = (G - k_d * G) / \cos\theta_z \tag{37}$$

Figure 23 shows the BRL diffuse fraction model fit (Equation (36)) to the clearness index ($k_t$) for the given data set. As shown, the fitting of the spread of the data is much better than a simple line.

*3.2. Daily Values*

3.2.1. Monthly Variability

Figure 14 presents the temporal evolution of daily solar irradiation of all shortwave radiation components. Data reveal a common evolution shape with maxima in summer and minima in winter, due to the daily minimum solar zenith angle and day-length (astronomical factors) variation during the year. Large fluctuations are occurred in spring and autumn during the transition from cold and warm weather and vice versa. The maximum daily global irradiation is reached in June or July, and it is almost 31 MJ/m².

The daily maximum of beam normal irradiation is slightly higher (37.4 MJ/m$^2$), while the maximum horizontal daily irradiation is 26.1 MJ/m$^2$. The maximum daily diffuse irradiation is 13.4 MJ/m$^2$ and is recorded in April.

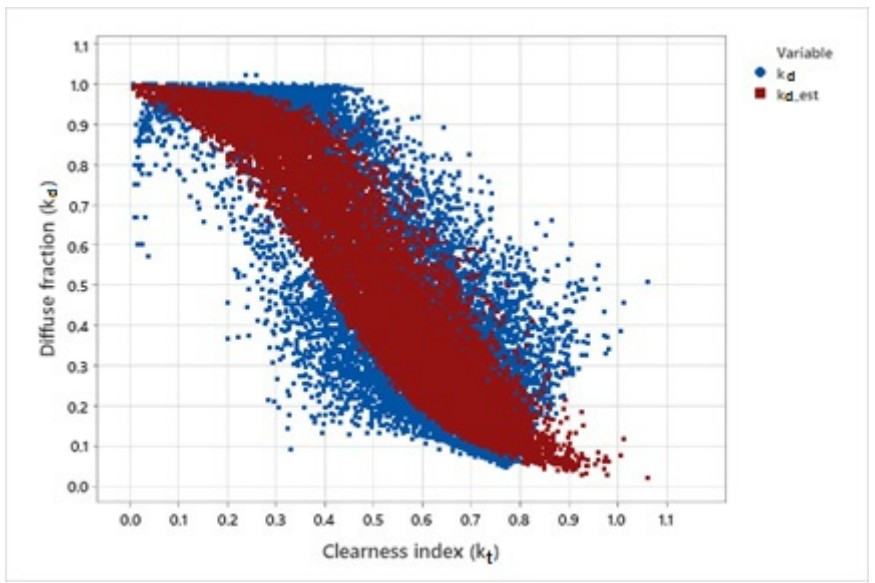

**Figure 23.** The BRL diffuse fraction model fit with the clearness index ($k_t$).

Table 8 shows the mean daily values and their standard deviation of the shortwave components for each month of the year. The variability of the daily values of all the radiation components is also demonstrated through the boxplots of each month of the year (Figure 24). The boxplot presents the mean and median, the IQR as well as the outliers of each variable. As indicated from the length of the boxplots, the spring and winter seasons show the greatest variability. Outliers are observed mainly in summer. For comparison of the levels of all radiation components the daily means are plotted on the same graph (Figure 25). $B_{nd}$ shows the highest values with slightly lower values of $G_d$, while the lowest values are recorded by $UVT_d$. Near Infrared Radiation ($NIR_d$) is slightly higher than photosynthetic active radiation ($PAR_d$). Almost similar values were obtained in Malta [45].

**Table 8.** Mean daily and extreme values and the standard deviations of the shortwave radiation components for each month of the year in MJ/m$^2$. The last column shows the respective daily values of sunshine duration. Note: * means no records.

| | $B_{nd}$ (MJ/m$^2$) | | | | $B_d$ (MJ/m$^2$) | | | | $D_d$ (MJ/m$^2$) | | | | $G_d$ (MJ/m$^2$) | | | | $S_d$ (hrs) | | | |
|---|---|---|---|---|---|---|---|---|---|---|---|---|---|---|---|---|---|---|---|---|
| Month | Mean | StDev | Min | Max | Mean | StDev | Min | Max | Mean | StDev | Min | Max | Mean | StDev | Min | Max | Mean | StDev | Min | Max |
| 1 | 10.6 | 7.2 | 0.6 | 24.7 | 4.6 | 3.2 | 0.2 | 10.4 | 4.3 | 1.4 | 1.9 | 6.8 | 9.5 | 2.7 | 3.2 | 14.0 | 5.1 | 2.6 | 0.3 | 9.1 |
| 2 | 18.0 | 8.1 | 0.0 | 30.2 | 9.1 | 4.2 | 0.0 | 15.6 | 4.3 | 1.6 | 1.6 | 7.3 | 14.3 | 3.5 | 3.9 | 18.4 | 7.6 | 2.6 | 0.0 | 10.1 |
| 3 | 16.2 | 8.5 | 0.5 | 31.2 | 9.5 | 5.0 | 0.3 | 17.8 | 6.3 | 2.3 | 2.1 | 10.1 | 17.8 | 4.2 | 7.6 | 23.8 | * | * | * | * |
| 4 | 19.7 | 10.0 | 1.4 | 35.9 | 13.3 | 6.8 | 0.8 | 23.6 | 7.1 | 2.9 | 2.3 | 13.4 | 22.6 | 5.0 | 12.2 | 27.9 | 9.0 | 3.6 | 0.7 | 12.5 |
| 5 | 27.5 | 5.7 | 13.2 | 34.4 | 19.4 | 3.7 | 9.7 | 23.8 | 5.4 | 2.6 | 2.5 | 11.4 | 27.7 | 1.7 | 22.6 | 29.8 | 12.0 | 1.4 | 8.9 | 14.0 |
| 6 | 29.3 | 5.9 | 14.7 | 37.4 | 20.8 | 4.1 | 9.4 | 26.1 | 5.5 | 2.2 | 2.7 | 11.8 | 28.3 | 2.7 | 21.1 | 30.9 | 12.5 | 1.7 | 8.3 | 13.6 |
| 7 | 31.1 | 2.5 | 25.8 | 35.6 | 22.0 | 1.5 | 18.5 | 25.1 | 4.3 | 1.2 | 2.4 | 7.9 | 28.1 | 0.8 | 26.3 | 30.4 | 12.8 | 0.6 | 11.3 | 13.6 |
| 8 | 29.4 | 3.4 | 21.1 | 35.2 | 20.0 | 2.3 | 14.4 | 23.9 | 3.9 | 1.2 | 2.0 | 6.5 | 25.7 | 1.5 | 21.9 | 27.9 | 12.2 | 0.5 | 10.9 | 12.9 |
| 9 | 23.1 | 3.0 | 13.4 | 27.0 | 14.5 | 2.0 | 8.5 | 17.2 | 4.3 | 1.1 | 2.8 | 8.4 | 21.0 | 1.4 | 17.4 | 23.0 | 10.8 | 0.9 | 8.2 | 11.8 |
| 10 | 20.1 | 5.4 | 7.6 | 28.3 | 10.9 | 3.2 | 3.7 | 16.1 | 4.0 | 1.4 | 2.0 | 7.3 | 15.9 | 2.2 | 11.8 | 19.6 | 9.0 | 1.4 | 5.5 | 10.4 |
| 11 | 10.7 | 6.6 | 0.1 | 22.0 | 4.9 | 3.1 | 0.0 | 9.4 | 4.2 | 1.0 | 2.0 | 5.9 | 9.6 | 2.8 | 2.9 | 13.4 | 5.1 | 2.9 | 0.0 | 8.9 |
| 12 | 11.9 | 7.1 | 0.0 | 22.2 | 5.0 | 2.9 | 0.0 | 8.9 | 3.2 | 1.2 | 1.6 | 5.4 | 8.7 | 2.5 | 1.9 | 11.2 | 5.5 | 2.6 | 0.1 | 8.5 |
| Year | 20.6 | 9.6 | 0.0 | 37.4 | 12.9 | 7.3 | 0.0 | 26.1 | 4.7 | 2.1 | 1.6 | 13.4 | 19.1 | 7.8 | 1.9 | 30.9 | 9.2 | 3.6 | 0.0 | 14.0 |

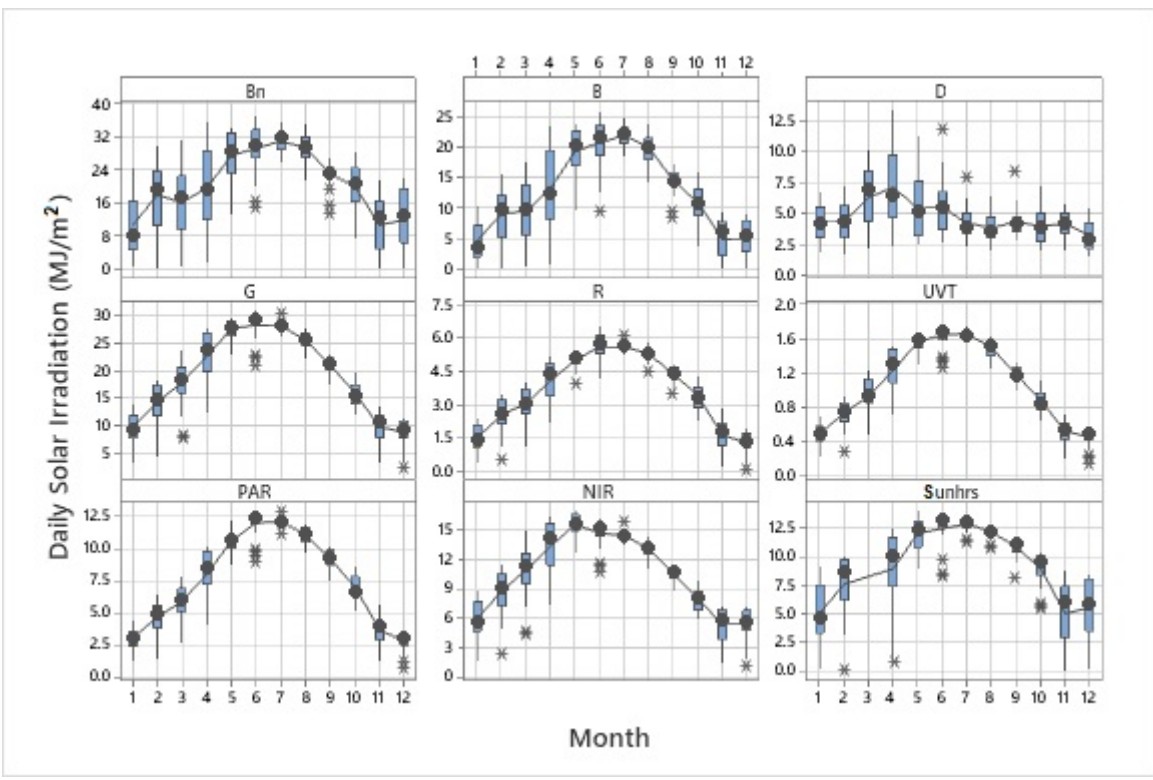

**Figure 24.** Boxplots of daily solar irradiation (MJ/m²) of all radiation components. The smooth line represents the mean daily values of each month for each variable.

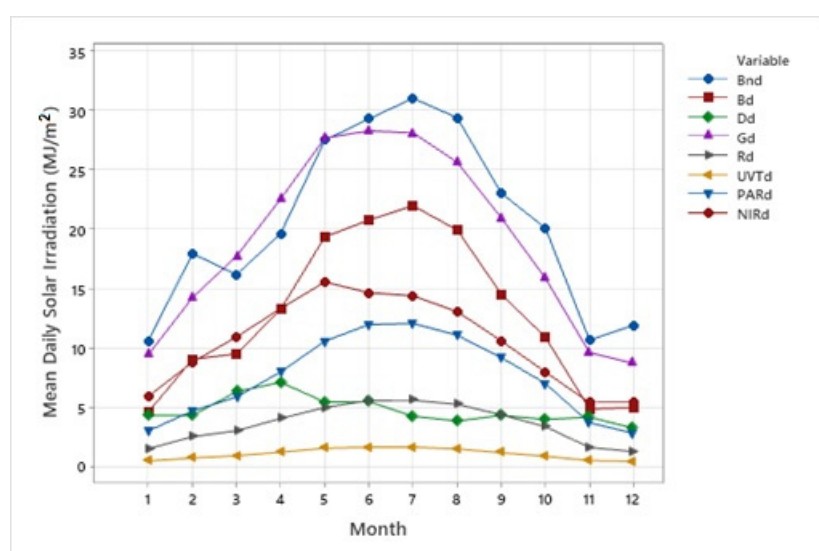

**Figure 25.** Monthly mean daily values (MJ/m²) for all radiation components.

### 3.2.2. Statistical Relationships

Quadratic relationships were established between the daily values of the various radiation components as shown in the following equations. Generally, the equations are associated with high coefficient of determination ($R^2$):

$$B_{nd} = -0.233 + 1.790 * S_d + 0.049 * S_d^2 \quad R^2 = 0.914 \tag{38}$$

$$B_d = -1.993 + 0.595 * G_d + 0.008 * G_d^2 \quad R^2 = 0.915 \tag{39}$$

$$B_d = 0.042 + 0.391 * B_{nd} + 0.009 * B_{nd}^2 \quad R^2 = 0.952 \tag{40}$$

$$B_d = 0.853 + 0.019 * S_d + 0.123 * S_d^2 \quad R^2 = 0.941 \tag{41}$$

$$G_d = 6.656 - 0.109 * S_d + 0.138 * S_d^2 \quad R^2 = 0.890 \tag{42}$$

A linear relationship exists between the daily sum of direct horizontal and diffuse irradiation and the daily global radiation. The coefficient of determination is closed to 1, indicating a perfect relationship. Almost all the points are within the 95% prediction interval:

$$B_d + D_d = 0.346 + 0.902 * G_d \quad R^2 = 0.991 \tag{43}$$

3.2.3. Daily Radiation Indices

The variation of the daily values of radiation indices throughout the year is shown in Figure 26. $K_D$ shows lower values in the summer, while $K_T$ shows higher values.

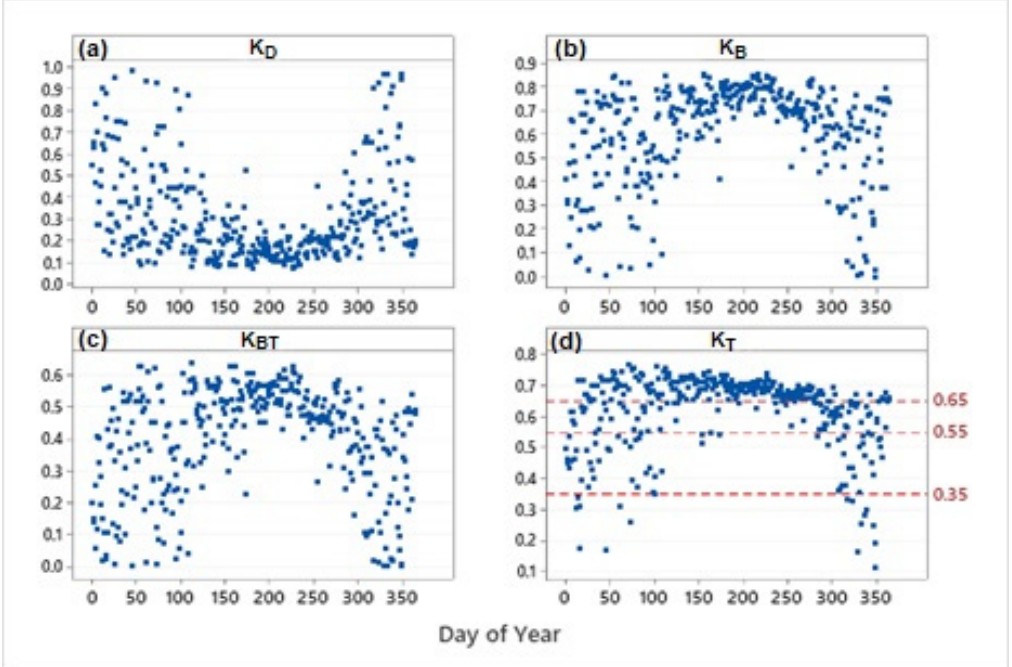

**Figure 26.** Daily values of radiation indices: (**a**) diffuse fraction ($K_D$); (**b**) direct horizontal fraction ($K_B$); (**c**) direct clearness index ($K_{BT}$); and (**d**) clearness index ($K_T$), throughout the year.

The variability of each daily radiation index is shown in Figure 27. As indicated by the length of boxplots, the indices show high variability during the winter and spring seasons since they are affected by the cloud cover. The pattern of $K_B$, $K_{BT}$ and $K_T$ is similar with highest values during the summer. In contrast, $K_D$ has its highest values in the winter.

Daily Clearness Index ($K_T$) and Relative Sunshine Duration ($\sigma$)

$K_T$ is an objective measure of the influence of cloud cover on solar radiation flux. As indicated earlier, the sky conditions could be classified in four intervals of clearness index. The monthly statistics of $K_T$ are reported in Table 9. Also listed in the table is the number of days according to the above classification for each month of the year. The annual average of $K_T$ is 0.620 with a standard deviation of 0.121. The monthly average values of the daily clearness index range between 0.497 in November to 0.692 in July. The annual number of cloudy days is 17 (4.66%), the respective number of partially cloudy days is 137 (37.53%) and 211 days are classified as clear days representing the 57.81% of the annual number of days. As it can be seen the summer months are classified mainly as clear days. It has to be noted that the year of measurements is considered as a dry year.

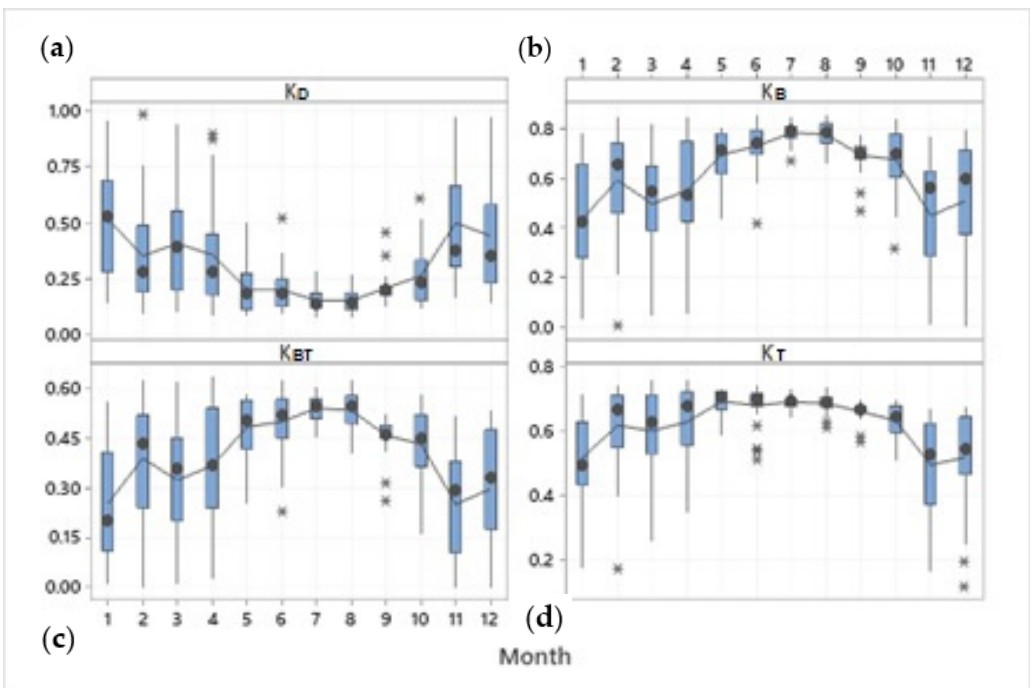

**Figure 27.** Monthly boxplots for each daily radiation index: (**a**) diffuse fraction ($K_D$); (**b**) direct horizontal fraction ($K_B$); (**c**) direct clearness index ($K_{BT}$); and (**d**) clearness index ($K_T$).

**Table 9.** Monthly statistics of KT and the number of days (N) and their percentage according to the classification of days with respect to KT for each month of the year.

| | | | $K_T$ | | | | | | I | | II | | III | | IV | |
|---|---|---|---|---|---|---|---|---|---|---|---|---|---|---|---|---|
| Month | Mean | StDev | CV(%) | Min | Q1 | Median | Q3 | Max | N | Perc(%) | N | Perc(%) | N | Perc(%) | N | Perc(%) |
| 1 | 0.519 | 0.137 | 26.460 | 0.177 | 0.432 | 0.493 | 0.629 | 0.715 | 4 | 12.90 | 13 | 41.94 | 8 | 25.81 | 6 | 19.35 |
| 2 | 0.619 | 0.130 | 20.970 | 0.170 | 0.550 | 0.666 | 0.714 | 0.744 | 1 | 3.57 | 6 | 21.43 | 5 | 17.86 | 16 | 57.14 |
| 3 | 0.603 | 0.132 | 21.930 | 0.261 | 0.532 | 0.628 | 0.716 | 0.764 | 2 | 6.45 | 8 | 25.81 | 9 | 29.03 | 12 | 38.71 |
| 4 | 0.630 | 0.129 | 20.430 | 0.347 | 0.558 | 0.678 | 0.722 | 0.761 | 1 | 3.33 | 5 | 16.67 | 5 | 16.67 | 19 | 63.33 |
| 5 | 0.694 | 0.036 | 5.130 | 0.585 | 0.667 | 0.707 | 0.724 | 0.735 | 0 | | 0 | | 3 | 9.68 | 28 | 90.32 |
| 6 | 0.681 | 0.064 | 9.360 | 0.510 | 0.680 | 0.703 | 0.718 | 0.744 | 0 | | 4 | 13.33 | 1 | 3.33 | 25 | 83.33 |
| 7 | 0.692 | 0.019 | 2.770 | 0.643 | 0.679 | 0.698 | 0.704 | 0.733 | 0 | | 0 | | 1 | 3.23 | 30 | 96.77 |
| 8 | 0.688 | 0.027 | 3.890 | 0.614 | 0.677 | 0.692 | 0.706 | 0.737 | 0 | | 0 | | 3 | 9.68 | 28 | 90.32 |
| 9 | 0.662 | 0.027 | 4.080 | 0.566 | 0.659 | 0.667 | 0.679 | 0.699 | 0 | | 0 | | 5 | 16.67 | 25 | 83.33 |
| 10 | 0.634 | 0.052 | 8.210 | 0.507 | 0.598 | 0.646 | 0.680 | 0.698 | 0 | | 2 | 6.45 | 15 | 48.39 | 14 | 45.16 |
| 11 | 0.497 | 0.138 | 27.700 | 0.163 | 0.373 | 0.530 | 0.626 | 0.671 | 4 | 13.33 | 12 | 40.00 | 13 | 43.33 | 1 | 3.33 |
| 12 | 0.520 | 0.151 | 29.010 | 0.115 | 0.466 | 0.549 | 0.646 | 0.677 | 5 | 16.13 | 11 | 35.48 | 8 | 25.81 | 7 | 22.58 |
| **Year** | 0.620 | 0.121 | 19.440 | 0.115 | 0.577 | 0.664 | 0.700 | 0.763 | 17 | 4.66 | 61 | 16.71 | 76 | 20.82 | 211 | 57.81 |

The annual average on clear days of the direct normal beam radiation is 27.3 MJ/m$^2$, while the respective value of direct horizontal is 17.7 MJ/m$^2$. The annual average of diffuse radiation is 4.2 MJ/m$^2$, while the global radiation is 23.9 MJ/m$^2$. Furthermore, the annual mean of daily sunshine duration on clear days is 11.5 h. The maximum daily sunshine duration is 14.0 h and it was recorded in May. The major difference between day types is the significant reduction in the beam radiation which also affects the global irradiation level. For example, the annual average daily direct normal radiation available for energy conversion on partially cloudy days is reduced by a factor of 0.61 relative to that available on clear days. The respective reduction of global radiation is 0.65.

The descriptive statistics of the sunshine duration as well as the relative sunshine duration are presented in Table 10 for each month of the year. The annual daily average of

sunshine duration is 9.2 h with an annual total of 3358 h. The monthly mean daily values range between 5.1 h in January to 12.8 h in July.

**Table 10.** Descriptive statistics of daily sunshine duration and its relative sunshine duration for each month of the year. Note: * means no records.

| | | Daily Sunshine Duration ($S_d$, hrs) | | | | | | | | Relative Sunshine Duration ($\sigma = S_d/S_{0d}$) | | | | | | | |
|---|---|---|---|---|---|---|---|---|---|---|---|---|---|---|---|---|---|
| Month | N | Mean | StDev | CV(%) | Min | Q1 | Median | Q3 | Max | Mean | StDev | CV(%) | Min | Q1 | Median | Q3 | Max |
| 1 | 31 | 5.1 | 2.6 | 51.2 | 0.3 | 3.4 | 4.7 | 7.5 | 9.1 | 0.510 | 0.260 | 51.020 | 0.025 | 0.341 | 0.484 | 0.754 | 0.907 |
| 2 | 28 | 7.6 | 2.6 | 34.6 | 0.0 | 6.3 | 8.6 | 9.7 | 10.1 | 0.708 | 0.240 | 33.900 | 0.002 | 0.605 | 0.784 | 0.903 | 0.937 |
| 3 | | * | * | * | * | * | * | * | * | * | * | * | * | * | * | * | * |
| 4 | 29 | 9.0 | 3.6 | 40.6 | 0.7 | 7.4 | 10.1 | 11.7 | 12.5 | 0.691 | 0.276 | 39.930 | 0.055 | 0.583 | 0.805 | 0.897 | 0.949 |
| 5 | 31 | 12.0 | 1.4 | 11.2 | 8.9 | 10.8 | 12.4 | 13.1 | 14.0 | 0.868 | 0.091 | 10.440 | 0.657 | 0.805 | 0.901 | 0.939 | 0.984 |
| 6 | 19 | 12.5 | 1.7 | 13.6 | 8.3 | 12.4 | 13.2 | 13.5 | 13.6 | 0.868 | 0.118 | 13.620 | 0.577 | 0.866 | 0.916 | 0.941 | 0.946 |
| 7 | 31 | 12.8 | 0.6 | 4.6 | 11.3 | 12.6 | 13.0 | 13.2 | 13.6 | 0.912 | 0.043 | 4.720 | 0.791 | 0.896 | 0.933 | 0.940 | 0.947 |
| 8 | 31 | 12.2 | 0.5 | 4.4 | 10.9 | 12.0 | 12.2 | 12.6 | 12.9 | 0.920 | 0.039 | 4.240 | 0.805 | 0.920 | 0.935 | 0.946 | 0.953 |
| 9 | 30 | 10.8 | 0.9 | 8.4 | 8.2 | 10.4 | 11.1 | 11.5 | 11.8 | 0.892 | 0.068 | 7.670 | 0.672 | 0.885 | 0.917 | 0.932 | 0.949 |
| 10 | 31 | 9.0 | 1.4 | 15.9 | 5.5 | 8.4 | 9.6 | 10.0 | 10.4 | 0.815 | 0.123 | 15.090 | 0.508 | 0.767 | 0.850 | 0.905 | 0.940 |
| 11 | 30 | 5.1 | 2.9 | 56.6 | 0.0 | 3.0 | 6.1 | 7.4 | 8.9 | 0.501 | 0.284 | 56.640 | 0.002 | 0.289 | 0.588 | 0.730 | 0.898 |
| 12 | 30 | 5.5 | 2.6 | 47.6 | 0.1 | 3.5 | 5.9 | 8.1 | 8.5 | 0.572 | 0.272 | 47.600 | 0.013 | 0.363 | 0.614 | 0.837 | 0.878 |
| **Year** | 321 | 9.2 | 3.6 | 39.4 | 0.0 | 7.1 | 10.0 | 12.2 | 14.0 | 0.748 | 0.246 | 32.830 | 0.002 | 0.661 | 0.856 | 0.924 | 0.984 |

Daily Diffuse Fraction ($K_D$)

Table 11 presents the descriptive statistics of this fraction. $K_D$ ranges from 0.152 in July to 0.514 in January. The daily maximum values exceed 0.9 during the winter months.

**Table 11.** Descriptive statistics of the daily fraction of diffuse to global radiation ($K_D$).

| Month | N | Mean | StDev | CV(%) | Min | Q1 | Median | Q3 | Max |
|---|---|---|---|---|---|---|---|---|---|
| 1 | 31 | 0.514 | 0.239 | 46.45 | 0.142 | 0.277 | 0.529 | 0.683 | 0.951 |
| 2 | 28 | 0.351 | 0.219 | 62.32 | 0.09 | 0.189 | 0.280 | 0.486 | 0.982 |
| 3 | 31 | 0.406 | 0.234 | 57.62 | 0.096 | 0.197 | 0.392 | 0.551 | 0.936 |
| 4 | 30 | 0.356 | 0.219 | 61.59 | 0.081 | 0.180 | 0.280 | 0.448 | 0.898 |
| 5 | 31 | 0.203 | 0.110 | 54.26 | 0.083 | 0.112 | 0.186 | 0.273 | 0.503 |
| 6 | 30 | 0.201 | 0.099 | 49.26 | 0.087 | 0.125 | 0.186 | 0.246 | 0.522 |
| 7 | 31 | 0.152 | 0.045 | 29.66 | 0.078 | 0.118 | 0.140 | 0.185 | 0.285 |
| 8 | 31 | 0.152 | 0.052 | 34.16 | 0.073 | 0.112 | 0.142 | 0.188 | 0.269 |
| 9 | 30 | 0.209 | 0.063 | 30.26 | 0.126 | 0.174 | 0.203 | 0.219 | 0.456 |
| 10 | 31 | 0.263 | 0.123 | 46.64 | 0.112 | 0.155 | 0.231 | 0.333 | 0.609 |
| 11 | 30 | 0.498 | 0.245 | 49.22 | 0.163 | 0.303 | 0.376 | 0.661 | 0.971 |
| 12 | 31 | 0.440 | 0.258 | 58.57 | 0.143 | 0.229 | 0.351 | 0.579 | 0.970 |
| **Year** | 365 | 0.312 | 0.216 | 69.39 | 0.073 | 0.154 | 0.229 | 0.392 | 0.982 |

Figure 28a shows the relationship between the $K_D$ and $K_T$, while Figure 28b shows the relationship between $K_D$ and daily relative sunshine ($\sigma$). The cubic equations of the above relationships have the following form:

$$K_D = 0.818 + 1.679 * K_T - 6.423 * K_T^2 + 3.866 * K_T^3 \quad R^2 = 0.871 \tag{44}$$

$$K_D = 0.975 - 1.382 * \sigma + 1.189 * \sigma^2 - 0.704 * \sigma^3 \quad R^2 = 0.905 \tag{45}$$

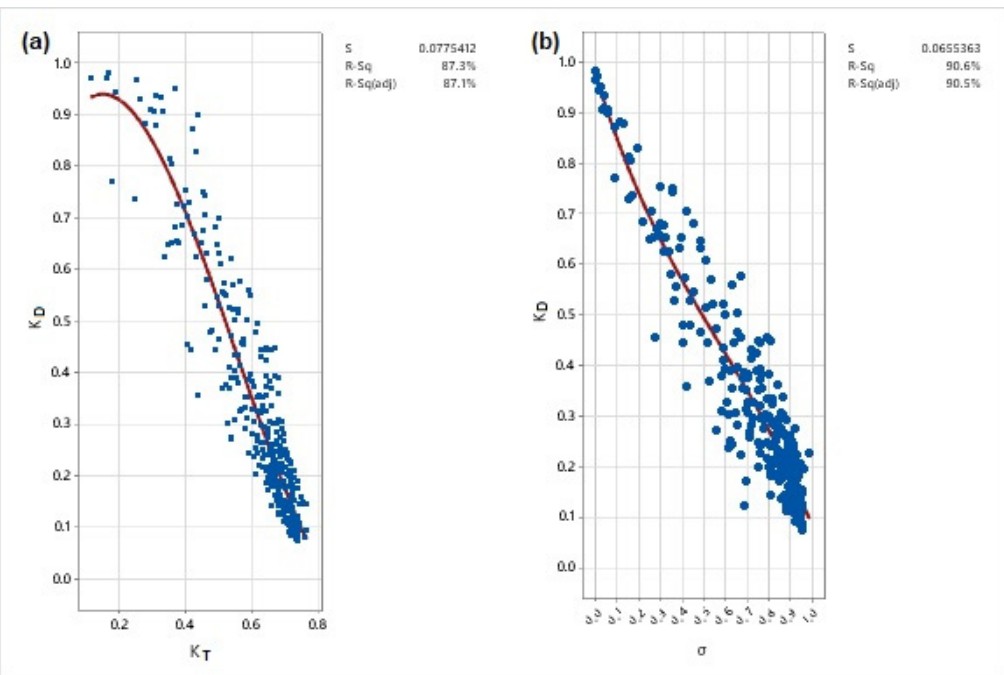

**Figure 28.** (**a**) relation between $K_D$ and $K_T$; and (**b**) relation between $K_D$ and $\sigma$.

The characteristic of these relations is that the coefficients of determination are closed to 0.9.

Daily Direct Horizontal Fraction ($K_B$) and Daily Direct Transmittance ($K_{BT}$)

The daily direct horizontal fraction is the ratio of daily direct to the daily global irradiation, while the direct transmittance is the ratio of direct horizontal to the extraterrestrial irradiation. Table 12 presents the descriptive statistics of these indices. $K_B$ ranges from 0.434 in January to 0.781 in July. The daily maximum values exceed 0.75 during the winter months and 0.8 in the rest of the months. $K_{BT}$ ranges from 0.25 in January to 0.541 in July. Maximum values exceed 0.5 throughout the year.

**Table 12.** Descriptive statistics of the daily fraction of direct to global radiation ($K_B$) and daily direct transmittance ($K_{BT}$).

| | | $K_B$ | | | | | | | | $K_{BT}$ | | | | | | | |
|---|---|---|---|---|---|---|---|---|---|---|---|---|---|---|---|---|---|
| Month | N | Mean | StDev | CV(%) | Min | Q1 | Median | Q3 | Max | Mean | StDev | CV(%) | Min | Q1 | Median | Q3 | Max |
| 1 | 31 | 0.434 | 0.226 | 52.03 | 0.029 | 0.276 | 0.423 | 0.657 | 0.784 | 0.252 | 0.172 | 68.26 | 0.011 | 0.109 | 0.202 | 0.406 | 0.561 |
| 2 | 28 | 0.59 | 0.208 | 35.20 | 0.003 | 0.455 | 0.657 | 0.742 | 0.849 | 0.39 | 0.171 | 43.86 | 0.001 | 0.241 | 0.437 | 0.520 | 0.627 |
| 3 | 31 | 0.495 | 0.208 | 42.04 | 0.038 | 0.383 | 0.543 | 0.649 | 0.818 | 0.324 | 0.172 | 53.28 | 0.010 | 0.200 | 0.357 | 0.452 | 0.622 |
| 4 | 30 | 0.548 | 0.213 | 38.89 | 0.052 | 0.423 | 0.528 | 0.75 | 0.846 | 0.367 | 0.181 | 49.17 | 0.023 | 0.239 | 0.371 | 0.542 | 0.639 |
| 5 | 31 | 0.694 | 0.098 | 14.10 | 0.428 | 0.615 | 0.709 | 0.78 | 0.802 | 0.485 | 0.088 | 18.23 | 0.251 | 0.416 | 0.502 | 0.566 | 0.586 |
| 6 | 30 | 0.728 | 0.093 | 12.81 | 0.415 | 0.694 | 0.737 | 0.794 | 0.856 | 0.501 | 0.099 | 19.68 | 0.226 | 0.451 | 0.521 | 0.568 | 0.628 |
| 7 | 31 | 0.781 | 0.04 | 5.08 | 0.672 | 0.761 | 0.793 | 0.808 | 0.846 | 0.541 | 0.039 | 7.30 | 0.451 | 0.510 | 0.549 | 0.568 | 0.604 |
| 8 | 31 | 0.776 | 0.052 | 6.75 | 0.657 | 0.741 | 0.787 | 0.818 | 0.855 | 0.535 | 0.054 | 10.17 | 0.404 | 0.496 | 0.547 | 0.578 | 0.627 |
| 9 | 30 | 0.69 | 0.063 | 9.06 | 0.464 | 0.677 | 0.700 | 0.724 | 0.777 | 0.458 | 0.054 | 11.89 | 0.263 | 0.451 | 0.463 | 0.490 | 0.526 |
| 10 | 31 | 0.674 | 0.125 | 18.49 | 0.313 | 0.601 | 0.696 | 0.779 | 0.837 | 0.433 | 0.108 | 24.88 | 0.159 | 0.365 | 0.452 | 0.522 | 0.582 |
| 11 | 30 | 0.448 | 0.231 | 51.62 | 0.004 | 0.282 | 0.559 | 0.628 | 0.770 | 0.251 | 0.159 | 63.10 | 0.001 | 0.106 | 0.293 | 0.382 | 0.517 |
| 12 | 31 | 0.507 | 0.246 | 48.48 | 0.001 | 0.372 | 0.594 | 0.713 | 0.794 | 0.297 | 0.173 | 58.23 | 0.000 | 0.176 | 0.331 | 0.476 | 0.537 |
| **Year** | 365 | 0.614 | 0.205 | 33.33 | 0.001 | 0.523 | 0.691 | 0.761 | 0.856 | 0.403 | 0.165 | 41.02 | 0.000 | 0.309 | 0.454 | 0.527 | 0.639 |

Figure 29 shows the relationships between $K_B$ and $K_T$, $K_B$ and $\sigma$, as well as between $K_B$ and $K_D$. Similar equations were developed for $K_{BT}$ and other radiation indices. The cubic equations of the above relationships have the following form:

$$K_B = 0.160 - 1.656 * K_T + 6.150 * K_T^2 - 3.704 * K_T^3 \; R^2 = 0.837 \tag{46}$$

$$K_B = 0.00004 + 1.310 * \sigma - 1.187 * \sigma^2 + 0.714 * \sigma^3 \; R^2 = 0.880 \tag{47}$$

$$K_B = 0.923 - 1.071 * K_D + 0.252 * K_D^2 - 0.127 * K_D^3 \; R^2 = 0.978 \tag{48}$$

$$K_{BT} = 0.117 - 1.142 * K_T + 3.104 * K_T^2 - 0.930 * K_T^3 \; R^2 = 0.911 \tag{49}$$

$$K_{BT} = 0.0048 + 0.428 * \sigma + 0.0017 * \sigma^2 + 0.175 * \sigma^3 \; R^2 = 0.894 \tag{50}$$

$$K_{BT} = -0.0023 + 0.277 * K_B + 0.714 * K_B^2 - 0.214 * K_B^3 \; R^2 = 0.979 \tag{51}$$

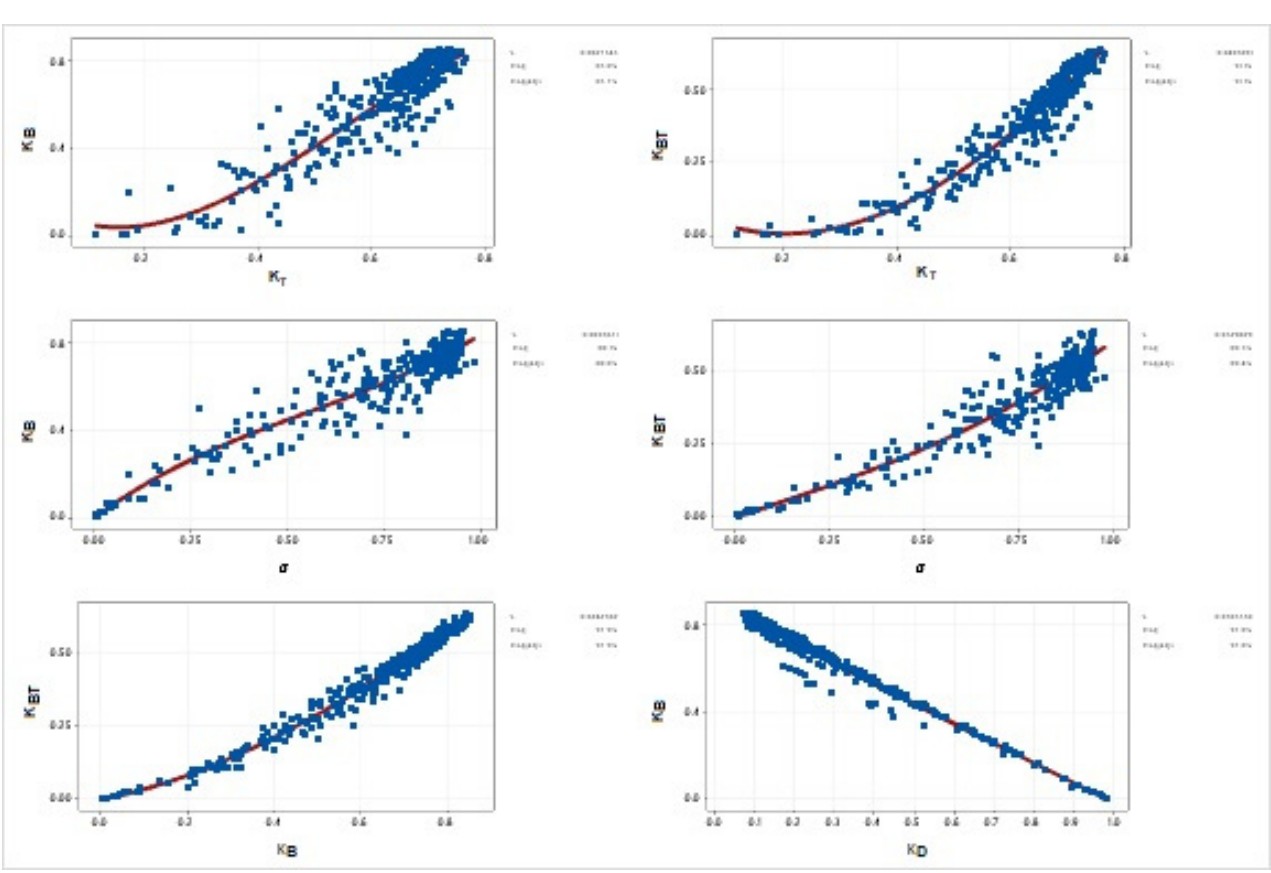

**Figure 29.** Relationships between various daily radiation indices.

## 4. Calculation of Monthly Total Energy on an Inclined Surface

The total solar energy which is received by a horizontal surface depends on the prevailing meteorological conditions and astronomical factors. For practical purposes, we use Tables with the monthly mean daily solar radiation values obtained from measurements on horizontal surfaces as shown in Table 13. The data refers to the mean daily values on the 15th day of each month. The table shows the day number ($d_n$), the solar declination ($\delta$), and the sunset hour angle ($\omega_s$) in degrees, the measured shortwave radiation in kWh/m²/d, the radiation indices, the daily totals at the top of the atmosphere in kWh/m²/d, and the monthly totals of the measured shortwave variables in kWh/m²/month. Details of the calculation procedure are given in [24,26].

In practice, photovoltaics are installed with a slope with respect to a horizontal surface and in southern direction. For the estimation of solar radiation on an inclined surface,

there are three types of solar radiation, namely, (i) beam; (ii) diffuse; and (iii) reflected solar radiation from the surfaces surrounding it. Hence, there are three conversion factors, i.e., for beam ($R_B$), diffuse ($R_D$) and reflected ($R_R$) solar radiation factors. These conversion factors convert the beam and diffuse solar radiation of a horizontal surface to those of an inclined surface.

The beam radiation which is received by an inclined surface depends on the incident angles on the horizontal and inclined surfaces. When the solar panel is installed on a horizontal surface with respect to south the incident angle is the solar zenith angle which is given in Equation (2). When the solar panel is installed with a slope $\beta$ with respect to the horizontal level and in the south direction, the incident angle ($\theta$) has the following form:

$$\cos(\theta) = \cos(\phi - \beta) * \cos(\delta) * \cos(\omega) + \sin(\phi - \beta) * \sin(\delta) \tag{52}$$

where $\varphi$ is the latitude, $\delta$ is the solar declination and $\omega$ is the hour angle. Then the factor $R_B$ can be estimated from the ratio of the incident angles:

$$R_B = \frac{\cos(\theta)}{\cos(\theta_z)} = \frac{\cos(\phi - \beta) * \cos(\delta) * \cos(\omega) + \sin(\phi - \beta) * \sin(\delta)}{\cos(\phi) * \cos(\delta) * \cos(\omega) + \sin(\phi) * \sin(\delta)} \tag{53}$$

As a result, $R_B$ depends on the latitude ($\varphi$), day of the year ($\delta$), hour of day ($\omega$) and the slope of the panel ($\beta$). Therefore, it changes during the day and the time of the year. The daily calculation of $R_B$ can be done through the integration of the nominator of the solar hour angle from sunrise to sunset of Equation (53) for the inclined surface:

$$R_B = \frac{\cos(\phi - \beta) * \cos(\delta) * \sin(\omega'_s) + (\pi/180) * \omega'_s * \sin(\phi - \beta) * \sin(\delta)}{\cos(\phi) * \cos(\delta) * \sin(\omega_s) + (\pi/180) * \omega_s * \sin(\phi) * \sin(\delta)} \tag{54}$$

where $\omega_s$ is the sunset hour angle when the panel is horizontal, and $\omega_s'$ is the sunset hour angle when the panel has a slope $\beta$ with respect to the horizontal surface. Both equations have the following form:

$$\omega_s = \cos^{-1}(-\tan(\phi) * \tan(\delta)) \tag{55}$$

$$\omega'_s = \min(\omega_s, \cos^{-1}(-\tan(\phi - \beta) * \tan(\delta))) \tag{56}$$

The conversion factor for the diffuse radiation ($R_D$) is defined as the ratio of diffuse radiation incident on an inclined surface to that in a horizontal surface. The diffuse factor can be estimated by considering the sky as an isotropic source of diffuse radiation. Then $R_D$ is given as follows:

$$R_D = (1 + \cos(\beta))/2 \tag{57}$$

The reflected solar radiation is the reflected radiation from the ground and other objects near surface of interest. Assuming that reflected radiation is diffuse and isotropic, the conversion factor for reflected radiation ($R_R$) is given by:

$$R_R = (1 - \cos(\beta))/2 \tag{58}$$

The monthly conversion factor ($R_M$) is the ratio of total solar radiation incident on an inclined/tilted surface to that on a horizontal surface and is given by:

$$R_M = (1 - D_d/G_d) * R_B + (D_d/G_d) * (1 + \cos(\beta))/2 + \rho * (1 - \cos(\beta))/2 \tag{59}$$

Or if we express the ratio of $D_d/G_d$ as the fraction of diffuse to global radiation ($K_D$):

$$R_M = (1 - K_D) * R_B + K_D * (1 + \cos(\beta))/2 + \rho * (1 - \cos(\beta))/2 \tag{60}$$

After the calculation of mean monthly conversion factor ($R_M$), the total solar energy received by the inclined surface ($G_{iM}$) is given by:

$$G_{iM} = R_M * G_{hM} \tag{61}$$

where $G_{hM}$ is the total horizontal irradiation in kWh/m$^2$/month. The following relations give the total monthly direct energy ($B_{iM}$), diffuse energy ($D_{iM}$) and reflected energy ($R_{iM}$) which is received by an incline panel:

$$B_{iM} = R_B * (1 - K_D) * G_{hM} \tag{62}$$

$$D_{iM} = R_D * K_D * G_{hM} \tag{63}$$

$$R_{iM} = R_R * \rho * G_{hM} \tag{64}$$

The results of these calculations are presented in Tables 14 and 15. Table 14 presents the estimated $R_B$ and $R_M$ conversion factors, while Table 15 presents the monthly total energy in kWh/m$^2$/month, received by an inclined surface. The highest values for each month and the optimal slope are also highlighted. As it is indicated from the tables the optimum slope angles during the winter-time range between 50 and 60°, while during summer the slopes are close to zero. In spring the optimum slope angle is 15°, in summer is 0°, in autumn is 45° and in winter is 55°. The annual optimum angle is about 26° which is about 10° less than the latitude of the location. Table 16 presents the monthly total energy in kWh/m$^2$/month, received by an inclined surface for annual optimum angles ranging from 20° to 32°.

Table 17 compares the annual energy totals for different slopes ($G_i$) with those measured on a horizontal surface ($G_h$). For different optimum slopes for each month of the year, we estimate an annual amount of energy equal of 2221 kWh/m$^2$, while the measured total horizontal energy is 1938 kWh/m$^2$. Therefore, the percentage increase of energy is 14.6%. For the case of a permanent slope angle of 26° throughout the year the inclined panel receives 2097 kWh/m$^2$ with the increase of about 8.2% from the horizontal level. For a slope angle of less than 15° from the latitude (35°) we have high production during the summer months. The annual increase is 7.7%. For the case of a slope which is higher by 15° from the latitude, we have high production in the winter months, but the annual increase is only 1.3%. Finally, for the case the slope $\beta = \varphi$ (35°) we have high production during the months of March and September. The annual increase in energy with respect to the horizontal level is 7.3%. Therefore, higher total energy can be obtained if we use two different slopes one for the summer and one for winter period. For instance, if we use a slope of 10° for the summer period (April–September), then the total energy for this period is 1313 kWh/m$^2$, while if we use a slope of 50° for the winter period (October–March), then the total energy for this period is 878 kWh/m$^2$. Therefore, the annual total by using the two slopes during the year is 2191 kW/m$^2$, which is higher by about 95 kWh/m$^2$ from the case of an annual permanent slope of 26°. As a result, the case with two slopes ($\beta = 10°$ for summer and $\beta = 50°$ for winter) could have an increase of total annual energy by 13% with respect to the horizontal one.

**Table 13.** Monthly mean daily values which are used to estimate total monthly radiation received by a horizontal surface. The data refers to the mean daily values on the 15th day of each month. Note: * means no records.

| Athalassa | φ = 35.15 | | | | Mean Daily (kWh/m²/d) | | | | | | | | | | | | | | $G_{0n}$ (kWh/m²/d) | $G_{0d}$ (kWh/m²/d) | Monthly Totals kWh/m²/month | | | | | | |
|---|---|---|---|---|---|---|---|---|---|---|---|---|---|---|---|---|---|---|---|---|---|---|---|---|---|---|---|
| Month | Days | $d_n$ | $\delta$, deg | $w_s$, deg | $B_n$ | $B$ | $D$ | $G$ | $R$ | $R_{ns}$ | $K_T$ | $K_B$ | $K_D$ | $\sigma$ | $\rho$ | $S_d$, hrs | $S_{0d}$, hrs | | | $B_n$ | $B$ | $D$ | $G$ | $R$ | $R_{ns}$ | $G_{0m}$ |
| 1 | 31 | 15 | −21.269 | 74.092 | 2.95 | 1.28 | 1.20 | 2.63 | 0.42 | 2.21 | 0.519 | 0.434 | 0.514 | 0.510 | 0.155 | 5.06 | 9.88 | 10.78 | 4.99 | 91.36 | 39.77 | 37.18 | 81.42 | 13.00 | 68.42 | 154.62 |
| 2 | 28 | 46 | −13.289 | 80.427 | 4.99 | 2.52 | 1.20 | 3.96 | 0.70 | 3.26 | 0.619 | 0.590 | 0.351 | 0.708 | 0.174 | 7.60 | 10.72 | 10.69 | 6.40 | 139.69 | 70.47 | 33.69 | 110.84 | 19.67 | 91.17 | 179.22 |
| 3 | 31 | 74 | −2.819 | 88.013 | 4.50 | 2.64 | 1.76 | 4.93 | 0.84 | 4.09 | 0.603 | 0.495 | 0.406 | 0.656 | 0.169 | * | 11.74 | 10.54 | 8.15 | 139.42 | 81.94 | 54.50 | 152.87 | 26.12 | 126.76 | 252.58 |
| 4 | 30 | 105 | 9.415 | 96.705 | 5.46 | 3.69 | 1.98 | 6.27 | 1.13 | 5.14 | 0.630 | 0.548 | 0.356 | 0.691 | 0.180 | 8.96 | 12.89 | 10.36 | 9.95 | 163.75 | 110.58 | 59.31 | 188.14 | 33.89 | 154.25 | 298.44 |
| 5 | 31 | 135 | 18.792 | 103.862 | 7.65 | 5.38 | 1.51 | 7.70 | 1.39 | 6.31 | 0.694 | 0.694 | 0.203 | 0.868 | 0.180 | 12.03 | 13.85 | 10.21 | 11.10 | 237.07 | 166.79 | 46.91 | 238.55 | 43.02 | 195.52 | 344.11 |
| 6 | 30 | 166 | 23.314 | 107.665 | 8.14 | 5.78 | 1.52 | 7.86 | 1.55 | 6.31 | 0.681 | 0.728 | 0.201 | 0.868 | 0.197 | 12.46 | 14.36 | 10.11 | 11.56 | 244.34 | 173.28 | 45.73 | 235.89 | 46.49 | 189.41 | 346.84 |
| 7 | 31 | 196 | 21.517 | 106.117 | 8.63 | 6.11 | 1.18 | 7.82 | 1.56 | 6.26 | 0.692 | 0.781 | 0.152 | 0.912 | 0.199 | 12.85 | 14.15 | 10.10 | 11.34 | 267.40 | 189.36 | 36.65 | 242.28 | 48.21 | 194.06 | 351.42 |
| 8 | 31 | 227 | 13.784 | 99.947 | 8.17 | 5.55 | 1.07 | 7.13 | 1.46 | 5.67 | 0.688 | 0.776 | 0.152 | 0.920 | 0.205 | 12.18 | 13.33 | 10.19 | 10.41 | 253.36 | 171.91 | 33.19 | 220.96 | 45.29 | 175.67 | 322.74 |
| 9 | 30 | 258 | 2.217 | 91.562 | 6.41 | 4.04 | 1.20 | 5.83 | 1.22 | 4.61 | 0.662 | 0.690 | 0.209 | 0.892 | 0.209 | 10.84 | 12.21 | 10.35 | 8.82 | 192.16 | 121.24 | 36.08 | 174.88 | 36.50 | 138.38 | 264.65 |
| 10 | 31 | 288 | −9.599 | 83.161 | 5.59 | 3.03 | 1.11 | 4.41 | 0.94 | 3.48 | 0.634 | 0.674 | 0.263 | 0.815 | 0.212 | 8.99 | 11.09 | 10.53 | 6.96 | 173.24 | 94.04 | 34.32 | 136.86 | 29.10 | 107.75 | 215.76 |
| 11 | 30 | 319 | −19.148 | 75.849 | 2.97 | 1.36 | 1.16 | 2.67 | 0.45 | 2.22 | 0.497 | 0.448 | 0.498 | 0.501 | 0.167 | 5.07 | 10.11 | 10.69 | 5.33 | 89.00 | 40.68 | 34.84 | 80.18 | 13.56 | 66.63 | 159.99 |
| 12 | 31 | 349 | −23.335 | 72.317 | 3.31 | 1.38 | 0.90 | 2.42 | 0.35 | 2.07 | 0.520 | 0.507 | 0.440 | 0.572 | 0.148 | 5.53 | 9.64 | 10.78 | 4.61 | 102.73 | 42.82 | 27.93 | 75.00 | 10.95 | 64.05 | 142.82 |

**Table 14.** $R_B$ and $R_M$ conversion factors for each month of the year for different slope angles ($\beta$) at Athalassa. The optimum slope angles for each month are highlighted.

| Slope | $R_B$ | | | | | | | | | | | | $R_M$ | | | | | | | | | | | |
|---|---|---|---|---|---|---|---|---|---|---|---|---|---|---|---|---|---|---|---|---|---|---|---|---|
| $\beta$, deg | Jan | Feb | Mar | Apr | May | Jun | Jul | Aug | Sep | Oct | Nov | Dec | Jan | Feb | Mar | Apr | May | Jun | Jul | Aug | Sep | Oct | Nov | Dec |
| 0 | 1.000 | 1.000 | 1.000 | 1.000 | 1.000 | 1.000 | 1.000 | 1.000 | 1.000 | 1.000 | 1.000 | 1.000 | 1.000 | 1.000 | 1.000 | 1.000 | 1.000 | 1.000 | 1.000 | 1.000 | 1.000 | 1.000 | 1.000 | 1.000 |
| 5 | 1.166 | 1.115 | 1.068 | 1.027 | 1.001 | 0.989 | 0.994 | 1.015 | 1.050 | 1.097 | 1.150 | 1.182 | 1.080 | 1.074 | 1.040 | 1.017 | 1.001 | 0.991 | 0.995 | 1.012 | 1.039 | 1.071 | 1.075 | 1.102 |
| 10 | 1.322 | 1.221 | 1.128 | 1.047 | 0.996 | 0.973 | 0.982 | 1.022 | 1.092 | 1.185 | 1.292 | 1.355 | 1.154 | 1.142 | 1.074 | 1.029 | 0.996 | 0.978 | 0.985 | 1.019 | 1.073 | 1.136 | 1.144 | 1.197 |
| 15 | 1.469 | 1.319 | 1.179 | 1.059 | 0.984 | 0.951 | 0.964 | 1.022 | 1.125 | 1.264 | 1.424 | 1.518 | 1.222 | 1.204 | 1.102 | 1.035 | 0.987 | 0.960 | 0.970 | 1.020 | 1.099 | 1.194 | 1.207 | 1.285 |
| 20 | 1.605 | 1.406 | 1.221 | 1.063 | 0.965 | 0.922 | 0.939 | 1.016 | 1.150 | 1.334 | 1.545 | 1.670 | 1.283 | 1.258 | 1.124 | 1.035 | 0.972 | 0.938 | 0.950 | 1.015 | 1.119 | 1.244 | 1.264 | 1.366 |
| 25 | 1.728 | 1.482 | 1.254 | 1.059 | 0.940 | 0.889 | 0.909 | 1.001 | 1.167 | 1.393 | 1.654 | 1.808 | 1.337 | 1.305 | 1.140 | 1.030 | 0.951 | 0.911 | 0.925 | 1.004 | 1.132 | 1.287 | 1.313 | 1.439 |
| 30 | 1.838 | 1.548 | 1.278 | 1.048 | 0.909 | 0.849 | 0.873 | 0.980 | 1.174 | 1.442 | 1.750 | 1.933 | 1.383 | 1.343 | 1.149 | 1.019 | 0.926 | 0.879 | 0.895 | 0.987 | 1.138 | 1.322 | 1.355 | 1.503 |
| 35 | 1.934 | 1.601 | 1.292 | 1.028 | 0.872 | 0.804 | 0.831 | 0.952 | 1.173 | 1.480 | 1.833 | 2.043 | 1.421 | 1.374 | 1.152 | 1.002 | 0.896 | 0.843 | 0.861 | 0.964 | 1.136 | 1.349 | 1.389 | 1.558 |
| 40 | 2.016 | 1.642 | 1.296 | 1.001 | 0.828 | 0.755 | 0.783 | 0.917 | 1.162 | 1.506 | 1.903 | 2.138 | 1.451 | 1.396 | 1.148 | 0.980 | 0.860 | 0.804 | 0.822 | 0.936 | 1.128 | 1.367 | 1.415 | 1.604 |
| 45 | 2.082 | 1.671 | 1.290 | 0.967 | 0.779 | 0.700 | 0.731 | 0.875 | 1.143 | 1.522 | 1.958 | 2.216 | 1.473 | 1.409 | 1.137 | 0.953 | 0.821 | 0.760 | 0.779 | 0.902 | 1.113 | 1.377 | 1.433 | 1.639 |
| 50 | 2.132 | 1.687 | 1.274 | 0.926 | 0.725 | 0.642 | 0.674 | 0.827 | 1.115 | 1.525 | 1.997 | 2.278 | 1.486 | 1.414 | 1.120 | 0.921 | 0.777 | 0.713 | 0.732 | 0.863 | 1.091 | 1.378 | 1.442 | 1.664 |
| 55 | 2.166 | 1.690 | 1.248 | 0.877 | 0.667 | 0.580 | 0.614 | 0.773 | 1.079 | 1.517 | 2.022 | 2.322 | 1.490 | 1.410 | 1.097 | 0.883 | 0.730 | 0.663 | 0.682 | 0.819 | 1.062 | 1.370 | 1.443 | 1.679 |
| 60 | 2.184 | 1.680 | 1.213 | 0.823 | 0.604 | 0.515 | 0.549 | 0.714 | 1.034 | 1.498 | 2.031 | 2.349 | 1.485 | 1.397 | 1.068 | 0.842 | 0.679 | 0.611 | 0.629 | 0.771 | 1.027 | 1.354 | 1.436 | 1.683 |
| 65 | 2.185 | 1.658 | 1.169 | 0.762 | 0.538 | 0.447 | 0.482 | 0.650 | 0.982 | 1.467 | 2.025 | 2.357 | 1.472 | 1.376 | 1.032 | 0.796 | 0.625 | 0.557 | 0.574 | 0.719 | 0.985 | 1.329 | 1.420 | 1.676 |
| 70 | 2.169 | 1.623 | 1.116 | 0.696 | 0.469 | 0.378 | 0.413 | 0.582 | 0.922 | 1.424 | 2.004 | 2.348 | 1.450 | 1.346 | 0.991 | 0.746 | 0.569 | 0.502 | 0.518 | 0.663 | 0.938 | 1.296 | 1.396 | 1.659 |
| 75 | 2.137 | 1.575 | 1.054 | 0.625 | 0.397 | 0.308 | 0.342 | 0.511 | 0.855 | 1.371 | 1.967 | 2.321 | 1.419 | 1.308 | 0.945 | 0.693 | 0.511 | 0.446 | 0.460 | 0.605 | 0.885 | 1.255 | 1.363 | 1.632 |
| 80 | 2.088 | 1.516 | 0.985 | 0.550 | 0.325 | 0.239 | 0.272 | 0.436 | 0.782 | 1.308 | 1.915 | 2.276 | 1.380 | 1.261 | 0.893 | 0.637 | 0.452 | 0.390 | 0.402 | 0.544 | 0.827 | 1.206 | 1.323 | 1.594 |
| 85 | 2.024 | 1.445 | 0.908 | 0.471 | 0.252 | 0.171 | 0.202 | 0.359 | 0.702 | 1.235 | 1.849 | 2.214 | 1.333 | 1.208 | 0.837 | 0.579 | 0.393 | 0.336 | 0.344 | 0.481 | 0.764 | 1.150 | 1.276 | 1.547 |
| 90 | 1.944 | 1.363 | 0.824 | 0.389 | 0.180 | 0.107 | 0.135 | 0.281 | 0.618 | 1.152 | 1.768 | 2.135 | 1.279 | 1.147 | 0.777 | 0.518 | 0.335 | 0.285 | 0.290 | 0.417 | 0.697 | 1.086 | 1.221 | 1.490 |

**Table 15.** Monthly total energy received by an inclined surface for each month of the year and for different slope angles ($\beta$) at Athalassa. The optimum slope angles for each month are highlighted.

| Slope | $G_M$ (kWh/m²) | | | | | | | | | | | | | | | | |
|---|---|---|---|---|---|---|---|---|---|---|---|---|---|---|---|---|---|
| $\beta$, deg | Jan | Feb | Mar | Apr | May | Jun | Jul | Aug | Sep | Oct | Nov | Dec | Spring | Summer | Autumn | Winter | Year |
| 0 | 81.42 | 110.84 | 152.87 | 188.14 | 238.55 | 235.89 | 242.28 | 220.96 | 174.88 | 136.86 | 80.18 | 75.00 | 579.56 | 699.13 | 391.92 | 267.26 | 1937.88 |
| 5 | 87.91 | 119.07 | 158.98 | 191.36 | 238.73 | 233.87 | 241.04 | 223.72 | 181.77 | 146.59 | 86.19 | 82.62 | 589.07 | 698.63 | 414.55 | 289.60 | 1991.85 |
| 10 | 93.95 | 126.62 | 164.21 | 193.55 | 237.68 | 230.77 | 238.62 | 225.19 | 187.56 | 155.45 | 91.74 | 89.78 | 595.44 | 694.58 | 434.75 | 310.34 | 2035.11 |
| 15 | 99.47 | 133.42 | 168.53 | 194.68 | 235.36 | 226.56 | 234.98 | 225.37 | 192.21 | 163.37 | 96.79 | 96.42 | 598.57 | 686.92 | 452.38 | 329.31 | 2067.17 |
| 20 | 104.45 | 139.43 | 171.90 | 194.75 | 231.78 | 221.25 | 230.13 | 224.23 | 195.67 | 170.30 | 101.32 | 102.49 | 598.43 | 675.61 | 467.29 | 346.36 | 2087.69 |
| 25 | 108.83 | 144.60 | 174.29 | 193.75 | 226.94 | 214.85 | 224.07 | 221.78 | 197.93 | 176.18 | 105.27 | 107.95 | 594.98 | 660.70 | 479.38 | 361.38 | 2096.45 |
| 30 | 112.60 | 148.89 | 175.69 | 191.68 | 220.88 | 207.40 | 216.85 | 218.02 | 198.95 | 180.97 | 108.63 | 112.76 | 588.26 | 642.27 | 488.55 | 374.25 | 2093.33 |
| 35 | 115.72 | 152.27 | 176.09 | 188.57 | 213.64 | 198.95 | 208.50 | 212.99 | 198.75 | 184.63 | 111.36 | 116.88 | 578.50 | 620.44 | 494.73 | 384.78 | 2078.33 |
| 40 | 118.16 | 154.71 | 175.49 | 184.43 | 205.26 | 189.56 | 199.09 | 206.72 | 197.31 | 187.13 | 113.45 | 120.27 | 565.18 | 595.36 | 497.88 | 393.15 | 2051.56 |
| 45 | 119.92 | 156.20 | 173.88 | 179.29 | 195.82 | 179.29 | 188.69 | 199.25 | 194.64 | 188.45 | 114.87 | 122.92 | 548.99 | 567.24 | 497.96 | 399.03 | 2013.23 |
| 50 | 120.96 | 156.72 | 171.29 | 173.20 | 185.39 | 168.24 | 177.40 | 190.66 | 190.78 | 188.58 | 115.63 | 124.80 | 529.87 | 536.30 | 494.99 | 402.48 | 1963.65 |
| 55 | 121.30 | 156.27 | 167.72 | 166.21 | 174.05 | 156.51 | 165.30 | 181.01 | 185.74 | 187.53 | 115.71 | 125.90 | 507.98 | 502.82 | 488.98 | 403.46 | 1903.25 |
| 60 | 120.91 | 154.85 | 163.21 | 158.36 | 161.92 | 144.21 | 152.51 | 170.37 | 179.58 | 185.30 | 115.11 | 126.21 | 483.50 | 467.10 | 479.98 | 401.97 | 1832.55 |
| 65 | 119.82 | 152.47 | 157.80 | 149.73 | 149.10 | 131.46 | 139.16 | 158.86 | 172.32 | 181.91 | 113.84 | 125.72 | 456.63 | 429.48 | 468.06 | 398.01 | 1752.19 |
| 70 | 118.02 | 149.15 | 151.52 | 140.39 | 135.72 | 118.41 | 125.40 | 146.56 | 164.03 | 177.37 | 111.90 | 124.45 | 427.63 | 390.36 | 453.31 | 391.63 | 1662.92 |
| 75 | 115.53 | 144.93 | 144.42 | 130.40 | 121.93 | 105.22 | 111.38 | 133.59 | 154.78 | 171.74 | 109.32 | 122.40 | 396.76 | 350.19 | 435.84 | 382.86 | 1565.64 |
| 80 | 112.37 | 139.82 | 136.56 | 119.87 | 107.90 | 92.10 | 97.32 | 120.10 | 144.63 | 165.05 | 106.10 | 119.58 | 364.33 | 309.51 | 415.78 | 371.77 | 1461.39 |
| 85 | 108.56 | 133.86 | 127.99 | 108.88 | 93.84 | 79.31 | 83.46 | 106.23 | 133.66 | 157.34 | 102.29 | 116.02 | 330.71 | 268.99 | 393.29 | 358.45 | 1351.44 |
| 90 | 104.13 | 127.12 | 118.78 | 97.55 | 80.01 | 67.22 | 70.15 | 92.17 | 121.96 | 148.69 | 97.89 | 111.75 | 296.34 | 229.54 | 368.55 | 343.00 | 1237.42 |

**Table 16.** Monthly total energy received by an inclined surface for each month of the year and for different slope angles ($\beta$) ranging from 20° to 32° at Athalassa. The optimum slope angles for each month are highlighted.

| Slope | $G_M$ (kWh/m²) | | | | | | | | | | | | | | | | |
|---|---|---|---|---|---|---|---|---|---|---|---|---|---|---|---|---|---|
| $\beta$, deg | Jan | Feb | Mar | Apr | May | Jun | Jul | Aug | Sep | Oct | Nov | Dec | Spring | Summer | Autumn | Winter | Year |
| 20 | 104.45 | 139.43 | 171.90 | 194.75 | 231.78 | 221.25 | 230.13 | 224.23 | 195.67 | 170.30 | 101.32 | 102.49 | 598.43 | 675.61 | 467.29 | 346.36 | 2087.69 |
| 21 | 105.37 | 140.53 | 172.45 | 194.64 | 230.91 | 220.06 | 229.01 | 223.85 | 196.22 | 171.56 | 102.15 | 103.63 | 598.00 | 672.92 | 469.94 | 349.54 | 2090.39 |
| 22 | 106.27 | 141.60 | 172.97 | 194.48 | 229.99 | 218.82 | 227.85 | 223.41 | 196.72 | 172.78 | 102.97 | 104.75 | 597.44 | 670.08 | 472.47 | 352.62 | 2092.62 |
| 23 | 107.15 | 142.63 | 173.45 | 194.28 | 229.03 | 217.54 | 226.64 | 222.92 | 197.17 | 173.96 | 103.76 | 105.84 | 596.76 | 667.10 | 474.89 | 355.63 | 2094.37 |
| 24 | 108.00 | 143.63 | 173.89 | 194.03 | 228.01 | 216.22 | 225.38 | 222.37 | 197.57 | 175.09 | 104.53 | 106.91 | 595.94 | 663.97 | 477.19 | 358.55 | 2095.65 |
| 25 | 108.83 | 144.60 | 174.29 | 193.75 | 226.94 | 214.85 | 224.07 | 221.78 | 197.93 | 176.18 | 105.27 | 107.95 | 594.98 | 660.70 | 479.38 | 361.38 | 2096.45 |
| 26 | 109.64 | 145.53 | 174.65 | 193.42 | 225.83 | 213.45 | 222.72 | 221.13 | 198.23 | 177.23 | 105.99 | 108.97 | 593.90 | 657.30 | 481.45 | 364.13 | 2096.78 |
| 27 | 110.42 | 146.42 | 174.97 | 193.05 | 224.67 | 212.00 | 221.32 | 220.43 | 198.49 | 178.23 | 106.69 | 109.96 | 592.69 | 653.75 | 483.40 | 366.79 | 2096.63 |
| 28 | 111.17 | 147.28 | 175.25 | 192.63 | 223.45 | 210.51 | 219.88 | 219.68 | 198.69 | 179.19 | 107.36 | 110.92 | 591.34 | 650.06 | 485.24 | 369.37 | 2096.01 |
| 29 | 111.90 | 148.10 | 175.49 | 192.18 | 222.19 | 208.98 | 218.39 | 218.87 | 198.85 | 180.10 | 108.00 | 111.85 | 589.87 | 646.24 | 486.95 | 371.85 | 2094.91 |

**Table 16.** *Cont.*

| Slope | $G_M$ (kWh/m$^2$) | | | | | | | | | | | | | | | | |
|---|---|---|---|---|---|---|---|---|---|---|---|---|---|---|---|---|---|
| $\beta$, deg | Jan | Feb | Mar | Apr | May | Jun | Jul | Aug | Sep | Oct | Nov | Dec | Spring | Summer | Autumn | Winter | Year |
| 30 | 112.60 | 148.89 | 175.69 | 191.68 | 220.88 | 207.40 | 216.85 | 218.02 | 198.95 | 180.97 | 108.63 | 112.76 | 588.26 | 642.27 | 488.55 | 374.25 | 2093.33 |
| 31 | 113.28 | 149.64 | 175.85 | 191.14 | 219.53 | 205.79 | 215.27 | 217.11 | 199.01 | 181.79 | 109.22 | 113.64 | 586.52 | 638.17 | 490.03 | 376.56 | 2091.28 |
| 32 | 113.93 | 150.35 | 175.97 | 190.56 | 218.13 | 204.14 | 213.64 | 216.16 | 199.02 | 182.57 | 109.80 | 114.49 | 584.66 | 633.94 | 491.38 | 378.77 | 2088.75 |

**Table 17.** Monthly total energy (kWh/m$^2$/month) received by an inclined surface for each month of the year and for different slope angles ($\beta$). The percentage of the increase of annual energy with respect to horizontal is shown in the last column of the table. The first row represents the optimum slope for each month of the year, while $\beta = 26°$ represents a permanent angle for the whole year.

| $\beta$, deg | Jan | Feb | Mar | Apr | May | Jun | Jul | Aug | Sep | Oct | Nov | Dec | $G_i$ | $G_h$ | $\varepsilon$ (%) |
|---|---|---|---|---|---|---|---|---|---|---|---|---|---|---|---|
| Mly $\beta$ opt. | 55 | 50 | 35 | 20 | 5 | 0 | 0 | 15 | 30 | 50 | 55 | 60 | Annual (kWh/m$^2$) | | $(G_i\text{-}G_h)/G_h$ |
| $\beta$ opt. | 121.30 | 156.72 | 176.09 | 194.75 | 238.73 | 235.89 | 242.28 | 225.37 | 198.95 | 188.58 | 115.71 | 126.21 | 2220.58 | 1937.88 | 14.59 |
| 0 ($G_h$) | 81.42 | 110.84 | 152.87 | 188.14 | 238.55 | 235.89 | 242.28 | 220.96 | 174.88 | 136.86 | 80.18 | 75.00 | 1937.88 | 1937.88 | 0.00 |
| ($\varphi - 15$) = 20 | 104.45 | 139.43 | 171.90 | 194.75 | 231.78 | 221.25 | 230.13 | 224.23 | 195.67 | 170.30 | 101.32 | 102.49 | 2087.69 | 1937.88 | 7.73 |
| $\varphi = 35$ | 115.72 | 152.27 | 176.09 | 188.57 | 213.64 | 198.95 | 208.50 | 212.99 | 198.75 | 184.63 | 111.36 | 116.88 | 2078.33 | 1937.88 | 7.25 |
| ($\varphi + 15$) = 50 | 120.96 | 156.72 | 171.29 | 173.20 | 185.39 | 168.24 | 177.40 | 190.66 | 190.78 | 188.58 | 115.63 | 124.80 | 1963.65 | 1937.88 | 1.33 |
| 26 | 109.64 | 145.53 | 174.65 | 193.42 | 225.83 | 213.45 | 222.72 | 221.13 | 198.23 | 177.23 | 105.99 | 108.97 | 2096.78 | 1937.88 | 8.20 |

## 5. Conclusions

Ten minute and hourly data of global horizontal ($G$), direct normal ($B_n$), reflected ($R$) and diffuse ($D$) irradiances were obtained from the actinometric station of Athalassa, an inland location in Cyprus at the height of 160 m, covering the period June 2020 to May 2021. Initially, the assessment of the measured radiation components was performed, followed by the modeling of the diffuse fraction and the estimation of diffuse and direct normal irradiance based on the BRL model. Finally, the estimation of solar energy on incline surfaces was carried out.

A detailed quality control procedure was implemented on the radiation components which are based on the suggested limits proposed by the BSRN network. The tests include physically possible limits (PPL), extremely rare limits (ERL), and configurable climatological limits (CCL). The comparison tests concentrated on the ratios of $k_b$, $k_{bt}$, $k_d$, and $k_t$ and their relationships. The ratios are generally lower than 1. Finally, step and persistency tests were applied on the hourly data set. Only a few observations were considered as invalid and were excluded from the analysis. The daily values were also tested against the extraterrestrial irradiation and the estimated highest daily sums on clear days as well as by comparing the daily global radiation with the daily sunshine duration.

From this investigation the following results can be highlighted:

Monthly mean hourly values of shortwave irradiances are shown by means of isoline diagrams. These values are considered representative of the solar radiation behavior along a typical year and can be useful for exploiting solar energy applications. July was found to be the month with maximum values of direct horizontal and global radiation. The hourly average global irradiance fluctuates between 450 W/m$^2$ in January and 950 W/m$^2$ in July at local noon. The respective ranges of direct horizontal irradiances are 230 W/m$^2$ and 800 W/m$^2$, while the diffuse irradiances fluctuate between 120 W/m$^2$ and 240 W/m$^2$.

The BRL model was used to obtain hourly diffuse radiation and from that the estimation of the direct normal solar radiation. Considering the error analysis, the results showed that the BRL model can estimate satisfactory both the diffuse solar irradiance as well as the direct normal irradiance. The model was developed based on a one-year dataset. The estimated parameters were close to those obtained by Ridley et al. at Adelaide of Australia, i.e., a location with similar climatic conditions with Cyprus.

The annual average daily global radiation intensity is around 19 MJ/m$^2$, whereas the horizontal beam and diffuse radiation is 12.9 MJ/m$^2$ and 4.7 MJ/m$^2$, respectively. Consequently, the fraction of the beam component of the global radiation is relatively high, i.e., the annual average daily fraction is $\geq 0.600$ at this site. The monthly mean daily values for the global radiation ranged between 8.7 and 28.3 MJ/m$^2$, for the direct horizontal radiation they ranged between 4.6 and 22.0 MJ/m$^2$ and for the diffuse radiation they fluctuated between 3.2 and 7.1 MJ/m$^2$.

Daily clearness index ($K_T$) is an objective measure of the influence of cloud cover on solar radiation flux. As indicated earlier, the sky conditions could be classified in four intervals of clearness index based on the relation between hourly irradiances of global, direct, and diffuse radiation and the clearness index:

Class I: Cloudy: $K_T \leq 0.35$ or $\sigma \leq 0.3$;

Class II: Partially cloudy with predominance of diffuse component: $0.35 < K_T \leq 0.55$ or $0.3 < \sigma \leq 0.6$;

Class III: Partially cloudy with predominance of direct component: $0.55 < K_T \leq 0.65$ or $0.6 < \sigma \leq 0.85$;

Class IV: Clear sky: $K_T > 0.65$ or $\sigma > 0.85$.

The annual average of $K_T$ is 0.620 with a standard deviation of 0.121. The average values of the daily clearness index range between 0.497 in November to 0.692 in July. The annual number of cloudy days is 17 (4.66%), the respective number of partially cloudy days is 137 (37.53%) and 211 days are classified as clear days representing the 57.81% of the annual number of days. As can be seen, the summer months are classified mainly as clear days.

The annual average of the direct normal beam radiation on clear days is 27.3 MJ/m², while the respective value of direct horizontal is 17.7 MJ/m². The annual average of diffuse radiation is 4.2 MJ/m², while the global radiation is 23.9 MJ/m². Finally, the annual mean of daily sunshine duration on clear days is 11.5 h. The major difference between day types according to $K_T$, is the significant reduction in the beam radiation which also affects the global irradiation level. For example, the annual average daily direct normal radiation available for energy conversion on partially cloudy days is reduced by a factor of 0.61 relative to that available on clear days. The respective reduction of global radiation is 0.65. Generally, the summer season is characterized by relatively small variations in the global radiation, horizontal beam radiation and their corresponding ratios. This means that a relatively constant solar energy source is available for energy conversion systems during this period.

Regarding the estimation of solar energy on incline surfaces, it was found that the optimum slope angles during the winter-time range between 50 and 60°, while during summer the slopes are close to zero. In spring the optimum slope angle is 15°, in summer is 0°, in autumn is 45° and in winter is 55°. The annual optimum angle is about 26° which is about 10° less than the latitude of the location. For different optimum slopes for each month of the year, the annual amount of energy is equal of 2221 kWh/m², while the measured total horizontal energy is 1938 kWh/m². Therefore, the percentage increase of energy is 14.6%. For the case of a permanent slope angle of 26° throughout the year the inclined panel receives 2097 kWh/m² with the increase of about 8.2% from the horizontal level. For the case the slope $\beta = \varphi$ (35°) we have high energy production during the months of March and September. The annual increase in energy with respect to the horizontal level is 7.3%. Therefore, higher total energy can be obtained if we use two different slopes one for the summer and one for winter period. For instance, if we use a slope of 10° for the summer period (April–September), then the total energy for this period is 1313 kWh/m², while if we use a slope of 50° for the winter period (October–March), then the total energy for this period is 878 kWh/m². Therefore, the annual total by using the two slopes during the year is 2191 kW/m², which is higher by about 95 kWh/m² from the case of an annual permanent slope of 26°. As a result, the case with two slopes ($\beta = 10°$ for summer and $\beta = 50°$ for winter) could have an increase of total annual energy by 13% with respect to the horizontal one.

The study has shown that the above methodology of analysis gives valuable information concerning the application of solar radiation in renewable energy resources projects.

**Author Contributions:** Investigation, S.P.; Methodology, A.P.; Resources, S.P.; Supervision, S.A.K. All authors have read and agreed to the published version of the manuscript.

**Funding:** This research received no external funding.

**Institutional Review Board Statement:** Not applicable.

**Informed Consent Statement:** Not applicable.

**Data Availability Statement:** These are confidential from the Meteorological Department.

**Acknowledgments:** The authors would like to thank the Meteorological Department of Cyprus for providing the meteorological data.

**Conflicts of Interest:** The authors declare no conflict of interest.

## Nomenclature

| | |
|---|---|
| *2AP* | Solar Tracker System |
| *B* | Direct (Beam) horizontal irradiance (BHI, W/m²) |
| $B_d$ | Daily Direct (Beam) horizontal irradiation (MJ/m²) |
| $B_n$ | Direct (Beam) normal irradiance (BNI, W/m²) |
| $B_{nd}$ | Daily Direct normal irradiation (MJ/m²) |
| $B_{iM}$ | Total monthly direct energy received by an incline surface (kWh/m²/month) |

| | |
|---|---|
| *BSRN* | Baseline Surface Radiation Network |
| *CCL* | Configurable Climatological Limits |
| *CDF* | Cumulative Density Function |
| $d_n$ | Day number of the year (1..365) |
| *D* | Diffuse horizontal irradiance (DHI, W/m$^2$) |
| $D_{iM}$ | Total monthly diffuse energy received by an incline surface (kWh/m$^2$/month) |
| *DPAR* | Diffuse Photosynthetic Active Irradiance (W/m$^2$) |
| $DPAR_d$ | Daily Diffuse Photosynthetic Active Irradiation (MJ/m$^2$) |
| $E_p$ | Pan A Evaporation (mm/day) |
| *ECV* | Essential Climate variable |
| *ERL* | 'Extremely rare' (ERL) limits (W/m$^2$) |
| *ESRA* | European Solar Radiation Atlas |
| *ETR* | Extraterrestrial Radiation |
| *G* | Global solar horizontal irradiance (GHI, W/m$^2$) |
| $G_d$ | Daily global solar irradiation (MJ/m$^2$) |
| $G_m$ | Monthly mean global solar irradiation (MJ/m$^2$) |
| $G_0$ | Extraterrestrial horizontal irradiance (W/m$^2$) |
| $G_{0d}$ | Daily extraterrestrial irradiation (ETR) (MJ/m$^2$) |
| $G_{0n}$ | Extraterrestrial normal irradiance (W/m$^2$) |
| $G_c$ | Clear-sky global solar irradiance (W/m$^2$) |
| $G_{cd}$ | Daily global irradiation for clear-sky conditions (MJ/m$^2$) |
| $G_{sc}$ | Solar constant (1367 Wm$^2$) |
| $G_{max}$ | Highest daily global solar irradiance (W/m$^2$) |
| $G_{hM}$ | Total monthly solar energy received by a horizontal surface (kWh/m$^2$/month) |
| $G_{iM}$ | Total monthly solar energy received by an inclined surface (kWh/m$^2$/month) |
| *GPPFD* | Total Photosynthetic Photon Flux Density (μmol/s/m$^2$) |
| *GPAR* | Total Photosynthetic Active Irradiance (W/m$^2$) |
| $GPAR_d$ | Daily Total Photosynthetic Active Irradiation (MJ/m$^2$) |
| *IQR* | Interquartile range |
| $k_d$ | Diffuse fraction (D/G) |
| $K_D$ | Daily Diffuse fraction (D$_d$/G$_d$) |
| $k_{dt}$ | Diffuse clearness index (D/G$_0$) |
| $k_b$ | Direct fraction (B/G) |
| $K_B$ | Daily Direct fraction (B$_d$/G$_d$) |
| $k_{bt}$ | Direct transmittance (B/G$_0$) |
| $K_{BT}$ | Daily Direct transmittance (B$_d$/G$_{0d}$) |
| $k_n$ | Direct Normal clearness index (B$_n$/G$_{0n}$) |
| $K_N$ | Daily Direct Normal clearness index (B$_{nd}$/G$_{0d}$) |
| $k_t$ | Clearness index (G/G$_0$) |
| $K_T$ | Daily clearness index (G$_d$/G$_{0d}$) |
| *LF* | Conversion factor for PAR irradiance (4.57 μmol s$^{-1}$ W$^{-1}$) [19] |
| *LT* | Local time (t) |
| *m* | Optical air mass |
| *Max* | Maximum |
| *Min* | Minimum |
| *MMH* | Mean Monthly Hourly |
| *n* | Number of observations |
| *N* | Number of days |
| *NIR* | Near Infrared Radiation (W/m$^2$) |
| $P_{rs}$ | Atmospheric Pressure (P, hPa) |
| $P_0$ | Standard atmospheric pressure (1013.25 hPa) |
| *PDF* | Probability Density Function |
| *PPL* | 'Physically possible' limits (W/m$^2$) |
| $P/P_0$ | Pressure correction for station height |
| *QC* | Quality control |
| *Q1* | First Quartile |
| *Q3* | Third Quartile |

| $R$ | Reflected horizontal irradiance (RHI, W/m$^2$) |
|---|---|
| $R_d$ | Daily Reflected irradiation (MJ/m$^2$) |
| $R_B$ | Beam radiation conversion factor |
| $R_D$ | Diffuse radiation conversion factor |
| $R_R$ | Reflect radiation conversion factor |
| $R_M$ | Monthly radiation conversion factor |
| $R_{iM}$ | Total monthly reflected energy received by an incline surface (kWh/m$^2$/month) |
| $R_L$ | Rayleigh diffuse limit (W/m$^2$) |
| $R_{ns}$ | Net Short-wave irradiance (W/m$^2$) |
| $R^2$ | Coefficient of determination |
| $S$ | Standard deviation of residuals |
| $S_d$ | Daily sunshine duration (hours) |
| $S_{0d}$ | Astronomical day length (hours) |
| $SD$ | Sunshine duration (S, hours) |
| $StDev$ | Standard deviation (Std) |
| $SAL$ | Solar Altitude Angle ($\alpha_s$, deg) |
| $SAZ$ | Solar Azimuth Angle (deg) |
| $SR$ | Solar Radiation |
| $SW$ | Shortwave radiation |
| $SZA$ | Solar Zenith Angle (deg) |
| $T_a(°C)$ | Air temperature at screen level (°C), $T_a$(K) = 273.16 + $T_a$(°C) |
| $UVT$ | UV total radiation (UVB + UVA) |
| WMO | World Meteorological Organization |
| $z$ | Station's elevation (m) |
| **Greek:** | |
| $\alpha_s$ | Solar altitude angle (degrees) |
| $\beta$ | Slope of inclined surface with respect to horizontal level (degrees) |
| $\beta_{opt}$ | Optimum slope angle (degrees) |
| $\delta$ | Solar declination angle (degrees) |
| $\varepsilon$ | Correction factor to mean solar distance |
| $\theta_z$ | Solar zenith angle (SZA) (degrees) |
| $\lambda$ | Longitude of the station in degrees (East positive) / Wavelength (nm) |
| $\lambda_{ST}$ | Reference longitude of the time zone in degrees (for Cyprus= 30°) |
| $\mu$ | cos(SZA) |
| $\sigma$ | Relative sunshine duration ($S_d/S_{0d}$) |
| $\varphi$ | Latitude of the station in degrees |
| $\psi$ | Average percistence of both a lag and lead of $k_t$ of each time interval of the day |
| $\omega$ | Hour angle (degrees) |
| $\omega_s$ | Sunset hour angle (degrees) |
| $\omega_s'$ | Sunset hour angle of an inclined surface (degrees) |

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
