# Peer review of "Shortwave Radiation on Horizontal and Incline Surfaces—One Year of Solar Radiation Measurements at Athalassa, an Inland Location in Cyprus"

_applsci, doi:10.3390/app122111035_

Round 1

Reviewer 1 Report

Paper analyses the characteristics of solar radiation in a location based on sky conditions classified according to kt, calculates the NIR and evaluates the efficiency of some old decomposition models for calculating radiation on inclined planes for photovoltaic estimation. It is an interesting article since it includes experimental data for a complete year, in a characteristic location of the Mediterranean climate and for the entire radiation spectrum. Novelty of the research is low because there are similar data in different locations on the planet, but it increases the number of data for checking decomposition models. The article includes three objectives that are perhaps too different from each other to be part of the same paper. The set of experimental data and the analysis of their quality according to internationally accepted criteria (BSNR and NREL), the calculation and analysis of the NIR and the application of decomposition models for the calculation of irradiance on tilted surfaces and their application to photovoltaic production. Perhaps, paper should be focused only on models for tilted planes or maybe on the calculation of NIR (although it does not refer to experimental measurements to check the results). I would recommend the authors to rethink the objectives of the article. On the other hand, the models used are very old (although they work, there is no doubt), and it would be more interesting to check new models. In addition, a number of details should clarify in the text:

·       In table 1, the list of existing sensors in the station is included, but in figure 4 GUV is represented (I assume it is the total ultraviolet radiation, but it is the first time it appears in the text), and it is not known if they are experimental data from the same station or another.

·       The quality of figures and tables should be improved

·       The format of the references is not the one indicated by the journal

·       The abstract does not reflect the most important points of the research

·       The introduction should explain the motivation of the research and the relationship between all the aspects included in the article.

Author Response

Reviewer 1.

  1. The paper was shortened and reorganized.
  2. The abstract and conclusions were reorganized
  3. References were formatted according to the Journal requirements
  4. Suggested corrections for UVT were accepted.
  5. Objectives
  6. Introduction
  7. Figures and Tables

Reviewer 2 Report

The whole paper should be rewritten and reorganized in order to reduce the page numbers. However, the topic and provided content is interesting for readers and contain valuable information.

Abstract and Conclusions must be rewritten more synthetical. The quality of certain picture should be improved (Fig 2, 3, 27 etc).

References are pretty old, and not proper formatted. Newer relevant references must be considered by authors.

 There are a lot of numerical coefficients in the presented equations, is not clear in all situations if they are empirically determined, by curve fitting, regression or other means. This should be clearer presented.

Author Response

Reviewer 2.

  1. The paper was shortened and reorganized.
  2. References were formatted according to the Journal requirements
  3. Abstract and conclusions rewritten more synthetically
  4. Empirical coefficients
  5. Newer references

Reviewer 3 Report

Dear Author/s,
Beside the well-managed paper there are some critical point which requires your attention.

At this stage i would suggest a major revision.

please find attached file.

Regards

Author Response

Reviewer 3.

  1. The paper was shortened and reorganized.
  2. Underline text with yellow?

Round 2

Reviewer 1 Report

None of the suggestions and comments of the reviewer have been taken into account in this second round, so I have to repeat my previous comments:

Paper analyses the characteristics of solar radiation in a location based on sky conditions classified according to kt, calculates the NIR and evaluates the efficiency of some old decomposition models for calculating radiation on inclined planes for photovoltaic estimation. It is an interesting article since it includes experimental data for a complete year, in a characteristic location of the Mediterranean climate and for the entire radiation spectrum. Novelty of the research is low because there are similar data in different locations on the planet, but it increases the number of data for checking decomposition models. The article includes three objectives that are perhaps too different from each other to be part of the same paper. The set of experimental data and the analysis of their quality according to internationally accepted criteria (BSNR and NREL), the calculation and analysis of the NIR and the application of decomposition models for the calculation of irradiance on tilted surfaces and their application to photovoltaic production. Perhaps, paper should be focused only on models for tilted planes or maybe on the calculation of NIR (although it does not refer to experimental measurements to check the results). I would recommend the authors to rethink the objectives of the article. On the other hand, the models used are very old (although they work, there is no doubt), and it would be more interesting to check new models. In addition, a number of details should clarify in the text:

·       In table 1, the list of existing sensors in the station is included, but in figure 4 GUV is represented (I assume it is the total ultraviolet radiation, but it is the first time it appears in the text), and it is not known if they are experimental data from the same station or another.

·       The quality of figures and tables should be improved

·       The format of the references is not the one indicated by the journal

·       The abstract does not reflect the most important points of the research

·       The introduction should explain the motivation of the research and the relationship between all the aspects included in the article.

Author Response

Reviewer 1.

  1. We understand the points and the views but no such study is available for Cyprus. This is not really novel but it is a valuable information for the local engineers and scientists.
  2. UVT Point addressed,
  3. Quality of Figures- For the statistical analysis and the graphs, Minitab program was used. Therefore, we cannot do anything more. The quality of the figures is reduced in cases when more than two graphs were inserted in order to reduce the size of the paper. The main purpose of the figures at least for the quality control, is to show that the observations are within the suggested BSRN limits.
  4. Format of references. Done
  5. Abstract – According to Reviewer 2, the abstract is further reduced. We believe that these are clearly shown.
  6. Introduction – Additional more recent references were added. The study has shown that the methodology of analysis gave valuable information concerning the application of solar radiation in renewable energy resources projects.

Reviewer 2 Report

The paper look more like a technical report than a scientific paper.

The Abstract and Conclusion sections are too long and must be reduced, and should contain only relevant information.

The whole paper has too many pages,  the resubmitted version is not reduced, but use a wider format that can lead to more pages when the paper is reformatted according to mdpi requirements.

Try to remove unnecessary/outdated/well known information and present especially author own findings.

References should be revised. For example I am not sure that an introductory book published 40 years ago is relevant for a scientific paper in 2022  (20. Iqbal M., 1983. An introduction of solar radiation. Academic Press)

The entire content of paper should be carefully and deeply  revised before publishing.

Author Response

Reviewer 2.

  1. More recent references were added using the format required by the Journal. Regarding the reference from Iqbal (1983) we use the equations which are still valid.
  2. Abstract and Conclusions were reduced.
  3. The whole paper was revised and reduced by 10 pages.

Reviewer 3 Report

Dear Author/s,

Manuscript has been improved well, however some small corrections are still required to be done.

Find attached file.

Regards,

Author Response

Reviewer 3.

  1. All points addressed by the reviewer were addressed.
  2. New recent references were added.
  3. Regarding the temperature of 40 0C it is considered that above this temperature heat stress will affect people.
  4. We left the equations [12-15] in one line in order to reduce the size of the paper.
  5. In some figures there are 2 or more graphs in order to reduce the size of the paper.

Round 3

Reviewer 1 Report

paper is ready to be published in the present form

Reviewer 2 Report

The paper has been improved and can be accepted for publishing, however further reduction of the page number is possible by removing unnecessary or well-known information.